# Accelerated Diffusion Models via Speculative Sampling

Valentin De Bortoli [* 1]   Alexandre Galashov [* 1]   Arthur Gretton [1]   Arnaud Doucet [1]

## Abstract

Speculative sampling is a popular technique for accelerating inference in Large Language Models by generating candidate tokens using a fast draft model and then accepting or rejecting them based on the target model's distribution. While speculative sampling was previously limited to discrete sequences, we extend it to diffusion models, which generate samples via continuous, vector-valued Markov chains. In this context, the target model is a high-quality but computationally expensive diffusion model. We propose various drafting strategies, including a simple and effective approach that does not require training a draft model and is applicable out-of-the-box to any diffusion model. We demonstrate significant generation speedup on various diffusion models, halving the number of function evaluations while generating exact samples from the target model. Finally, we also show how this procedure can be used to accelerate Langevin diffusions to sample unnormalized distributions.

## 1. Motivation

Denoising diffusion models (DDMs), introduced by Sohl-Dickstein et al. (2015) and further developed by Ho et al. (2020) and Song et al. (2021), are generative models exhibiting state-of-the-art performance in a wide variety of domains. The core concept behind DDMs is the progressive transformation of a data distribution into a Gaussian distribution through the addition of noise. Sample generation is achieved by simulating an approximation of the time-reversal of this noising process. This requires multiple evaluations of a neural network that approximates the scores of the noising process, and typically involves simulating a Markov chain over hundreds of steps.

Since sample generation is computationally expensive, sev-

eral techniques have been proposed to accelerate it. These include distillation techniques (e.g. Salimans & Ho, 2022; Meng et al., 2023; Song et al., 2023), better sampling schemes (e.g. Karras et al., 2022; Lu et al., 2022; Zhang & Chen, 2023) and parallel simulation methods (e.g. Shih et al., 2023; Chen et al., 2024). However, distillation techniques inherently require training a student model and often underperform compared to the teacher model (Dieleman, 2024). While better sampling schemes can improve performance, using too small a number of steps does degrade performance, see e.g. (Karras et al., 2022). Finally, parallel simulation methods relying on Picard iterations over a sliding window have been proposed (Shih et al., 2023; Chen et al., 2024; Tang et al., 2024). However, they are inherently iterative, requiring repeated parallel sampling within a window until errors fall below a pre-specified tolerance.

In the context of Large Language Models (LLMs), various techniques have also been proposed to speed up inference. Notably, *speculative sampling*, first introduced by Leviathan et al. (2023) and later proposed independently by Chen et al. (2023), has become prominent in this area and has spawned numerous extensions (Xia et al., 2024). Given a target LLM, this algorithm enables faster sampling than serial token decoding without compromising quality, as the sampled tokens remain exactly distributed according to the target model's distribution. This is achieved by considering a smaller and faster LLM model generating a draft sequence. The target model is then used to compute *in parallel* the conditional probabilities of these draft tokens, and these probabilities are used to decide sequentially whether to accept or reject the draft tokens. Upon the first rejection, a new token is sampled using an adjusted distribution combining the draft and target distributions. Many extensions of speculative sampling have been proposed to reduce latency; see Xia et al. (2024) and further related works in Section 6.

In the present work, we adapt speculative sampling to accelerate DDMs. We assume a computationally cheap draft model that generates a sequence of draft states for the denoising Markov chain of a target DDM. The transition probability densities of these states under the target model are then computed *in parallel* and used to sequentially accept/reject the draft states. At rejection, a new state is sampled from an adjusted distribution dependent on both the draft and target distributions; see Figure 1. As for LLMs, the pro-

[*]Equal contribution   [1]Google DeepMind. Correspondence to: Valentin De Bortoli <vdebortoli@google.com>.

*Proceedings of the 42$^{nd}$ International Conference on Machine Learning*, Vancouver, Canada. PMLR 267, 2025. Copyright 2025 by the author(s).

cedure is designed such that it outputs samples distributed exactly according to the target DDM.

Wang et al. (2024) concurrently proposed an adaptation of speculative sampling for continuous-valued autoregressive processes, specifically for Masked Autoregressive models (Li et al., 2024b). In this setting, they sample from the adjusted distribution appearing at rejection using a standard rejection sampling algorithm. However, as demonstrated in Section 3.2, this approach is, on average, more computationally expensive than directly sampling from the target model in our context. Furthermore, it exhibits counter-intuitive performance degradation as the draft model more closely approximates the target model. We present a method to circumvent these issues while retaining the optimality properties of speculative sampling. Our contributions are summarized below. Proofs are in the Supplementary Material.

- By leveraging the connections between speculative sampling and coupling techniques (Lindvall, 1992), first observed by Sun et al. (2023) in the context of LLMs, we show in Section 3.3 that we can sample efficiently from a novel adjusted distribution for DDMs using reflection maximal coupling (Bou-Rabee et al., 2020). Our procedure returns exact samples from the target model, and it is optimal in the sense that it maximizes the probability of accepting each draft state.

- We investigate several drafting strategies (Section 3.1 and Appendix B). As with LLMs, one can rely on a "cheap" diffusion model as draft model, or use a draft model learned from the target model. We propose here instead a simple and effective approach that proposes a draft model relying solely on the target model. This eliminates any need for learning a separate draft model, and is readily applicable to any diffusion model.

- We present a complexity analysis and a lower bound on the acceptance ratio of the draft states in Section 4 .

- We explain in Section 5 how this method can be adapted to accelerate Langevin diffusions to sample unnormalized distributions.

- The proposed method achieves significant speed-ups for image generation on CIFAR10, and LSUN using pixel space diffusion models, without any loss of quality (Section 7). Furthermore, we show similar speed-ups in robotics for policy generation.

## 2. Speculative Sampling for LLMs

We begin with a review of speculative sampling for LLMs. Consider two probability distributions $q$ and $p$ for sequences on some finite space $\mathcal{X}$. In this context, $q$ corresponds to the joint distribution of tokens for the target LLM, while $p$ represents the draft model.

### 2.1. Speculative Sampling for Autoregressive Targets

Speculative sampling generates $L$ candidate tokens according to the draft model $p$ which are scored in parallel using the target model $q$. They are then accepted sequentially using an adjusted rejection sampling algorithm. At the first rejection, one needs to sample a new token from an adjusted distribution denoted $r$. A new set of $L$ candidate tokens is then generated, and so on. This is detailed in Algorithm 1 using notation $z_{k:\ell} = (z_k, z_{k+1}, ..., z_\ell)$ for $k \leq \ell$ and $z_{k:\ell} = \emptyset$ for $k > \ell$ for any sequence $(z_k)_{k \in \mathbb{N}}$ and $[k] = \{1, ..., k\}$ for any positive integer $k$. We denote *sequential* computations by **(Seq.)** and *parallel* computations by **(Par.)**.

---

**Algorithm 1** Speculative Sampling for LLM

---

**Require:** Lookahead integer $L$, maximum length $K$, draft model $p$, target model $q$, initial context $X_{0:n_0}$.

Set $n \leftarrow n_0$

**while** $n < n_0 + K$ **do**

**(Seq.)** Sample $\tilde{X}_{n+1:n+L} \sim p(\cdot|X_{0:n})$

Get $p_{n+j} = p(\cdot|X_{0:n}, \tilde{X}_{n+1:n+j-1})$, $j \in [L]$.

**(Par.)** Get $q_{n+j} = q(\cdot|X_{0:n}, \tilde{X}_{n+1:n+j-1})$, $j \in [L]$.

    **for** $k = n+1 : n+L$ **do**

        $(X_k, \texttt{bool}) \leftarrow \texttt{REJECTION}\,(p_k, q_k, \tilde{X}_k)$

        **if** $\texttt{not(bool)}$ or $X_k = \texttt{EOS}$ **then**

            Exit For Loop

        **end if**

    **end for**

    Set $n \leftarrow k$

**end while**

**return** $X_{n_0+1:n}$

---

The rejection mechanism is described in Algorithm 2.

---

**Algorithm 2** REJECTION $(p, q, \tilde{X})$

---

**Require:** Proba. distributions $p, q$ and $\tilde{X} \sim p$.

Sample $U \sim \text{Unif}[0, 1]$.

$\texttt{bool} = \mathbb{I}[U \leq \min(1, q(\tilde{X})/p(\tilde{X}))]$.

**if** $\texttt{bool}$ **then**

    Set $X = \tilde{X}$.

**else**

    $X \sim r(\cdot)$, $r(x) \propto \max(0, q(x) - p(x))$

**end if**

**return** $(X, \texttt{bool})$ where $X \sim q$.

---

In (Chen et al., 2023; Leviathan et al., 2023), the draft sequence is sampled using a "cheap" autoregressive LLM, i.e, $\tilde{X}_{n+1} \sim p(\cdot|X_{0:n})$, $\tilde{X}_{n+2} \sim p(\cdot|X_{0:n}, \tilde{X}_{n+1})$, ..., $\tilde{X}_{n+L} \sim p(\cdot|X_{0:n}, \tilde{X}_{n+1:n+L-1})$. However, this does not have to be the case, and any distribution $p(x_{n+1:n+L}|x_{0:n})$ can be used, e.g., in Medusa (Cai et al., 2024) one samples the draft tokens in parallel by considering a factorized draft distribution $p(x_{n+1:n+L}|x_{0:n}) =$

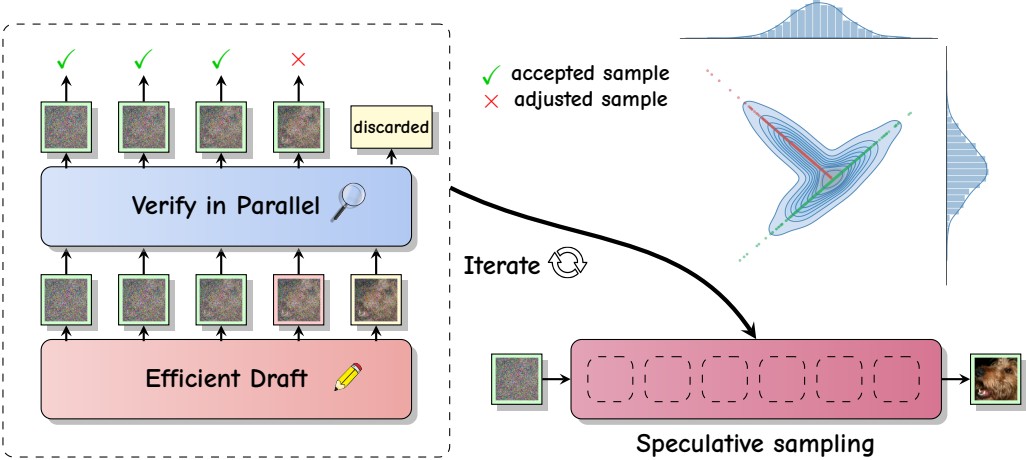

*Figure 1.* Speculative Sampling for diffusion models. Draft states are efficiently generated and verified in parallel. Upon the first rejection, a new state is sampled using an adjusted distribution combining draft & target models, and the remainder of the draft sequence is discarded.

$\prod_{k=n+1}^{n+L} p(x_k | x_{0:n})$. Note that in this context, the distributions $\{p(x_{n+1:n+L} | x_{0:n})\}_{n \geq n_0}$ are usually not compatible, i.e., they are not the conditional distributions of a joint distribution. To be more precise, we should write $p_n(x_{n+1:n+L} | x_{0:n})$ instead of $p(x_{n+1:n+L} | x_{0:n})$ but we slightly abuse notation here.

## 2.2. Adjusted Rejection Sampling as Maximal Coupling

At the core of speculative sampling lies an adjusted rejection sampling mechanism which allows for sampling from the (conditional) distribution of a token $q(x) := q(x | \text{past tokens})$ for a target LLM given the (conditional) distribution $p(x) := p(x | \text{past tokens})$ of a token for a draft model.

As pointed out by Sun et al. (2023), this procedure, summarized in Algorithm 2, is well-known in the probability literature, and the joint distribution of $(X, Y)$ it induces is a so-called maximal coupling; see e.g. (Lindvall, 1992), Section 4.5 in (Thorisson, 2000) and (Jacob, 2021) for a comprehensive introduction. Maximal couplings denote any distribution on $(X, Y)$ maximizing the probability that $X = Y$ while $X \sim p$ and $Y \sim q$. For completeness, without any claim for originality, see Proposition 2.1 for a formal statement and the supplementary material for a proof.

> **Proposition 2.1:** *Let $\tilde{X} \sim p$ then Algorithm 2 outputs $X \sim q$. This procedure is optimal in the sense that it maximizes the probability that $X = \tilde{X}$ under the constraints $\tilde{X} \sim p$, $X \sim q$. Additionally, we have*
>
> $$\mathbb{P}(X \neq \tilde{X}) = ||p - q||_{\text{TV}},$$
>
> *where $||p - q||_{\text{TV}} := \frac{1}{2} \sum_{x \in \mathcal{X}} |p(x) - q(x)|.$*

## 3. Speculative Sampling for Diffusion Models

We now present our main contribution, which is the adaptation of speculative sampling to DDMs. Our DDM target model and some drafting strategies are given in Section 3.1, leading to our speculative sampling procedure in Algorithm 3. As for LLMs, this algorithm requires an adjusted rejection sampling procedure. After analyzing the difficulties of an implementation of Algorithm 2 in the context of DDMs (Section 3.2), an original solution resolving these difficulties is presented in Section 3.3.

### 3.1. Denoising diffusion models, draft models and speculative sampling

We first define the target DDM model we want to sample from. Following Song et al. (2021), consider a *forward* noising process where $\mathbf{X}_0 \sim q_{\text{data}}$ and $d\mathbf{X}_t = f_t \mathbf{X}_t dt + g_t d\mathbf{B}_t$, where $(\mathbf{B}_t)_{t \in [0,1]}$ is a $d$-dimensional Brownian motion. Let $q_t$ the density of $\mathbf{X}_t$, we select $f_t, g_t$ such that $q_1 \approx \mathcal{N}(0, \text{Id})$. We then consider the process $(\mathbf{Y}_t)_{t \in [0,1]}$

$$d\mathbf{Y}_t = b_t(\mathbf{Y}_t) + \varepsilon g_{1-t} d\mathbf{W}_t, \quad \mathbf{Y}_0 \sim q_1, \qquad (1)$$

$$b_t(x) = -f_{1-t} x + \frac{1+\varepsilon^2}{2} g_{1-t}^2 s_{1-t}(x),$$

where $s_t(x) = \nabla \log q_t(x)$ is the *Stein score*, $(\mathbf{W}_t)_{t \in [0,1]}$ is another Brownian motion and $\varepsilon \geq 0$ is a hyperparameter which controls the stochasticity level of $(\mathbf{Y}_t)_{t \in [0,1]}$ (Albergo et al., 2023), referred to as the *churn* parameter in the literature (Karras et al., 2022). This process is such that $\mathbf{Y}_{1-t} \sim q_t$ for all $t \in [0, 1]$ and corresponds to the *time-reversal* of $(\mathbf{X}_t)_{t \in [0,1]}$ for $\varepsilon = 1$. In practice, $b_t$ is approximated using a neural network denoted $b_t^q$. At inference we consider $K + 1$ discretization steps and let $\gamma = 1/K$ and $(t_k)_{k=0}^K$ with $t_k = k\gamma$; the corresponding distribution of the resulting Markov chain obtained by the Euler–Maruyama

discretisation of (1) and initialized at $\mathcal{N}(0, \mathrm{Id}) \approx q_1$ is denoted $q(y_{0:K}) = q(y_0) \prod_{k=1}^{K} q(y_k|y_{k-1})$ where

$$q(y_k|y_{k-1}) = \mathcal{N}(y_k; m_{k-1}^q(y_{k-1}), \sigma_{k-1}^2 \mathrm{Id}), \quad (2)$$

with $q(y_0) = \mathcal{N}(y_0; 0, \mathrm{Id}) \approx q_1(y_0)$, $m_k^q(y_k) = y_k + \gamma b_{t_k}^q(y_k)$ and $\sigma_k = \sqrt{\gamma \varepsilon} g_{1-t_k}$. The distribution (2) defines the target model in our speculative sampling procedure.

Speculative sampling requires specifying a draft model. All the draft models we consider are of the form $p(y_{n+1:n_L}|y_n) = \prod_{k=n+1}^{n_L} p(y_k|y_{n:k-1})$ where $n_L = \min(n+L, K)$, $L$ is the length of the draft sequence and

$$p(y_k|y_{n:k-1}) = \mathcal{N}(y_k; m_{k-1}^p(y_{n:k-1}), \sigma_{k-1}^2 \mathrm{Id}). \quad (3)$$

**Independent draft model.** A first choice, similar to the original speculative sampling algorithm (Leviathan et al., 2023), is to consider a draft model with the same sampling strategy as $q$ but with an approximation $b_t^p$ which is cheaper to evaluate than $b_t^q$. Hence, the draft model satisfies $p(y_k|y_{n:k-1}) = p(y_k|y_{k-1})$ with

$$m_k^p(y_{n:k}) = y_k + \gamma b_{t_k}^p(y_k), \quad \sigma_k = \sqrt{\gamma \varepsilon} g_{1-t_k}. \quad (4)$$

This choice of draft requires the availability of a cheaper DDM. For $p$ and $q$ to be close and to obtain better performance (i.e., higher acceptance rate of the draft states), this requires training $p$ on the same dataset as $q$, which would be costly and might not be feasible. Even if $p$ and $q$ are trained with the same architecture on the same dataset, there can still be a significant mismatch between $b^p$ and $b^q$.

**Frozen target draft model.** Another popular choice in speculative sampling is to derive a draft model directly from the target model, see for instance (Cai et al., 2024). In the context of diffusion models, we consider here a very simple draft model where $p(y_k|y_{n:k-1}) = p(y_k|y_n, y_{k-1})$ with

$$m_k^p(y_{n:k}) = y_k + \gamma b_{t_n}^q(y_n), \quad \sigma_k = \sqrt{\gamma \varepsilon} g_{1-t_k}. \quad (5)$$

This draft model is similar to the target model, except that we replace $b_{t_k}^q(y_k)$ by $b_{t_n}^q(y_n)$. Importantly, on a window of size $L$, we only need to query the target model *once* in order to draw a draft sequence. This strategy is thus computationally inexpensive, requires no additional training and allows parallel sampling of the draft sequence. However, the differences between the draft and target models can be large near the data distribution as the score function typically exhibits significant variation. Consequently, $b_{t_n}^q(y_n)$ may deviate substantially from $b_{t_k}^q(y_k)$ when $k, n, K$ are close, rendering the approximation $b_{t_k}^q(y_k) \approx b_{t_n}^q(y_n)$ inaccurate.[1] This issue can be addressed at higher computational cost using alternative and more involved drafting procedures, as discussed in Appendix B.

---

[1] More sophisticated approximations, such as local linearization (Shoji & Ozaki, 1998), could improve accuracy but their computational cost is prohibitive in high dimension.

---

**Algorithm 3** Speculative Sampling for DDM
---
**Require:** Lookahead integer $L$, sequence length $K$, target model $q$ and draft model $p$.
    Sample $Y_0 \sim \mathcal{N}(0, \mathrm{Id})$ and set $n = 0$.
    **while** $n < K$ **do**
        Set $\tilde{Y}_n \leftarrow Y_n$
        Set $n_L \leftarrow \min(n+L, K)$ and $\tilde{L} = n_L - n$
(Seq.)  Sample draft states $\tilde{Y}_{n+1:n_L} \sim p(\cdot|\tilde{Y}_n)$ using (3).
        Get means of $p_{n+j} = p(\cdot|\tilde{Y}_{n:n+j-1})$, $j \in [\tilde{L}]$.
(Par.)  Get means of $q_{n+j} = q(\cdot|\tilde{Y}_{n+j-1})$, $j \in [\tilde{L}]$.
        **for** $k = n+1 : n_L$ **do**
            $(Y_k, \texttt{bool}) \leftarrow \texttt{REJECTION}(p_k, q_k, \tilde{Y}_k)$.
            **if** $\texttt{not}(\texttt{bool})$ **then**
                Exit For Loop
            **end if**
        **end for**
        Set $n \leftarrow k$.
    **end while**
    **return** $Y_{0:K}$

---

Having now defined the target and draft model, we present Algorithm 3, our speculative sampling algorithm for diffusion models. This algorithm is similar in principle to Algorithm 1 for LLMs. The evaluation of the means of $q(y_{n+j}|\tilde{Y}_{n:n+j-1})$ for $j \in [n_L - n]$ is done in parallel. The rejection steps within the for loop are also implemented in parallel. However, the REJECTION step in our algorithm requires a substantially different implementation compared to the one defined by Algorithm 2 used for LLMs. This difference arises because directly applying the rejection mechanism of Algorithm 2 to diffusion models presents significant challenges, as we will demonstrate.

### 3.2. Adjusted Rejection Sampling: Implementation Issues

Using Algorithm 2 to define REJECTION in Algorithm 3 would yield a valid speculative sampling algorithm for diffusion models, i.e., this algorithm would produce a Markov chain exactly distributed according to the target model, $Y_{0:K} \sim q$, and Proposition 2.1 would also apply directly.[2] However, we show below that implementing Algorithm 2 is problematic in the context of diffusion models. If a draft state is rejected at iteration $k$, where $k > n$, we must then sample $Y_k$ from

$$r(x) = \frac{\max(0, q(x) - p(x))}{\int_{\mathbb{R}^d} \max(0, q(x) - p(x)) \mathrm{d}x}, \quad (6)$$

for $q(y_k) := q(y_k|y_{k-1})$, $p(y_k) := p(y_k|y_{n:k-1})$. Although straightforward for LLMs due to the discrete nature of $r(x)$, a satisfactory solution for continuous state-spaces

---

[2] The proof of Proposition 2.1 recalled in Appendix A extends straightforwardly from $\mathcal{X}$ finite to $\mathbb{R}^d$

remains elusive. Leveraging the fact that

$$r(x) \propto q(x)(1 - \min(1, p(x)/q(x))), \qquad (7)$$

we could sample from $r(x)$ using standard rejection sampling. Using $q(x)$ as proposal, the acceptance probability is $1 - \min(1, p(x)/q(x))$ so that the average acceptance probability is

$$\int q(x)(1 - \min(1, p(x)/q(x)))\mathrm{d}x = ||p - q||_{\mathrm{TV}}.$$

This is an approach analyzed by Jacob (2021) and adopted by Wang et al. (2024) for continuous-valued autoregressive processes. From standard results on rejection sampling, it is known that the number of trials to simulate from the target $q$ before acceptance follows a geometric distribution with parameter $||p-q||_{\mathrm{TV}}$. This distribution has mean $1/||p-q||_{\mathrm{TV}}$ and variance $(1-||p-q||_{\mathrm{TV}})/||p-q||_{\mathrm{TV}}^2$ (see (Jacob, 2021) for instance). This implementation of Algorithm 2 proves inefficient, as demonstrated by the following simple analysis. With probability $||p - q||_{\mathrm{TV}}$, one needs to sample from (7) and, due to the properties of the geometric distribution, the expected number of samples from $q$ we need is $||p - q||_{\mathrm{TV}} \times (1/||p - q||_{\mathrm{TV}}) = 1$. This rejection sampling procedure is thus practically useless, as it requires sampling on average from both $p$ and $q$, as well as computing the acceptance probability $\min(1, q(x)/p(x))$. Another undesirable property of this implementation is that the variance of the number of samples from $q$ one would have to simulate increases rapidly as the draft model $p$ better approximates the target $q$ (i.e., as $||p-q||_{\mathrm{TV}}$ decreases). These issues have been extensively reported in the literature (Jacob, 2021).

### 3.3. Adjusted Rejection Sampling via Reflection-Maximal Coupling

As discussed in Section 2.2, the adjusted rejection sampling procedure from Algorithm 2 is identical to a specific maximal coupling described, for example, in (Lindvall, 1992). For DDMs, we have shown that implementing this procedure is challenging. However, it is essential to note that maximal couplings are *not* unique. Bou-Rabee et al. (2020) proposed an algorithm known as *reflection maximal coupling* to implement a maximal coupling for two Gaussian distributions $\mathcal{N}(m^p, \sigma^2\mathrm{Id})$ and $\mathcal{N}(m^q, \sigma^2\mathrm{Id})$. This is directly applicable to diffusion models, since $p(y_k|y_{n:k-1}) = \mathcal{N}(y_k; m_{k-1}^p(y_{n:k-1}), \sigma_{k-1}^2\mathrm{Id})$ and $q(y_k|y_{k-1}) = \mathcal{N}(y_k; m_{k-1}^q(y_{k-1}), \sigma_{k-1}^2\mathrm{Id})$ are Gaussian distributions with different means but identical variances. Introduced to establish convergence results for Hamiltonian Monte Carlo, this procedure is noteworthy for its conciseness and its bounded and short running time. We detail it in Algorithm 4.

Direct calculations show that the acceptance probability of the proposal $\tilde{Y} \sim \mathcal{N}(m^p, \sigma^2\mathrm{Id})$ computed with this

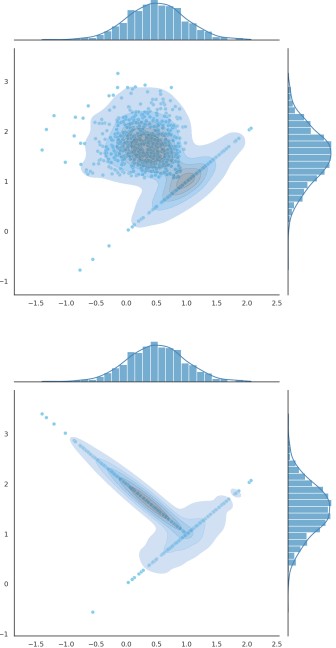

*Figure 2.* Two maximal couplings between $p = \mathcal{N}(0.5, 0.25)$ and $q = \mathcal{N}(1.5, 0.25)$: the one given by Algorithm 2 (top) and the reflection maximal coupling from Algorithm 4 (bottom). By definition, both couplings have $p$ and $q$ as their marginals. As they are maximal couplings, their probability mass on the diagonal is identical and is the maximum among all valid couplings.

procedure is identical to the one used in Algorithm 2. This follows from the fact that $q(\tilde{Y})/p(\tilde{Y}) = \mathcal{N}(Z + \Delta; 0, \mathrm{Id})/\mathcal{N}(Z; 0, \mathrm{Id})$ with $\Delta = (m^p - m^q)/\sigma$ for $\tilde{Y} = m^p + \sigma Z$. At acceptance, we also have $Y = \tilde{Y}$ as in Algorithm 2. However, Algorithm 4 differs fundamentally from Algorithm 2, as upon rejection of the draft state, the new state is computed *deterministically* as a function of the rejected state, instead of sampling from (6); see Figure 2 for an illustration. This requires only one evaluation of the target $q(y_k|y_{k-1})$ to obtain the state $Y_k$. Therefore, we use Algorithm 4 for REJECTION in our implementation of speculative sampling. A detailed full implementation is provided in Algorithm 6. The following proposition, which parallels Proposition 2.2, establishes the correctness of the method and follows Section 2.3.2 from Bou-Rabee et al. (2020).

**Proposition 3.1 (Reflection Coupling):** *Let* $p(x) = \mathcal{N}(x; m^p, \sigma^2\mathrm{Id})$, $q(x) = \mathcal{N}(x; m^q, \sigma^2\mathrm{Id})$ *and* $\tilde{Y} \sim p$. *Algorithm 4 outputs* $Y \sim q$. *Additionally, it maximizes the probability that* $Y = \tilde{Y}$ *and*

$$\mathbb{P}(Y \neq \tilde{Y}) = ||p - q||_{\mathrm{TV}} = 2\Phi(\sigma^{-1}||m^p - m^q||/2) - 1,$$

*where* $||p - q||_{\mathrm{TV}} = \frac{1}{2}\int |p(x) - q(x)|\mathrm{d}x$ *and* $\Phi$ *is the c.d.f. of the standard normal random variable.*

This result shows that the efficiency of speculative sampling at time $k$ – i.e., the probability of accepting a draft state – is a decreasing function of $||m_{k-1}^p(\tilde{Y}_{n:k-1}) - m_{k-1}^q(\tilde{Y}_{k-1})||/\sigma_{k-1}$. This means that, as expected, a draft model must reasonably approximate the target for good performance.

---

**Algorithm 4** REJECTION $(p, q, \tilde{Y})$ for two Gaussians with same covariance

---

**Require:** Gaussians $p(x) = \mathcal{N}(x; m^p, \sigma^2\text{Id}), q(x) = \mathcal{N}(x; m^q, \sigma^2\text{Id})$ and $\tilde{Y} \sim p$.
  Set $\Delta = (m^p - m^q)/\sigma$ and $e = \Delta/||\Delta||$.
  Let $Z = (\tilde{Y} - m^p)/\sigma$.
  Sample $U \sim \text{Unif}[0, 1]$.
  $\texttt{bool} = \mathbb{I}\Big[U \le \min\Big(1, \frac{\mathcal{N}(Z+\Delta;0,\text{Id})}{\mathcal{N}(Z;0,\text{Id})}\Big)\Big]$.
  **if** $\texttt{bool}$ **then**
      Set $Y = \tilde{Y}$.
  **else**
      Set $Y = m^q + \sigma(\text{Id} - 2ee^\top)Z$.
  **end if**
  **return** $(Y, \texttt{bool})$ where $Y \sim q$.

---

# 4. Theoretical analysis

We provide an analysis of the proposed methodology. We derive an approximation of the complexity of speculative sampling in Section 4.1, and a lower-bound on the acceptance ratio when using an independent draft model, in Section 4.2.

## 4.1. Complexity analysis

We analyze here the computational benefits of using speculative sampling for DDMs under a simplified computational model. We assume an independent draft model $p$ given by (4). The cost of evaluating $b_t^q$ and $b_t^p$ are $C_q$ and $C_p$ respectively with $C_q > C_p$. Using a window size $L$, each step of speculative sampling increases the iteration index $n$ by a random variable $\hat{L} \in \{1, \ldots, L\}$. The cost of running the target model for $K$ iterations is $C_{\text{original}} = KC_q$, while speculative sampling approximately requires $C_{\text{spec}} = (K/\hat{L})(LC_p + C_q)$. This simplified computational model leads directly to the following proposition.

**Proposition 4.1 (Average cost ratio):** *We have that*

$$\mathbb{E}[C_{\text{original}}/C_{\text{spec}}] = \frac{\mathbb{E}[\hat{L}]}{1 + LC_p/C_q}. \qquad (8)$$

Note that the average cost ratio (8) is independent of $K$. Speculative sampling is beneficial if this ratio exceeds one,

which occurs if and only if

$$\mathbb{E}[\hat{L}]/L \ge C_p/C_q + 1/L. \qquad (9)$$

Under the simplifying assumption that the acceptance probability of any draft state is lower bounded by $\alpha$, independent across the state sequence, one has

$$\mathbb{E}[\hat{L}] \ge \sum_{\ell=0}^{L-1}(\ell+1)\alpha^\ell(1-\alpha) + L\alpha^L = 1 - \alpha^L + L\alpha^{L+1}.$$

This highlights the competing factors in speculative sampling. To satisfy (9), we aim for $\mathbb{E}[\hat{L}]/L$ to be as close to one as possible, indicating a high acceptance ratio and thus a draft model that closely approximates the target model. This typically implies that $C_p \approx C_q$. Conversely, while (9) is made easier to satisfy by minimizing $C_p/C_q$, this will in practice cause the acceptance ratio to deteriorate, consequently decreasing $\mathbb{E}[\hat{L}]/L$.

## 4.2. Lower bound on acceptance ratio

We shed light here on how the acceptance ratio depends on the problem parameters and an independent draft model. Let $a_n = \mathcal{N}(Z+\Delta_n; 0, \text{Id})/\mathcal{N}(Z; 0, \text{Id})$ for $Z \sim \mathcal{N}(0, \text{Id})$ where

$$||\Delta_n||^2 = \frac{1}{4}\gamma(\varepsilon + \tfrac{1}{\varepsilon})^2 g_{1-t_n}^2 ||s_{1-t_n}^p(\tilde{Y}_n) - s_{1-t_n}^q(\tilde{Y}_n)||^2,$$

for $b_t^q(x) = -f_{1-t}x + \frac{1+\varepsilon^2}{2}g_{1-t}^2 s_{1-t}^q(x)$, where both $s_t^q$ and $s_t^p$ approximate the true score $s_t$ at differing computational cost. The draft state at time $n+1$ is accepted with probability $\min(1, a_n)$. We have the following lower bound.

**Lemma 4.2 (Control of acceptance ratio):** *We have*

$$\mathbb{E}[a_n] \ge \exp\left[-\tfrac{1}{2}\mathbb{E}[||\Delta_n||^2]\right].$$

Similar results can be established for the frozen draft model.

We next assume that the target model has access to the exact score for distribution $q_{\text{data}}$, and that the draft model score corresponds to an exact score for some distribution $p_{\text{data}}$ (we can think of this as a means of characterizing the inexactness of the draft model). We obtain the following result.

**Theorem 4.3 (Control of acceptance ratio (II)):** *Under assumptions on $p_{data}$ and $q_{data}$ detailed in the supplementary material, we have*

$$\mathbb{E}[a_n] \ge \exp\Big[-C\frac{\gamma g_{s_n}^2}{8}(\varepsilon + \tfrac{1}{\varepsilon})^2$$
$$\times \min\Big((\tfrac{1}{\sigma_{s_n}} - \sigma_{s_n})^2 + \alpha_{s_n}^2, \tfrac{1}{\alpha_{s_n}^2}\text{D}(p_{data}, q_{data})\Big)\Big],$$

*where $s_n = 1 - t_n$, $C$ a constant and $\text{D}(p_{data}, q_{data})$ is some divergence between $p_{data}$ and $q_{data}$ explicit in the proof.*

There are different factors influencing the lower bound of Theorem 4.3:

- As $\gamma \to 0$, we have $\mathbb{E}[a_n] \geq 1$, implying that a smaller discretization step size leads to higher acceptance rates of draft states. However, a smaller step size also necessitates a larger total number of steps to reach the target.

- If $\mathrm{D}(p_{\mathrm{data}}, q_{\mathrm{data}}) \to 0$ then $\mathbb{E}[a_n] \geq 1$. This means that if the draft and target models approximate the same data distribution then we obtain a higher acceptance rate.

- If $g_t^2((\frac{1}{\sigma_t} - \sigma_t)^2 + \alpha_t^2) \to 0$ as $t \to 1$ then $\mathbb{E}[a_n] \geq 1$ for $n$ close to 0. This is the case for classical schedules $(f_t, g_t)$ used in practice. Hence at the beginning of the denoising process, the acceptance rate is high.

- The dependency with respect to $\varepsilon$ is such that both low and high values worsen the lower bound. There exists an optimal parameter $\varepsilon$ ($\varepsilon = 1.0$ in this bound). In practice, we sweep over $\varepsilon > 0$.

## 5. Speculative Sampling for Langevin Diffusions

Consider a scenario where we are interested in sampling from an unnormalized density $\pi(x)$ on $\mathbb{R}^d$, i.e.

$$\pi(x) = \frac{\exp(-E(x))}{Z}, \qquad Z = \int \exp(-E(x)) \mathrm{d}x,$$

where the energy function $E(x)$ can be evaluated pointwise, but each evaluation is computationally expensive, and $Z$ is intractable. To sample from such distributions, we typically use Markov chain Monte Carlo (MCMC) techniques which are iterative algorithms requiring evaluating the energy function at each iteration. We show here how we can accelerate MCMC methods when we have access to a computationally cheap proxy energy function $\hat{E}(x) \approx E(x)$ defining $\hat{\pi}(x) \propto \exp(-\hat{E}(x))$ using speculative sampling. Access to such proxies is common in many domains of computational science and engineering; see e.g. (Christen & Fox, 2005; Cui et al., 2011; Sherlock et al., 2017; Peherstorfer et al., 2018).

A standard MCMC technique to sample from $\pi$ is the Langevin diffusion defined by

$$\mathrm{d}\mathbf{X}_t = -\nabla E(\mathbf{X}_t)\mathrm{d}t + \sqrt{2}\mathrm{d}\mathbf{B}_t,$$

where $(\mathbf{B}_t)_{t \geq 0}$ is a Brownian motion. The limiting distribution of this diffusion is $\pi$. In practice, the so-called unadjusted Langevin algorithm (ULA) (Durmus & Moulines, 2017; Vempala & Wibisono, 2019) is often implemented

$$X_{k+1} = X_k - \gamma \nabla E(X_k) + \sqrt{2\gamma}W_k, \qquad (10)$$

for a stepsize $\gamma > 0$ and $W_k \overset{\text{i.i.d.}}{\sim} \mathcal{N}(0, \mathrm{Id})$. Due to this time discretization, ULA only samples from an approximation

of $\pi$, but explicit bounds on the bias incurred are available (Durmus & Moulines, 2017).

The speculative sampling procedure for DDMs presented in Algorithm 3 and relying on reflection maximal coupling (Algorithm 4) can be easily modified to accelerate the simulation of (10). In this scenario, (10) plays the role of the target model while

$$X_{k+1} = X_k - \gamma \nabla \hat{E}(X_k) + \sqrt{2\gamma}W_k, \qquad (11)$$

is the draft model. As a cheap proxy, we can also use the frozen draft model strategy, that is set $\nabla \hat{E}(x_{n+k}) = \nabla E(x_n)$ for $k = 1, ..., n_L$.

Speculative sampling can be interpreted here as a prefetching technique (Brockwell, 2006; Angelino et al., 2014); see Appendix K for a detailed description of the algorithm. It is an alternative to recent methods proposed to accelerate Langevin diffusions relying also on on parallel evaluations of $\nabla E$ (Shen & Lee, 2019; Anari et al., 2024; Yu & Dalalyan, 2024; Zhou & Sugiyama, 2024).

## 6. Related works

**Speculative sampling.** Introduced in the context of LLMs by Leviathan et al. (2023); Chen et al. (2023), speculative sampling relies on a draft model based on a cheap LLM. An early drafting methodology proposing multiple tokens at once was put forth by Stern et al. (2018), while drafting with independent models was explored in (Chen et al., 2023; Leviathan et al., 2023; Spector & Re, 2023; Sun et al., 2023; Christopher et al., 2024). Efficient drafting using the target model with additional feedforward neural network (FFN) heads was considered in (Stern et al., 2018; Sun et al., 2021; Xia et al., 2023; Cai et al., 2024). Finally, it has been proposed very recently by Christopher et al. (2024) to use a discrete DDM (Austin et al., 2021; Campbell et al., 2022) as draft model for an autoregressive target model. For a comprehensive review of speculative sampling techniques for LLMs, we refer to Xia et al. (2024). Wang et al. (2024) has adapted speculative sampling to continuous state space but sample from the adjusted distribution (6) using rejection sampling, which is computationally inefficient in our context.

**Acceleration of diffusion models.** One line of work distills a teacher DDM into a student DDM for faster sampling; see (Luhman & Luhman, 2021; Salimans & Ho, 2022; Berthelot et al., 2023; Liu et al., 2023; Meng et al., 2023; Sauer et al., 2024; Song et al., 2023; Katzir et al., 2024; Kim et al., 2024; Xu et al., 2024; Yin et al., 2024). For a review of distillation methods, we refer to Luo (2023); Dieleman (2024). Another line of work pursues accelerating sampling through improved integrators (Dockhorn et al., 2022; Liu et al., 2022; Lu et al., 2022; Xiao et al., 2022; Zhang & Chen, 2023). Additionally, parallel sampling of DDMs has been

explored in (Shih et al., 2023; Chen et al., 2024; Li et al., 2024a; Ma et al., 2024; Tang et al., 2024). Our approach complements these approaches and can be combined with parallel sampling and/or better integrators. In Appendix J, we support this claim by combining our method with the parallel sampling integrator from (Shih et al., 2023). Specifically, we show that our method can benefit in terms of both NFE and FID from using a single parallel call. In addition, our method can be seamlessly used in combination with timestep distillation methods, such as those in (Sabour et al., 2024; Tong et al., 2024).

# 7. Experiments

In all of our experiments, we track two different types of metrics. First, we assess the quality of the output distribution obtained with the speculative sampling strategy (Wasserstein-2 in the low dimensional case, FID (Heusel et al., 2017) and IS (Salimans et al., 2016) in the image experiments and reward (Chi et al., 2023) in the robotics setting). We also report the Number of Function Evaluations of the target model; a function evaluation is defined as a call to the target model with a batch of data, irrespective of the batch size. Experiments to accelerate Langevin diffusions can be found in Appendix K.

**Low dimensional experiments.** We first investigate Algorithm 3 in a low dimensional setting in order to better understand the effect of key hyperparameters of the algorithm. We consider a mixture of Gaussians target distribution with dimension varying between $[2, 4, 8, 16, 32]$ and 16 components. All diffusion models are trained with a velocity objective, see Appendix I.1. We consider two drafting strategies: the INDEPENDENT strategy and the FROZEN strategy as described in Section 3.1. We also refer to Appendix I.1 for the architectural details. In Figure 3, we display the effects on the performance of the algorithm of the stochasticity $\varepsilon$ in the sampler and the window size $L$.

Figure 3 illustrate that FROZEN drafting is more efficient than INDEPENDENT drafting as it provides a better reduction of the NFE of the target models. This is in accordance with findings in speculative decoding/sampling for LLMs, see e.g. (Cai et al., 2024). Regarding the amount of stochasticity $\varepsilon$, there appears to be an optimal value $\varepsilon$ for which the speculative sampling gains are optimal in agreement with theoretical insights derived in Theorem 4.3. Finally, increasing the window size $L$ improves the performance of speculative sampling.

**Image space experiments.** Next, we demonstrate speculative sampling in higher dimensional settings on two datasets: CIFAR10 ($32 \times 32 \times 3$) and LSUN ($64 \times 64 \times 3$). In all settings, the backbone architecture is a U-Net. We

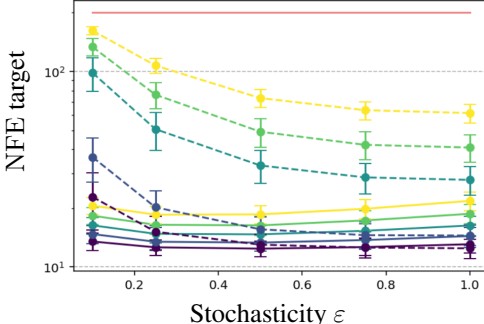

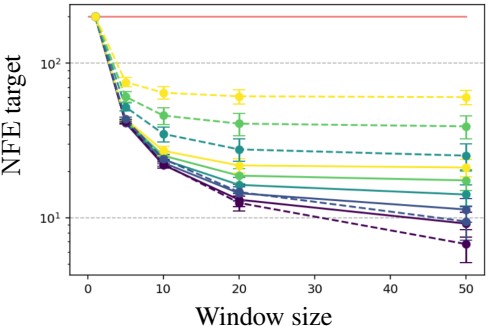

*Figure 3.* In each figure, the $y$ axis corresponds to the number of evaluations of the target model. Without speculative sampling, we evaluate the target model with 200 steps and show the improvements obtained using our approach. Each dotted line corresponds to INDEPENDENT drafting, each solid line to FROZEN drafting. The color gradient purple to yellow corresponds to different dimensions of the target distribution $[2, 4, 8, 16, 32]$.

refer to Appendix I.1 for architectural and training details. For all experiments we report FID score computed on 50k training samples. Our results are reported in Table 1. We investigate the effect of temperature on our models. More precisely, we introduce an hyperparameter $\tau > 0$ such that $\mathcal{N}(Z + \Delta; 0, \mathrm{Id})/\mathcal{N}(Z; 0, \mathrm{Id})$ is replaced by $\mathcal{N}(Z + \Delta; 0, \tau\mathrm{Id})/\mathcal{N}(Z; 0, \tau\mathrm{Id})$, see Appendix F.1 for more details. Note that upon choosing $\tau > 1$ we do not sample exactly from $q$ but improve the acceptance rate. We sweep over the values of $\varepsilon$ and $\tau$ in Table 1 on the CIFAR10 dataset. The main conclusion is that our proposed speculative sampling algorithm provides a significant speed-up (x2 to x3) while maintaining the quality of the target model. For example, on CIFAR10, we reach a FID score of 2.34 with only 35 calls to the target model, while the classical sampling procedure requires 100 calls to the target model to reach a FID score of 2.45, which represents a reduction of 65% of the number of calls to the target model. Running the target model for only 30 steps on the other hand reduces the image quality, as the FID score worsens to 4.32. It is worth noting that increasing the temperature marginally im-

| Configuration | Draft (100 steps) | | Target (100 steps) | | Target (30 steps) | | Speculative | | |
|---|---|---|---|---|---|---|---|---|---|
| | FID ↓ | IS ↑ | FID ↓ | IS ↑ | FID ↓ | IS ↑ | FID ↓ | IS ↑ | NFE ↓ |
| $\varepsilon = 0.01, \tau = 0.5$ | **17.05** | **8.67** | 2.86 | 10.10 | **4.32** | 10.83 | 2.84 | 10.11 | 65.36 |
| $\varepsilon = 0.01, \tau = 1.0$ | **17.05** | **8.67** | 2.86 | 10.10 | **4.32** | 10.83 | 2.84 | 10.11 | 61.64 |
| $\varepsilon = 0.01, \tau = 2.0$ | **17.05** | **8.67** | 2.86 | 10.10 | **4.32** | 10.83 | 2.83 | 10.12 | 57.47 |
| $\varepsilon = 0.25, \tau = 0.5$ | 81.58 | 7.60 | **2.45** | 10.31 | 7.68 | 11.32 | 2.42 | 10.24 | 42.44 |
| $\varepsilon = 0.25, \tau = 1.0$ | 81.58 | 7.60 | **2.45** | 10.31 | 7.68 | 11.32 | 2.35 | 10.25 | 39.31 |
| $\varepsilon = 0.25, \tau = 2.0$ | 81.58 | 7.60 | **2.45** | 10.31 | 7.68 | 11.32 | **2.34** | 10.32 | **35.40** |
| $\varepsilon = 0.5, \tau = 0.5$ | 115.57 | 5.25 | 2.81 | 10.72 | 10.28 | **11.55** | 2.71 | 10.59 | 43.08 |
| $\varepsilon = 0.5, \tau = 1.0$ | 115.57 | 5.25 | 2.81 | 10.72 | 10.28 | **11.55** | 2.71 | 10.57 | 40.37 |
| $\varepsilon = 0.5, \tau = 2.0$ | 115.57 | 5.25 | 2.81 | 10.72 | 10.28 | **11.55** | 2.74 | 10.52 | 36.72 |
| $\varepsilon = 1.0, \tau = 0.5$ | 188.29 | 2.64 | 7.09 | **11.22** | 28.93 | 11.48 | 7.12 | 11.14 | 46.54 |
| $\varepsilon = 1.0, \tau = 1.0$ | 188.29 | 2.64 | 7.09 | **11.22** | 28.93 | 11.48 | 7.10 | **11.18** | 44.81 |
| $\varepsilon = 1.0, \tau = 2.0$ | 188.29 | 2.64 | 7.09 | **11.22** | 28.93 | 11.48 | 7.11 | 11.14 | 42.11 |

*Table 1.* CIFAR-10 evaluation. For each column, we report the best result in **bold**.

| Configuration | Target | | Speculative | |
|---|---|---|---|---|
| | Reward ↑ | NFE ↓ | Reward ↑ | NFE ↓ |
| $L = 20, K = 100$ | $0.889 \pm 0.008$ | 100 | $0.898 \pm 0.008$ | $27.245 \pm 0.002$ |
| $L = 20, K = 80$ | $0.882 \pm 0.008$ | 80 | $0.899 \pm 0.008$ | $23.890 \pm 0.003$ |
| $L = 20, K = 40$ | $0.898 \pm 0.008$ | 40 | $0.875 \pm 0.008$ | $15.544 \pm 0.005$ |
| $L = 20, K = 20$ | $0.887 \pm 0.008$ | 20 | $0.901 \pm 0.008$ | $9.430 \pm 0.004$ |
| $L = 10, K = 10$ | $0.901 \pm 0.008$ | 10 | $0.903 \pm 0.007$ | $5.053 \pm 0.001$ |
| $L = 5, K = 5$ | $0.876 \pm 0.008$ | 5 | $0.870 \pm 0.009$ | $3.000 \pm 0.000$ |

*Table 2.* PushT evaluation.

prove FID and IS score for some values of $\varepsilon$. We observe similar improvements (around halving the NFE) in the case of LSUN, see Appendix J.

**PushT dataset.** Finally, we conclude our experimental study by showing that speculative sampling also yields improvements for a robotics task, where the policy is generated using a diffusion model following (Chi et al., 2023). In our setting, we focus on the PushT dataset. The state space is of dimension $(16, 2)$, where 16 corresponds to the prediction horizon and 2 is the dimension of the action. We refer to Appendix I.1 for more details. The metric we report is the reward $r \in [0, 1]$ where $r = 1.0$ means that the policy achieves perfect coverage over an episode. For robustness we run 1000 episodes to compute the mean of the maximum rewards. For each episode we run the policy for 300 steps or stop if we reach the maximum reward. We follow the setting of (Chi et al., 2023), see also Appendix I.1. For our speculative sampler we fix $\tau = 1$ and do not perform any parallel call. We only consider the FROZEN drafting strategy. We report our results in Table 2. We consistently observe that the speculative sampling strategy reduces the number of call to the target model while preserving the quality of the model. For instance, with only 5 calls to the target model, our speculative sampler achieves a reward of $0.903 \pm 0.007$ while running the target model with only 5 steps yields a reward of $0.876 \pm 0.008$.

## 8. Discussion

We have developed here a novel speculative sampling procedure to accelerate diffusion models. This was achieved by exploiting the connections between speculative sampling and maximal coupling, specifically through the use of reflection maximal coupling. We have demonstrated that significant speed-up can be achieved while sampling exactly from the target distribution.

This approach also has limitations. It is not directly applicable to deterministic samplers, although noise can be added in a principled way to such samplers to obtain a valid stochastic sampler, to which speculative sampling can then be applied (see e.g. Section 3.1). Moreover, similarly to Picard iteration techniques (Shih et al., 2023; Chen et al., 2024), it increases memory overhead due to the parallel calls to the sampler during the verification procedure.

In particular, while LLMs are memory-bound and therefore heavily benefit from speculative decoding/sampling techniques that increase arithmetic intensity and reduce latency, this is not necessarily the case for diffusion models, which already benefit from parallelism. The applicability of parallel techniques for serving diffusion models, therefore, remains an active area of investigation.

## Impact Statement

This paper presents work whose goal is to advance the field of Machine Learning. There are many potential societal consequences of our work, none which we feel must be specifically highlighted here.

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

## Organization of the Supplementary Material

This supplementary material is organized as follows. Proofs of the main results are gathered in Appendix A. Potential alternative drafting strategies are discussed in Appendix B. A detailed implementation of speculative sampling for diffusion models is presented in Appendix C. In Appendix D, we present various results on the first time to rejection and how this could be efficiently approximated numerically. An extension of the maximal coupling strategy for Gaussians with different variances is proposed in Appendix E. In Appendix F, we investigate alternative acceptance criteria relying on either the introduction of a temperature parameter or the use of the "typical acceptance criterion" of (Cai et al., 2024) introduced for LLMs. Appendix G presents an extension of speculative sampling to incorporate some spatial transform. In Appendix H, we establish a lower bound on the expectation of the log-acceptance ratio. Experimental details are gathered in Appendix I. Finally, Appendix K details how the speculative sampling procedure proposed in this work can be used to accelerate simulation of Langevin diffusions to sample from unnormalized target distributions and presents simulations in this context.

## A. Proofs of the Main Results

### A.1. Proof of Proposition 2.1

The joint distribution of $(\tilde{X}, X)$ generated by Algorithm 2 is

$$f(\tilde{x}, x) = p(\tilde{x})(\alpha(\tilde{x})\delta_{\tilde{x}}(x) + (1 - \alpha(\tilde{x}))r(x)),$$

with $\delta_{\tilde{x}}(x)$ the Kronecker-delta symbol and $\alpha(\tilde{x}) = \min(1, q(\tilde{x})/p(\tilde{x}))$. That is we first sample $\tilde{X} \sim p$ then set $X = \tilde{X}$ with probability $\alpha(\tilde{X})$ and sample $X \sim r$ otherwise. It follows that the marginal distribution of $X$ is given by

$$f(x) = \sum_{\tilde{x} \in \mathcal{X}} f(\tilde{x}, x) = \alpha(x)p(x) + \Big(1 - \sum_{\tilde{x} \in \mathcal{X}} \alpha(\tilde{x})p(\tilde{x})\Big)r(x). \tag{12}$$

We have

$$r(x) \propto \max(0, q(x) - p(x))$$
$$= q(x) - \min(p(x), q(x))$$
$$= q(x) - \alpha(x)p(x).$$

Therefore, we have that

$$r(x) = \frac{q(x) - \alpha(x)p(x)}{1 - \sum_{\tilde{x} \in \mathcal{X}} \alpha(\tilde{x})p(\tilde{x})}.$$

Hence, by substituting the expression of $r(x)$ in (12), we obtain $f(x) = q(x)$, that is $X \sim q$. Now by construction, we have that

$$\mathbb{P}(X \neq \tilde{X}) = 1 - \sum_{x \in \mathcal{X}} p(x)\alpha(x)$$
$$= 1 - \sum_{x \in \mathcal{X}} \min(p(x), q(x))$$
$$= ||p - q||_{\text{TV}},$$

as $\min(a, b) = \frac{1}{2}(a + b - |a - b|)$ for any $a, b$. However, Lindvall's inequality (Lindvall, 1992) (also known as the coupling inequality) shows that any pair of random variables $X, \tilde{X}$ satisfying marginally $\tilde{X} \sim p$ and $X \sim q$ verify

$$||p - q||_{\text{TV}} \leq \mathbb{P}(X \neq \tilde{X}). \tag{13}$$

Algorithm 2 generates a joint distribution for which the inequality (13) becomes an equality; hence it is optimal.

### A.2. The need for adjusted rejection sampling

One could wonder if an algorithm where we sample from $q$ under rejection and not from the modified probability $r(x) \propto \max(0, q(x) - p(x))$ would work. In particular, we could consider Algorithm 5.

---

**Algorithm 5 INCORRECT** `REJECTION` $(p, q, \tilde{X})$

---

**Require:** Proba. distributions $p, q$ and $\tilde{X} \sim p$.

    Sample $U \sim \text{Unif}[0, 1]$.

    `bool` $= \mathbb{I}[U \leq \min(1, q(\tilde{X})/p(\tilde{X}))]$.

    **if** `bool` **then**

        Set $X = \tilde{X}$.

    **else**

        $X \sim q(\cdot)$.

    **end if**

    **return** $(X,$ `bool`$)$ where $X \sim q$.

---

It can easily be shown that Algorithm 5 does not output $Y$ with distribution $q$ as in this case the joint distribution of $X, Y$ is

$$f(\tilde{x}, x) = p(\tilde{x})(\alpha(\tilde{x})\delta_{\tilde{x}}(x) + (1 - \alpha(\tilde{x}))q(x)),$$

so the marginal distribution of $X$ is given by

$$f(x) = \alpha(x)p(x) + \left(1 - \sum_{\tilde{x} \in \mathcal{X}} \alpha(\tilde{x})p(\tilde{x})\right)q(x) \neq q(x).$$

In particular, the following example illustrates the problems with Algorithm 5. Consider $p = \text{Unif}(\{0, 1\})$ and $q = \text{Unif}(\{0, 1, 2, 3\})$. In that case, we have that `bool` $= \text{Ber}(1/2)$. Hence, we accept $\tilde{X}$ half of the time in expectation. If we were sampling from $q = \text{Unif}(\{0, 1, 2, 3\})$ upon rejection then the output distribution $f(x)$ of $X$ would be given by

$$X \sim \frac{3}{4}\text{Unif}(\{0, 1\}) + \frac{1}{4}\text{Unif}(\{2, 3\}).$$

This means that we sample *too much* on the set $\{0, 1\}$. In order to get $X \sim q(\cdot)$ we need to sample more on the set which is *outside* of the support of $p$. This is exactly the purpose of $r(\cdot)$. Indeed, we have that $r(x) = \text{Unif}(\{2, 3\})$. Hence, using Algorithm 2 we get that

$$X \sim \frac{1}{2}\text{Unif}(\{0, 1\}) + \frac{1}{2}\text{Unif}(\{2, 3\}),$$

that is, $X \sim q$ as required.

### A.3. Proof of Proposition 3.1

We have $\tilde{Y} \sim \mathcal{N}(m^p; \sigma^2 \text{Id})$. We check here that the algorithm also returns $Y \sim \mathcal{N}(m^q; \sigma^2 \text{Id})$. To show this, we leverage the fact that we can rewrite $Y = m^q + \sigma\tilde{Z}$ for some random variable $\tilde{Z}$ whose distribution follows

$$f(\tilde{z}) = \int \delta_{z+\Delta}(\tilde{z}) \min\left(1, \frac{\mathcal{N}(z + \Delta; 0, \text{Id})}{\mathcal{N}(z; 0, \text{Id})}\right)\mathcal{N}(z; 0, \text{Id})\mathrm{d}z$$
$$+ \int \delta_{(\text{Id}-2ee^\top)z}(\tilde{z}) \max\left(0, 1 - \frac{\mathcal{N}(z + \Delta; 0, \text{Id})}{\mathcal{N}(z; 0, \text{Id})}\right)\mathcal{N}(z; 0, \text{Id})\mathrm{d}z,$$

where we have used $1 - \min(1, a) = \max(0, 1 - a)$ for $a \geq 0$. Hence to show the validity of the procedure, we need now to show that $\tilde{Z} \sim \mathcal{N}(0, \text{Id})$, i.e., $f(\tilde{z}) = \mathcal{N}(\tilde{z}; 0, \text{Id})$. We have

$$\int \delta_{z+\Delta}(\tilde{z}) \min\left(1, \frac{\mathcal{N}(z + \Delta; 0, \text{Id})}{\mathcal{N}(z; 0, \text{Id})}\right)\mathcal{N}(z; 0, \text{Id})\mathrm{d}z$$
$$= \int \delta_{z+\Delta}(\tilde{z}) \min\left(\mathcal{N}(z; 0, \text{Id}), \mathcal{N}(z + \Delta; 0, \text{Id})\right)\mathrm{d}z$$
$$= \min\left(\mathcal{N}(\tilde{z} - \Delta; 0, \text{Id}), \mathcal{N}(\tilde{z}; 0, \text{Id})\right).$$

In addition, we have that

$$\int \delta_{(\mathrm{Id}-2ee^\top)z}(\tilde{z}) \max\Big(0, 1 - \frac{\mathcal{N}(z+\Delta; 0, \mathrm{Id})}{\mathcal{N}(z; 0, \mathrm{Id})}\Big) \mathcal{N}(z; 0, \mathrm{Id}) \mathrm{d}z$$

$$= \int \delta_{(\mathrm{Id}-2ee^\top)z}(\tilde{z}) \max\Big(0, \mathcal{N}(z; 0, \mathrm{Id}) - \mathcal{N}(z+\Delta; 0, \mathrm{Id})\Big) \mathrm{d}z$$

$$= \max\Big(0, \mathcal{N}((\mathrm{Id}-2ee^\top)\tilde{z}; 0, \mathrm{Id}) - \mathcal{N}((\mathrm{Id}-2ee^\top)\tilde{z} + \Delta; 0, \mathrm{Id})\Big)$$

$$= \max(0, \mathcal{N}(\tilde{z}; 0, \mathrm{Id}) - \mathcal{N}(\tilde{z} - \Delta; 0, \mathrm{Id}))$$

as $\tilde{z} = (\mathrm{Id}-2ee^\top)z$ implies that $z = (\mathrm{Id}-2ee^\top)\tilde{z}$ and $\mathcal{N}((\mathrm{Id}-2ee^\top)\tilde{z}; 0, \mathrm{Id}) = \mathcal{N}(\tilde{z}; 0, \mathrm{Id})$ because $||(\mathrm{Id}-2ee^\top)\tilde{z}|| = ||\tilde{z}||$. Finally we used the fact that $\mathcal{N}((\mathrm{Id}-2ee^\top)\tilde{z} + \Delta; 0, \mathrm{Id}) = \mathcal{N}(\tilde{z} - \Delta; 0, \mathrm{Id})$ as

$$||(\mathrm{Id}-2ee^\top)\tilde{z} + \Delta||^2 = ||\Delta||^2 + ||(\mathrm{Id}-2ee^\top)\tilde{z}||^2 + 2\Delta^\top \tilde{z} - 4\Delta^\top ee^\top \tilde{z}$$

$$= ||\Delta||^2 + ||\tilde{z}||^2 - 2\Delta^\top \tilde{z}$$

$$= ||\tilde{z} - \Delta||^2,$$

as $ee^\top = \Delta\Delta^\top / ||\Delta||^2$. Combining these results, we obtain that

$$f(\tilde{z}) = \min(\mathcal{N}(\tilde{z} - \Delta; 0, \mathrm{Id}), \mathcal{N}(\tilde{z}; 0, \mathrm{Id})) + \max(0, \mathcal{N}(\tilde{z}; 0, \mathrm{Id}) - \mathcal{N}(\tilde{z} - \Delta; 0, \mathrm{Id}))$$

$$= \mathcal{N}(\tilde{z}; 0, \mathrm{Id}).$$

We thus have proved that $\tilde{Z} \sim \mathcal{N}(0, \mathrm{Id})$, so $Y \sim \mathcal{N}(m^q, \sigma^2 \mathrm{Id})$. To prove now that this coupling is a maximal coupling, we compute $\mathbb{P}(Y \neq \tilde{Y})$. Recall that $Y = \tilde{Y}$ if $U \le \min(1, \mathcal{N}(z+\Delta; 0, \mathrm{Id})/\mathcal{N}(z; 0, \mathrm{Id}))$ so

$$\mathbb{P}(Y \neq \tilde{Y}) = 1 - \int \min\Big(\mathcal{N}(z; 0, \mathrm{Id}), \mathcal{N}(\Delta + z; 0, \mathrm{Id})\Big) \mathrm{d}z.$$

It is straightforward to check that this is indeed equal to

$$||p - q||_{\mathrm{TV}} = ||\mathcal{N}(\mu_1, \sigma^2 \mathrm{Id}) - \mathcal{N}(\mu_2, \sigma^2 \mathrm{Id})||_{\mathrm{TV}} = 2\Phi(||\Delta||/2) - 1,$$

where $\Phi$ is the cumulative distribution function of the standard normal random variable. Hence, it follows from Lindvall's inequality (Lindvall, 1992) that Algorithm 4 outputs a maximal coupling.

Similarly to Appendix A.2, it can be easily shown that we cannot sample simply independently from $q$ in the case we reject as it would output unconditionally a random variable $Y$ whose distribution differs from $p$.

### A.4. Optimality of reflection maximal coupling

In Appendix A.3 we have shown that the reflection coupling is a maximal coupling. In what follows, we denote $\mathcal{C}(m^p, m^q)$ the set of coupling, i.e., distributions on $\mathbb{R}^d \times \mathbb{R}^d$ with marginals $\mathcal{N}(m^p, \sigma^2 \mathrm{Id})$ and $\mathcal{N}(m^q, \sigma^2 \mathrm{Id})$. We also denote by $\Pi_{\mathrm{reflection}} \in \mathcal{C}(m^p, m^q)$ the reflection coupling. Proposition 3.1 shows that

$$\Pi_{\mathrm{reflection}} \in \mathrm{argmin}_{\Pi \in \mathcal{C}(m^p, m^q)} \mathbb{E}_{(\tilde{Y}, Y) \sim \Pi}[\mathbf{1}_{\tilde{Y} \neq Y}].$$

In fact, Hsu & Sturm (2013, Theorem 4.2) show that

$$\Pi_{\mathrm{reflection}} \in \mathrm{argmin}_{\Pi \in \mathcal{C}(m^p, m^q)} \mathbb{E}_{(\tilde{Y}, Y) \sim \Pi}[\phi(||\tilde{Y} - Y||)],$$

for every non-negative, strictly increasing and strictly concave function $\phi$ with $\phi(0) = 0$. Hence, the reflection coupling also naturally appears if one considers other cost functions than $(\tilde{y}, y) \mapsto \mathbf{1}_{\tilde{y} \neq y}$.

## B. Alternative drafting strategies for diffusion models

**Medusa-like correction.** To improve the frozen target as draft model (see (16)), we can introduce a correction term to the frozen model. Our correction is inspired by the Medusa architecture (Cai et al., 2024). More precisely, we consider a smaller *correction model* $c_{s,t}^\theta$ trained with the following loss

$$\mathcal{L}(\theta) = \int_0^1 \int_0^1 ||b_t^q(x_t) - b_s^q(x_s) - c_{s,t}^\theta(x_s, x_t)||^2 p_{s,t}(x_s, x_t) \mathrm{d}x_s \mathrm{d}x_t. \tag{14}$$

Here $p_{s,t}$ can be any distribution with support on $\mathbb{R}^d \times \mathbb{R}^d$ as the minimizer for $c_{s,t}(x_s, x_t)$ is then always $b_t^q(x_t) - b_t^q(x_s)$. However, in practice, one may choose $p_{s,t}(x_s, x_t)$ defined by the following procedure

$$\mathbf{X}_t = \alpha_t \mathbf{X}_0 + \sigma_t \mathbf{Z}_t, \qquad \mathbf{X}_s = \frac{\alpha_t}{\alpha_s} \mathbf{X}_s + (\sigma_t^2 - (\frac{\sigma_s \alpha_t}{\alpha_s})^2)^{1/2} \mathbf{Z}_s,$$

where $\mathbf{Z}_t$ and $\mathbf{Z}_s$ are independent Gaussian random variables with zero mean and identity covariance matrix where this joint distribution of $\mathbf{X}_s$ and $\mathbf{X}_t$ is induced by the diffusion

$$d\mathbf{X}_u = f_u \mathbf{X}_u + g_u d\mathbf{B}_u.$$

If the correction model is expressive enough then the minimizer of (14) is given by $c_{s,t}^\theta(x_s, x_t) = b_t^q(x_t) - b_s^q(x_s)$ and in that case the draft model is equal to the target model.

We then have a model $p(y_k|y_{n:k-1}) = \mathcal{N}(y_k; m_{k-1}^p(y_{n:k-1}), \sigma_{k-1}^2 \text{Id})$ with

$$m_k^p(y_{n:k}) = y_k + \gamma\{b_{t_n}^q(y_n) + c_{t_n, t_k}^\theta(y_n, y_k)\}, \qquad \sigma_k = \varepsilon\sqrt{\gamma}g_{1-t_k}.$$

Hence, on a window of size $L$, we only need to evaluate the target model once while the (cheap) correction model is evaluated $L$ times. If $c_{s,t}^\theta = 0$ then we recover the draft model proposed in (16). Note that similarly to the frozen model, we can sample the draft states in parallel.

**Combining draft models.** Assume we have $N_p$ draft models such that $p_\ell(y_k|y_{k-1}) = \mathcal{N}(y_k; m_{k-1}^{p,\ell}(y_{n:k-1}), \sigma_{k-1}^2 \text{Id})$ and let $\alpha_{k-1}^\ell(y_{k-1}) \geq 0$ such that $\sum_{\ell=1}^{N_p} \alpha_k^\ell(y_{k-1}) = 1$. We can define a new draft distribution

$$p_{\text{mix}}^\alpha(y_k|y_{k-1}) = \mathcal{N}(y_k; m_{k-1}^{\text{mix}}(y_{k-1}), \sigma_{k-1}^2 \text{Id}), \quad m_{k-1}^{\text{mix}}(y_{k-1}) = \sum_{\ell=1}^{N_p} \alpha_{k-1}^\ell(y_{k-1}) m_{k-1}^{p,\ell}(y_{k-1}). \tag{15}$$

The distribution in (15) mixes together $N_p$ draft models by considering a convex combination of their means. Since it is a Gaussian, Section 3.3 applies and we get that

$$\mathbb{P}(Y_k \neq \tilde{Y}_k | Y_{k-1}) = 2\Phi(\sigma^{-1}||m^p(\cdot|Y_{k-1}) - m_{k-1}^{\text{mix}}(Y_{k-1})||/2) - 1,$$

The parameters, $\alpha_k(y_{n-1})$ can either be hyperparameters (constants) specified by the practitioner, or can be represented by a mapping $\alpha_k^\ell(y_k; \theta)$ with parameters $\theta$. These parameters $\theta$ can be learned by minimizing the sum of average rejection probabilities $\mathbb{E}_{Y_{0:K} \sim q} \left[ \sum_{k=1}^K \mathbb{P}(Y_k \neq \tilde{Y}_k | Y_{k-1}) \right]$.

**Parallel sampling and speculative correction.** We now show here how one can combine Picard iterations from ParaDiGMS (Shih et al., 2023) and speculative sampling by using the output of ParaDiGMS as a draft model. We start by recalling the Picard iterations from Shih et al. (2023). Consider a draft sequence initialized with some deterministic transformation of $\tilde{Y}_n$, i.e. $\tilde{Y}_{n+1:n_L}^0 = (F_{n+1}^0(\tilde{Y}_n), ..., F_{n_L}^0(\tilde{Y}_n))$ where $n_L = \max(K, n+L)$. Then, we define the Picard iterations as

$$\tilde{Y}_k^m = \tilde{Y}_{k-1}^{m-1} + \gamma\bar{b}_{t_{k-1}}^q(\tilde{Y}_{k-1}^{m-1}), \qquad \tilde{Y}_n^m = \tilde{Y}_n,$$

where $k \in \{n+1, ..., n_L\}$ and $m \in \{1, ..., M-1\}$. Here we use Picard iterations for the *deterministic* sampler, that is $\varepsilon = 0$ and $\bar{b}_t^q(x) = -f_{1-t}(x) + \frac{1}{2}g_{1-t}^2 s_{1-t}(x)$, see Equation (1). Hence, for any $k \in \{n+1, ..., n_L\}$ there exists a deterministic function $F_k^m$ such that

$$\tilde{Y}_k^m = F_k^m(\tilde{Y}_n).$$

Lastly, we consider a last Picard iteration

$$\tilde{Y}_k^M = \tilde{Y}_{k-1}^{M-1} + \gamma b_{t_{k-1}}^q(\tilde{Y}_{k-1}^{M-1}) + \varepsilon\sqrt{\gamma}g_{t_{k-1}}Z_k,$$

where $Z_k \sim \mathcal{N}(0, \text{Id})$. Hence, we have

$$\tilde{Y}_k^M = F_{k-1}^{M-1}(\tilde{Y}_n) + \gamma b_{t_{k-1}}^q(F_{k-1}^{M-1}(\tilde{Y}_n)) + \varepsilon\sqrt{\gamma}\sigma_{t_{k-1}}Z_k.$$

We consider the sequence $\tilde{Y}_{n+1:n_L}^M$ as a draft sequence. In that case we still have that $\tilde{Y}_{n+1:n_L}^M \sim p(y_{n+1:n_L}|y_n)$ with

$$p(y_{n+1:n_L}|y_n) = \prod_{k=n+1}^{n_L} p(y_k|y_{n:k-1}) = \prod_{k=n+1}^{n_L} \mathcal{N}(y_k; m_{k-1}^p(y_{n:k-1}), \sigma_{k-1}^2 \mathrm{Id}),$$

as required where

$$m_k^p(y_{n:k-1}) = F_k^{M-1}(y_n) + \gamma b_{t_k}^q(F_k^{M-1}(y_n)), \qquad \sigma_k = \varepsilon\sqrt{\gamma}g_{1-t_k}.$$

**Efficient frozen draft strategy.** We start by recalling the frozen draft strategy described in Section 3. In that case, we consider here a very simple draft model where $p(y_k|y_{n:k-1}) = p(y_k|y_n, y_{k-1})$ with

$$m_k^p(y_{n:k}) = y_k + \gamma b_{t_n}^q(y_n), \quad \sigma_k = \sqrt{\gamma}\varepsilon g_{1-t_k}. \tag{16}$$

This draft model is similar to the target model, except that we replace $b_{t_k}^q(y_k)$ by $b_{t_n}^q(y_n)$. Importantly, on a window of size $L$, we only need to query the target model *once* in order to draw a draft sequence. In practice, a more efficient modification of this frozen draft strategy can be obtained if we replace $b_{t_n}^q(y_n)$ with $b_{t_n}^q(\tilde{y}_n)$. Note that if, in the previous window, all the samples have been accepted then $b_{t_n}^q(y_n)$ coincides with $b_{t_n}^q(\tilde{y}_n)$. Otherwise they do not. The main advantage of this procedure is that we can leverage the quantities computed on the previous window during the iterated speculative sampling procedure. Indeed, $b_{t_n}^q(\tilde{y}_n)$ is *always* computed when doing speculative sampling on the previous window, in order to perform the verification stage, see Algorithm 3. This drastically reduces the cost of the draft model since, we do not need to call *any* model to compute the proposals, since $b_{t_n}^q(\tilde{y}_n)$ has already been computed at the previous verification stage. The only caveat to this method is that it requires to be initialized, i.e., we need to compute $b_{t_n}^q(\tilde{y}_0)$. In that case, we simply compute $b_{t_n}^q(y_0)$ (and therefore the cost of running the whole draft model is one function evaluation of the target model). Finally, another alternative strategy is to use $b_{t_{n-1}}^q(y_{n-1})$ in place of $b_{t_n}^q(\tilde{y}_n)$ when $\tilde{y}_n \neq y_n$.

## C. Detailed implementation of speculative sampling for diffusion models

We present in Algorithm 6 a detailed implementation of speculative sampling for diffusion models, an algorithm combining Algorithm 3 and Algorithm 4.

## D. Distribution of time to rejection

Consider the following process $(\tilde{Y}_k, Y_k)_{k \geq 0}$ following the following distribution

$$\Gamma(\tilde{y}_{0:n}, y_{0:n}) = p(\tilde{y}_0)\delta_{\tilde{y}_0}(y_0) \prod_{k=1}^{n} p(\tilde{y}_k|y_{k-1})\Big(\alpha_k(y_{k-1}, \tilde{y}_k)\delta_{\tilde{y}_k}(y_k) + (1 - \alpha_k(y_{k-1}, \tilde{y}_k))r(y_k|y_{k-1}, \tilde{y}_k)\Big).$$

where

$$\alpha_k(y_{k-1}, \tilde{y}_k) = \min\Big(1, \frac{q(\tilde{y}_k|y_{k-1})}{p(\tilde{y}_k|y_{k-1})}\Big)$$

and

$$r(y_k|y_{k-1}, \tilde{y}_k) = \delta_{f(y_{k-1}, \tilde{y}_k)}(y_k)$$

corresponds to reflection maximal coupling for an appropriate function $f$. This distribution describes the speculative sampling algorithm for diffusions (not describing the drafting process over an horizon of $L$ but this is irrelevant). By construction, we have $\int \Gamma(\tilde{y}_{0:n}, y_{0:n})\mathrm{d}\tilde{y}_{0:n} = q(y_{0:n})$. Starting from $\tilde{Y}_0 = Y_0$, we can look at the first time $\tau$, $\tau \geq 1$, the draft state is rejected. This is equivalent to look at the first time that $\tilde{Y}_k \neq Y_k$. We have from direct calculations that

$$\mathbb{P}(\tau > k) = \int \cdots \int p(\tilde{y}_{0:k}) \prod_{i=1}^{k} \alpha_i(\tilde{y}_{i-1}, \tilde{y}_i)\mathrm{d}\tilde{y}_{0:k}.$$

Hence $\mathbb{P}(\tau > k)$ is given by a Feynman–Kac formula (Del Moral, 2004) so we could estimate numerically efficiently this quantity by running a particle filter (Del Moral, 2004; Doucet et al., 2001). A similar expression can be obtained for the distribution of $\tau_n$ the $n^{\text{th}}$ time the draft state is rejected starting from the last time one rejected a draft, $p(\tilde{y}_0)$ being replaced by $p(\tilde{y}_{\tau_{n-1}+1}|y_{\tau_n})$.

---

**Algorithm 6** Speculative Sampling for DDM

---

**Require:** Lookahead integer $L$, sequence length $K$, target model $q$ (see eq. (2)) and draft model $p$ (see eq. (3)).

Sample $Y_0 \sim \mathcal{N}(0, \mathrm{Id})$ and set $n = 0$.

**while** $n < K$ **do**

    Set $\tilde{Y}_n \leftarrow Y_n$ and $n_L = \min(n + L, K)$.

    **for** $k = n + 1 : n_L$ **do**

        Sample $\tilde{Y}_k \sim \mathcal{N}(m_{k-1}^p(\tilde{Y}_{n:k-1}), \sigma_{k-1}^2 \mathrm{Id})$.

    **end for**

    In parallel, compute $m_n^q(\tilde{Y}_n)$, $m_{n+1}^q(\tilde{Y}_{n+1}), ..., m_{n_L-1}^q(\tilde{Y}_{n_L-1})$.

    **for** $k = n + 1 : n_L$ **do**

        Set $\Delta_{k-1} = (m_{k-1}^p(\tilde{Y}_{n:k-1}) - m_{k-1}^q(\tilde{Y}_{k-1}))/\sigma_{k-1}$ and $e = \Delta_{k-1}/\|\Delta_{k-1}\|$.

        Let $Z_{k-1} = (\tilde{Y}_k - m_{k-1}^p(\tilde{Y}_{n:k-1}))/\sigma_{k-1}$.

        Sample $U \sim \mathrm{Unif}[0, 1]$.

        `bool` $= \mathbb{I}[U \le \min(1, \mathcal{N}(Z_{k-1} + \Delta_{k-1}; 0, \mathrm{Id})/\mathcal{N}(Z_{k-1}; 0, \mathrm{Id}))]$.

        **if** `bool` **then**

            Set $Y_k = \tilde{Y}_k$.

        **else**

            Set $Y_k = m_{k-1}^q(\tilde{Y}_{k-1}) + \sigma_{k-1}(\mathrm{Id} - 2ee^\top)Z_{k-1}$.

        **end if**

        **return** $(Y_k, $ `bool`$)$.

        **if** `not(bool)` **then**

            Exit For Loop

        **end if**

    **end for**

    Set $n \leftarrow k$.

**end while**

**return** $Y_K$

---

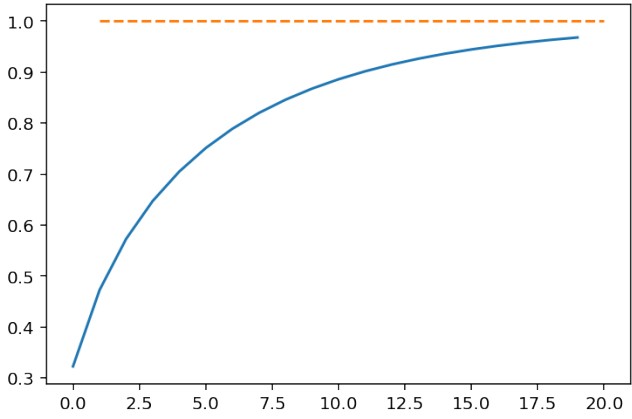

Figure 4. Evolution of the rejection probability ($y$-axis) with the dimension $d$ ($x$-axis) for $\sigma_1 = 0.2$ and $\sigma_2 = 0.1$

In the case where we have

$$p(y_k|y_{k-1}) = \mathcal{N}(y_k; y_{k-1} + \gamma b^p(y_{k-1}), \gamma \mathrm{Id}), \quad q(y_k|y_{k-1}) = \mathcal{N}(y_k; y_{k-1} + \gamma b^q(y_{k-1}), \gamma \mathrm{Id}).$$

Under the simplifying assumption that $||b^p(x) - b^q(x)|| \geq M$, we have

$$\alpha_{\gamma,k} = \mathbb{P}(\tilde{Y}_0 = Y_0) \prod_{i=1}^{k} \mathbb{P}(\tilde{Y}_i = Y_i | \tilde{Y}_{i-1} = Y_{i-1})$$

$$= \prod_{i=1}^{k} 2\Phi\left(-\sqrt{\gamma}||b^p(Y_{i-1}) - b^q(Y_{i-1})||/2\right)$$

$$\leq (2\Phi(-\sqrt{\gamma}M/2))^k.$$

As we have $2\Phi(-x) = 1 - \sqrt{\frac{2}{\pi}}x + o(x^3)$ so

$$\lim_{\gamma \to 0} \alpha_{\gamma,1/\gamma} = 0.$$

## E. Maximal coupling between Gaussian distributions with different covariance matrices

Algorithm 4 is restricted to Gaussian random variables admitting the same covariance matrix. This implies that the draft and the target samplers introduce the same amount of noise at each step. One can wonder if the two samplers could have instead different noise levels. In the following examples, we show that, even if the means are equal, then the probability of rejection is extremely high as the dimension increases.

Indeed, consider two centered $d-$dimensional normals $p(x) = \mathcal{N}(x; 0, \sigma_1^2 \mathrm{Id})$ and $q(x) = \mathcal{N}(x; 0, \sigma_2^2 \mathrm{Id})$. We assume that $\sigma_2 < \sigma_1$. In that case, we have that $p(x) \leq q(x)$ if and only if $\|x\|^2 \leq R^2$ with

$$R^2 = d \log(\sigma_1^2/\sigma_2^2)(1/\sigma_2^2 - 1/\sigma_1^2)^{-1}.$$

Hence, in the ideal case where the acceptance probability $r$ is given by $\|p - q\|_{\mathrm{TV}}$, we get that

$$r = 1 - \mathbb{E}[\mathbf{1}_{\sigma_1^2 Q \leq R^2}] - \mathbb{E}[\mathbf{1}_{\sigma_2^2 Q \geq R^2}] = \mathbb{E}[\mathbf{1}_{Q \leq R^2/\sigma_2^2}] - \mathbb{E}[\mathbf{1}_{Q \leq R^2/\sigma_1^2}].$$

where $Q$ is a $\chi^2$ random variable with $d$ degrees of freedom. Unfortunately this probability is extremely close to 0 as $d$ increases, see Figure 4. Therefore, the probability of coupling is very low in high dimensions.

## F. Alternative verification strategies

In this section, we investigate different verification strategies for Algorithm 4. First, we introduce a temperature parameter in Appendix F.1. Then, in Appendix F.2, we adapt the *typical acceptance* criterion of Stern et al. (2018); Cai et al. (2024) to our setting.

## F.1. Influence of the temperature

Consider Algorithm 7, which is a version of Algorithm 4 including an additional temperature parameter $\tau > 0$.

---

**Algorithm 7** (Temperature) REJECTION $(p, q, \tilde{Y})$ for two Gaussians with same covariance

---

**Require:** Probability densities $p(x) = \mathcal{N}(x; m^p, \sigma^2 \mathrm{Id}), q(x) = \mathcal{N}(x; m^q, \sigma^2 \mathrm{Id}), \tilde{Y} \sim p$, temperature $\tau > 0$.
    Set $\Delta = (m^p - m^q)/\sigma$ and $e = \Delta/||\Delta||$.
    Let $Z = (\tilde{Y} - m^p)/\sigma$.
    Sample $U \sim \mathrm{Unif}[0, 1]$.
    `bool` $= \mathbb{I}[U \leq \min(1, \mathcal{N}(Z + \Delta; 0, \tau\mathrm{Id})/\mathcal{N}(Z; 0, \tau\mathrm{Id}))]$.
    **if** `bool` **then**
        Set $Y = \tilde{Y}$.
    **else**
        Set $Y = m^q + \sigma(\mathrm{Id} - 2ee^\top)Z$.
    **end if**
    **return** $(Y, $ `bool`$)$.

---

Setting $\tau = 1$, we recover Algorithm 4 but for $\tau > 1$ we have a larger probability of accepting the current proposal $\tilde{Y}$. This higher acceptance rate, of course, is not without its drawbacks since we are no longer sampling from the correct distribution. In what follows, we analyze how the distribution is shifted when tuning the temperature parameter $\tau$. To shorten notation, we use in the following proof the notation $\varphi_\tau(z) = \mathcal{N}(z; 0, \tau\mathrm{Id})$ and $\varphi = \varphi_1$ for $\tau = 1$. We recall that $\mathrm{C}_c(\mathbb{R}^d)$ is the set of continuous functions with compact support. For any $f \in \mathrm{C}_c(\mathbb{R}^d)$, using that for any $x \in \mathbb{R}^d$, $\|x\| = \|\hat{x}\|$ for $\hat{x} = (\mathrm{Id} - 2ee^\top)x$, and $x \mapsto (\mathrm{Id} - 2ee^\top)x$ is an involution, we have

$$
\begin{aligned}
\mathbb{E}[f(Y)] &= \int_{\mathbb{R}^d} f(m^p + \sigma z) \min(1, \varphi_\tau(z + \Delta)/\varphi_\tau(z))\varphi(z)\mathrm{d}z \\
&\quad + \int_{\mathbb{R}^d} f(m^q + \sigma\hat{z})(1 - \min(1, \varphi_\tau(z + \Delta)/\varphi_\tau(z)))\varphi(z)\mathrm{d}z, \\
&= \int_{\mathbb{R}^d} f(m^p + \sigma z) \min(1, \varphi_\tau(z + \Delta)/\varphi_\tau(z))\varphi(z)\mathrm{d}z \\
&\quad + \int_{\mathbb{R}^d} f(m^q + \sigma z)(1 - \min(1, \varphi_\tau(\hat{z} + \Delta)/\varphi_\tau(z)))\varphi(z)\mathrm{d}z, \\
&= \int_{\mathbb{R}^d} f(m^p + \sigma z) \min(1, \varphi_\tau(z + \Delta)/\varphi_\tau(z))\varphi(z)\mathrm{d}z \\
&\quad - \int_{\mathbb{R}^d} f(m^q + \sigma z) \min(1, \varphi_\tau(\hat{z} + \Delta)/\varphi_\tau(z)))\varphi(z)\mathrm{d}z, \\
&\quad + \int_{\mathbb{R}^d} f(m^q + \sigma z)\varphi(z)\mathrm{d}z \\
&= \int_{\mathbb{R}^d} f(m^q + \sigma z) \min(1, \varphi_\tau(z)/\varphi_\tau(z - \Delta))\varphi(z - \Delta)\mathrm{d}z \\
&\quad - \int_{\mathbb{R}^d} f(m^q + \sigma z) \min(1, \varphi_\tau(\hat{z} + \Delta)/\varphi_\tau(z)))\varphi(z)\mathrm{d}z, \\
&\quad + \int_{\mathbb{R}^d} f(m^q + \sigma z)\varphi(z)\mathrm{d}z.
\end{aligned}
$$

We have that $\widehat{\hat{z} + \Delta} = z - \Delta$. Hence, we get that

$$
\begin{aligned}
\mathbb{E}[f(Y)] = & \int_{\mathbb{R}^d} f(m^q + \sigma z) \min(1, \varphi_\tau(z)/\varphi_\tau(z - \Delta)) \varphi(z - \Delta) \mathrm{d}z \\
& - \int_{\mathbb{R}^d} f(m^q + \sigma z) \min(1, \varphi_\tau(\hat{z} + \Delta)/\varphi_\tau(z))) \varphi(z) \mathrm{d}z, \\
& + \int_{\mathbb{R}^d} f(m^q + \sigma z) \varphi(z) \mathrm{d}z \\
= & \int_{\mathbb{R}^d} f(m^q + \sigma z) \min(\varphi(z - \Delta)/\varphi(z), (\varphi_\tau(z)\varphi(z - \Delta))/(\varphi_\tau(z - \Delta)\varphi(z))) \varphi(z) \mathrm{d}z \\
& - \int_{\mathbb{R}^d} f(m^q + \sigma z) \min(1, \varphi_\tau(\hat{z} + \Delta)/\varphi_\tau(z))) \varphi(z) \mathrm{d}z, \\
& + \int_{\mathbb{R}^d} f(m^q + \sigma z) \varphi(z) \mathrm{d}z \\
= & \int_{\mathbb{R}^d} f(m^q + \sigma z) \min(\varphi(z - \Delta)/\varphi(z), (\varphi_\tau(z)\varphi(z - \Delta))/(\varphi_\tau(z - \Delta)\varphi(z))) \varphi(z) \mathrm{d}z \\
& - \int_{\mathbb{R}^d} f(m^q + \sigma z) \min(1, \varphi_\tau(z - \Delta)/\varphi_\tau(z))) \varphi(z) \mathrm{d}z, \\
& + \int_{\mathbb{R}^d} f(m^q + \sigma z) \varphi(z) \mathrm{d}z.
\end{aligned}
$$

Hence, we get that

$$
\mathbb{E}[f(Y)] = \int_{\mathbb{R}^d} f(m^q + \sigma z)(1 + a_\tau(z)) \varphi(z) \mathrm{d}z, \tag{17}
$$

where

$$
a_\tau(z) = -\min(1, \varphi_\tau(z - \Delta)/\varphi_\tau(z))) + \min((\varphi_\tau(z)\varphi(z - \Delta))/(\varphi_\tau(z - \Delta)\varphi(z)), \varphi(z - \Delta)/\varphi(z)).
$$

Note that for $\tau = 1$ we have that $a_\tau(z) = 0$. If we let $\tau \to +\infty$ then we get that $a_\tau(z) = -1 + \varphi(z - \Delta)/\varphi(z)$ so $Y \sim \mathcal{N}(m^p, \sigma^2 \mathrm{Id})$ from (17), i.e., we always accept the draft model. In Figure 5, we show the effect of the temperature on the output distribution. The influence of the temperature in more realistic settings is studied in Appendix I.

**Link with guidance.** We first give the following result and subsequently explain its connections with (Karras et al., 2024).

> **Proposition F.1 (Link with guidance):** *Let $(\tilde{Y}, Y)$ be the output of Algorithm 7. We have that*
>
> $$\mathbb{E}[Y] = m^q + (1/\sigma)C_\tau(\|\Delta\|)(m^p - m^q),$$
>
> *with $\Delta = (m^p - m^q)/\sigma$ and $C_\tau(\|\Delta\|) \leq 0$ if $\tau \leq 1$ and $C_\tau(\|\Delta\|) \geq 0$ otherwise. In particular, we have that $C_\tau(\|\Delta\|) = 0$ if $\tau = 1$. In addition, $C_\tau(\|\Delta\|)$ is explicit in the proof.*

We can interpret this result as follows. For $\tau = 1$, we recover that the mean of $Y$ is the mean of the target as expected as we have a maximal coupling in this case so $Y$ follows the correct target distribution. For $\tau > 1$, we increase the acceptance probability: this has intuitively the effect of moving the distribution of $Y$ towards the distribution of $\tilde{Y}$. Looking at the mean, we can interpret this effect as a guidance effect, where we push towards $m^p$ and away for $m^q$. For $\tau < 1$, we are pushing towards the target distribution even more than with $\tau = 1$. Looking at the mean of $Y$, we can interpret this effect as a guidance term, i.e., pushing away from the draft model and towards the target model. This last setting is similar to (Karras et al., 2024), which consider an explicit guidance of a "good" model with a "bad" model.

*Proof.* Using (17), we have that

$$
\begin{aligned}
\mathbb{E}[Y] &= \int_{\mathbb{R}^d} (m^q + \sigma z)(1 + a_\tau(z)) \varphi(z) \mathrm{d}z \\
&= m^q + m^q \int_{\mathbb{R}^d} a_\tau(z) \varphi(z) \mathrm{d}z + \sigma \int_{\mathbb{R}^d} z a_\tau(z) \varphi(z) \mathrm{d}z.
\end{aligned}
$$

First, we show that $\int_{\mathbb{R}^d} a_\tau(z)\mathrm{d}z = 0$. Indeed, using the change of variable $z \mapsto -z$ and $z \mapsto z - \Delta$ we get

$$
\begin{aligned}
\int_{\mathbb{R}^d} a_\tau(z)\varphi(z)\mathrm{d}z &= \int_{\mathbb{R}^d} \min((\varphi_\tau(z)\varphi(z-\Delta))/(\varphi_\tau(z-\Delta)\varphi(z)), \varphi(z-\Delta)/\varphi(z))\varphi(z)\mathrm{d}z \\
&\quad - \int_{\mathbb{R}^d} \min(1, \varphi_\tau(z-\Delta)/\varphi_\tau(z)))\varphi(z)\mathrm{d}z \\
&= \int_{\mathbb{R}^d} \min((\varphi_\tau(z)\varphi(z+\Delta))/(\varphi_\tau(z+\Delta)\varphi(z)), \varphi(z+\Delta)/\varphi(z))\varphi(z)\mathrm{d}z \\
&\quad - \int_{\mathbb{R}^d} \min(1, \varphi_\tau(z-\Delta)/\varphi_\tau(z)))\varphi(z)\mathrm{d}z \\
&= \int_{\mathbb{R}^d} \min((\varphi_\tau(z-\Delta)\varphi(z))/(\varphi_\tau(z)\varphi(z-\Delta)), \varphi(z)/\varphi(z-\Delta))\varphi(z-\Delta)\mathrm{d}z \\
&\quad - \int_{\mathbb{R}^d} \min(1, \varphi_\tau(z-\Delta)/\varphi_\tau(z)))\varphi(z)\mathrm{d}z \\
&= \int_{\mathbb{R}^d} \min((\varphi_\tau(z-\Delta)\varphi(z))/\varphi_\tau(z), \varphi(z))\mathrm{d}z \\
&\quad - \int_{\mathbb{R}^d} \min(\varphi(z), \varphi(z)\varphi_\tau(z-\Delta)/\varphi_\tau(z)))\mathrm{d}z = 0.
\end{aligned}
$$

Hence, we have that

$$
\mathbb{E}[Y] = \int_{\mathbb{R}^d} (m^q + \sigma z)(1 + a_\tau(z))\varphi(z)\mathrm{d}z = m^q + \sigma \int_{\mathbb{R}^d} z a_\tau(z)\varphi(z)\mathrm{d}z.
$$

We are going to show that

$$
\int_{\mathbb{R}^d} \langle z, e \rangle a_\tau(z)\varphi(z)\mathrm{d}z = 0,
$$

where we recall that $e = \Delta/\|\Delta\|$. For any $z \in \mathbb{R}^d$, we have that $z = z_e e + \sum_{i=1}^{d-1} z_{e_i} e_i$, where $z_e = \langle z, e \rangle$ and $z_{e_i} = \langle z, e_i \rangle$ with $\{e, e_i\}_{i=1}^{d-1}$ an orthonormal basis. Note in particular that for any $i \in \{1, \ldots, d-1\}$, $\langle e_i, \Delta \rangle = 0$. We have that

$$
\begin{aligned}
\int_{\mathbb{R}^d} z a_\tau(z)\varphi(z)\mathrm{d}z &= \int_{\mathbb{R}^d} z \min((\varphi_\tau(z)\varphi(z-\Delta))/(\varphi_\tau(z-\Delta)\varphi(z)), \varphi(z-\Delta)/\varphi(z))\varphi(z)\mathrm{d}z \\
&\quad - \int_{\mathbb{R}^d} z \min(1, \varphi_\tau(z-\Delta)/\varphi_\tau(z)))\varphi(z)\mathrm{d}z \\
&= \int_{\mathbb{R}^d} z \min(\varphi_\tau(z)\varphi(z-\Delta)/\varphi_\tau(z-\Delta), \varphi(z-\Delta))\mathrm{d}z \\
&\quad - \int_{\mathbb{R}^d} z \min(\varphi(z), \varphi_\tau(z-\Delta)\varphi(z)/\varphi_\tau(z)))\mathrm{d}z \\
&= -\int_{\mathbb{R}^d} (z-\Delta) \min(\varphi_\tau(z-\Delta)\varphi(z)/\varphi_\tau(z), \varphi(z))\mathrm{d}z \\
&\quad - \int_{\mathbb{R}^d} z \min(\varphi(z), \varphi_\tau(z-\Delta)\varphi(z)/\varphi_\tau(z)))\mathrm{d}z \\
&= -2\int_{\mathbb{R}^d} z \min(\varphi_\tau(z-\Delta)\varphi(z)/\varphi_\tau(z), \varphi(z))\mathrm{d}z \\
&\quad + \Delta \int_{\mathbb{R}^d} \min(\varphi_\tau(z-\Delta)\varphi(z)/\varphi_\tau(z), \varphi(z))\mathrm{d}z. \tag{18}
\end{aligned}
$$

Next, we look at $z \mapsto \min(\varphi_\tau(z-\Delta)\varphi(z)/\varphi_\tau(z), \varphi(z))$. For any $z$, let $z_{e^\perp} = z - z_e e$. Note that $\langle z_{e^\perp}, \Delta \rangle = 0$. We have

that

$$
\begin{aligned}
\min(\varphi_\tau(z - \Delta)\varphi(z)/\varphi_\tau(z), \varphi(z)) \\
&= \min(\varphi_\tau(z_e - \|\Delta\|)\varphi_\tau(z_{e^\perp})\varphi(z_e)\varphi(z_{e^\perp})/(\varphi_\tau(z_e)\varphi_\tau(z_{e^\perp})), \varphi(z_e)\varphi(z_{e^\perp})) \\
&= \min(\varphi_\tau(z_e - \|\Delta\|)\varphi(z_e)\varphi(z_{e^\perp})/\varphi_\tau(z_e), \varphi(z_e)\varphi(z_{e^\perp})) \\
&= \varphi(z_{e^\perp})\min(\varphi_\tau(z_e - \|\Delta\|)\varphi(z_e)/\varphi_\tau(z_e), \varphi(z_e)).
\end{aligned}
$$

Using this result, (18) and the fact that for any $i \in \{1, \ldots, d-1\}$, $\langle e_i, \Delta \rangle = 0$, we get

$$
\left\langle \int_{\mathbb{R}^d} z a_\tau(z)\varphi(z)\mathrm{d}z, e_i \right\rangle = 0.
$$

Therefore, we get that

$$
\mathbb{E}[Y] = m^q + (1/\sigma)C_\tau(\|\Delta\|)(m^p - m^q).
$$

In the rest of the proof, we give an explicit expression for the parameter $C_\tau(\Delta)$. We first find $z$ such that $\varphi_\tau(z_e - \|\Delta\|)\varphi(z_e)/\varphi_\tau(z_e) \leq \varphi(z_e)$, i.e., we find $z$ such that $\log(\varphi_\tau(z_e - \|\Delta\|)) \leq \log(\varphi_\tau(z_e))$, i.e., $z_e \leq \|\Delta\|/2$. In particular, we have that

$$
\begin{aligned}
\int_{\mathbb{R}} &\min(\varphi_\tau(z_e - \|\Delta\|)\varphi(z_e)/\varphi_\tau(z_e), \varphi(z_e))\mathrm{d}z_e \\
&= \int_{-\infty}^{\|\Delta\|/2} \varphi(z_e)\mathrm{d}z_e + \int_{\|\Delta\|/2}^{+\infty} \varphi_\tau(z_e - \|\Delta\|)\varphi(z_e)/\varphi_\tau(z_e)\mathrm{d}z_e \\
&= \Phi(\|\Delta\|/2) + \int_{\|\Delta\|/2}^{+\infty} \varphi_\tau(z_e - \|\Delta\|)\varphi(z_e)/\varphi_\tau(z_e)\mathrm{d}z_e.
\end{aligned}
$$

In addition, we have that

$$
\varphi_\tau(z_e - \|\Delta\|)\varphi(z_e)/\varphi_\tau(z_e) = \varphi(z_e - \|\Delta\|/\tau)\exp[\|\Delta\|^2/\tau^2(1 - \tau)].
$$

Therefore, we get that

$$
\begin{aligned}
\int_{\mathbb{R}} &\min(\varphi_\tau(z_e - \|\Delta\|)\varphi(z_e)/\varphi_\tau(z_e), \varphi(z_e))\mathrm{d}z_e \\
&= \Phi(\|\Delta\|/2) + \exp[\|\Delta\|^2/\tau^2(1 - \tau)](1 - \Phi(\|\Delta\|(\tfrac{1}{2} - \tfrac{1}{\tau}))) \\
&= \Phi(\|\Delta\|/2) + \exp[\|\Delta\|^2/\tau^2(1 - \tau)]\Phi(\|\Delta\|(\tfrac{1}{\tau} - \tfrac{1}{2})).
\end{aligned} \tag{19}
$$

Similarly, we have that

$$
\begin{aligned}
\int_{\mathbb{R}} z_e &\min(\varphi_\tau(z_e - \|\Delta\|)\varphi(z_e)/\varphi_\tau(z_e), \varphi(z_e))\mathrm{d}z_e \\
&= \int_{-\infty}^{\|\Delta\|/2} z_e\varphi(z_e)\mathrm{d}z_e + \int_{\|\Delta\|/2}^{+\infty} z_e\varphi_\tau(z_e - \|\Delta\|)\varphi(z_e)/\varphi_\tau(z_e)\mathrm{d}z_e \\
&= \int_{-\infty}^{\|\Delta\|/2} z_e\varphi(z_e)\mathrm{d}z_e + \exp[\|\Delta\|^2/\tau^2(1 - \tau)]\int_{\|\Delta\|/2}^{+\infty} z_e\varphi(z_e - \|\Delta\|/\tau)\mathrm{d}z_e \\
&= \int_{-\infty}^{\|\Delta\|/2} z_e\varphi(z_e)\mathrm{d}z_e + \exp[\|\Delta\|^2/\tau^2(1 - \tau)]\int_{\|\Delta\|/2}^{+\infty} (z_e - \Delta)\varphi(z_e - \|\Delta\|/\tau)\mathrm{d}z_e \\
&\quad + \|\Delta\|\int_{\|\Delta\|/2}^{+\infty} \exp[\|\Delta\|^2/\tau^2(1 - \tau)]\varphi(z_e - \|\Delta\|/\tau)\mathrm{d}z_e \\
&= \varphi(\|\Delta\|(\tfrac{1}{2} - \tfrac{1}{\tau})) - \varphi(\|\Delta\|/2) + \|\Delta\|\int_{\|\Delta\|/2}^{+\infty} \exp[\|\Delta\|^2/\tau^2(1 - \tau)]\varphi(z_e - \|\Delta\|/\tau)\mathrm{d}z_e \\
&= \varphi(\|\Delta\|(\tfrac{1}{\tau} - \tfrac{1}{2})) - \varphi(\|\Delta\|/2) + \|\Delta\|\exp[\|\Delta\|^2/\tau^2(1 - \tau)]\Phi(\|\Delta\|(\tfrac{1}{\tau} - \tfrac{1}{2})).
\end{aligned} \tag{20}
$$

Combining (19), (20) and (18), we get that

$$
\begin{aligned}
\left\langle e, \int_{\mathbb{R}^d} z a_\tau(z) \mathrm{d}z \right\rangle = {}& -2\varphi(\|\Delta\|(\tfrac{1}{\tau} - \tfrac{1}{2})) + 2\varphi(\|\Delta\|/2) \\
& - 2\exp[\|\Delta\|^2/\tau^2(1-\tau)]\|\Delta\|\Phi(\|\Delta\|(\tfrac{1}{\tau} - \tfrac{1}{2})) + \|\Delta\|\Phi(\|\Delta\|/2) \\
& + \|\Delta\|\exp[\|\Delta\|^2/\tau^2(1-\tau)]\Phi(\|\Delta\|(\tfrac{1}{\tau} - \tfrac{1}{2})) \\
= {}& -2\varphi(\|\Delta\|(\tfrac{1}{\tau} - \tfrac{1}{2})) + 2\varphi(\|\Delta\|/2) \\
& + \|\Delta\|\Phi(\|\Delta\|/2) - \|\Delta\|\exp[\|\Delta\|^2/\tau^2(1-\tau)]\Phi(\|\Delta\|(\tfrac{1}{\tau} - \tfrac{1}{2})).
\end{aligned}
$$

Therefore, we have

$$
\begin{aligned}
C_\tau(\|\Delta\|) = {}& -\tfrac{2}{\|\Delta\|}\varphi(\|\Delta\|(\tfrac{1}{\tau} - \tfrac{1}{2})) + \tfrac{2}{\|\Delta\|}\varphi(\|\Delta\|/2) \\
& + \Phi(\|\Delta\|/2) - \exp[\|\Delta\|^2/\tau^2(1-\tau)]\Phi(\|\Delta\|(\tfrac{1}{\tau} - \tfrac{1}{2})).
\end{aligned}
$$

It can be checked that $C_\tau(\|\Delta\|) \leq 0$ if $\tau \leq 1$ and $C_\tau(\|\Delta\|) \geq 0$ otherwise. In particular, we have that $C_\tau(\|\Delta\|) = 0$ if $\tau = 1$. $\qquad\square$

### F.2. Typical acceptance in the Gaussian case

We adapt here the *typical acceptance* criterion introduced in (Cai et al., 2024) to our setting; i.e. we consider the following acceptance ratio

$$
a(x) = \min(1, \max(q(x)/\kappa, q(x)\exp[\mathrm{H}(q)]/\delta)). \tag{21}
$$

where $\mathrm{H}(q)$ is the differential entropy of $q$, i.e., $\mathrm{H}(q) = -\int_{\mathbb{R}^d} q(x)\log q(x)\mathrm{d}x$. The hyperparameters $\kappa, \delta > 0$ are assumed to be fixed. We recall that $q(x) = \mathcal{N}(x; m^q, \sigma^2 \mathrm{Id})$. In that case we have that

$$
\mathrm{H}(q) = (d/2)(1 + \log(2\pi) + \sigma^2).
$$

Now, if we replace the acceptance criterion in Algorithm 4 with (21) and, if the sample is rejected, apply a deterministic orthogonal transformation $z \mapsto \hat{z}$ to the Gaussian noise to obtain $Y$, we get that for any $f \in \mathrm{C}_c(\mathbb{R}^d)$

$$
\mathbb{E}[f(Y)] = \int_{\mathbb{R}^d} [a(m^p + \sigma z)f(m^p + \sigma z) + (1 - a(m^p + \sigma z))f(m^q + \sigma \hat{z})]\mathcal{N}(z; 0, \mathrm{Id})\mathrm{d}z
$$

Hence, we get

$$
\begin{aligned}
\mathbb{E}[f(Y)] &= \int_{\mathbb{R}^d} [a(m^p + \sigma z)f(m^p + \sigma z) + (1 - a(m^p + \sigma z))f(m^q + \sigma \hat{z})]\mathcal{N}(z; 0, \mathrm{Id})\mathrm{d}z \\
&= \int_{\mathbb{R}^d} [a(m^p + \sigma z)f(m^p + \sigma z) + (1 - a(m^p + \sigma \hat{z}))f(m^q + \sigma z)]\mathcal{N}(z; 0, \mathrm{Id})\mathrm{d}z \\
&= \int_{\mathbb{R}^d} \left[ \frac{\mathcal{N}(z - \Delta; 0, \mathrm{Id})}{\mathcal{N}(z; 0, \mathrm{Id})} a(m^q + \sigma z)f(m^q + \sigma z) + (1 - a(m^p + \sigma \hat{z}))f(m^q + \sigma z) \right] \mathcal{N}(z; 0, \mathrm{Id})\mathrm{d}z \\
&= \int_{\mathbb{R}^d} \left[ \frac{\mathcal{N}(z - \Delta; 0, \mathrm{Id})}{\mathcal{N}(z; 0, \mathrm{Id})} a(m^q + \sigma z) + (1 - a(m^p + \sigma \hat{z})) \right] f(m^q + \sigma z)\mathcal{N}(z; 0, \mathrm{Id})\mathrm{d}z
\end{aligned}
$$

Therefore, we have

$$
\mathbb{E}[f(Y)] = \int_{\mathbb{R}^d} f(m^q + \sigma z)\left( 1 + \frac{\mathcal{N}(z - \Delta; 0, \mathrm{Id})}{\mathcal{N}(z; 0, \mathrm{Id})} a(m^q + \sigma z) - a(m^p + \sigma \hat{z}) \right) \mathcal{N}(z; 0, \mathrm{Id})\mathrm{d}z.
$$

## G. Projection and extension to operators

In this section, we show that we can introduce an acceptance criterion so that two random variables are maximally coupled in a latent space. This relaxes the criterion introduced in Algorithm 4. In particular, it is possible to reach higher acceptance

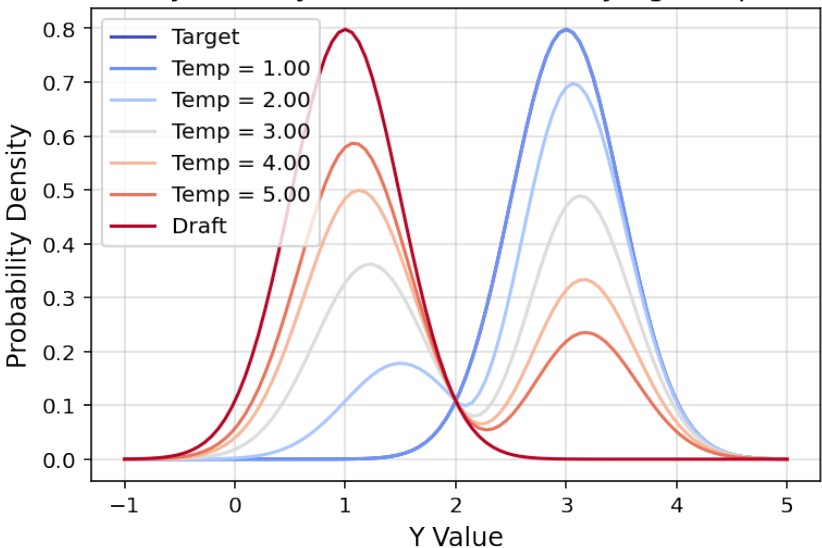

*Figure 5.* Effect of the temperature on the distribution of $Y$. Draft model has mean 1.0 and standard deviation 0.5. Target model has mean 3.0 and standard deviation 0.5.

---

**Algorithm 8** REJECTION $(p_{\mathrm{A}}, q_{\mathrm{A}}, \tilde{Y}_{\mathrm{A}})$ for two Gaussians with same (full) covariance

---

**Require:** matrix A, $p_{\mathrm{A}}(x) = \mathcal{N}(x; \mathrm{A}m^p, \sigma^2 \mathrm{A}\mathrm{A}^\top \mathrm{Id}), q_{\mathrm{A}}(x) = \mathcal{N}(x; \mathrm{A}m^q, \sigma^2 \mathrm{A}\mathrm{A}^\top \mathrm{Id}), \tilde{Y}_{\mathrm{A}} \sim p_{\mathrm{A}}.$
   Set $\Delta_{\mathrm{A}} = (\mathrm{A}\mathrm{A}^\top)^{-1/2} \mathrm{A}(m^p - m^q)/\sigma$ and $e_{\mathrm{A}} = \Delta_{\mathrm{A}}/||\Delta_{\mathrm{A}}||.$
   Let $Z_{\mathrm{A}} = (\mathrm{A}\mathrm{A}^\top)^{-1/2}(\tilde{Y}_{\mathrm{A}} - \mathrm{A}m^p)/\sigma.$
   Sample $U \sim \mathrm{Unif}[0, 1].$
   $\texttt{bool} = \mathbb{I}[U \leq \min(1, \mathcal{N}(Z_{\mathrm{A}} + \Delta_{\mathrm{A}}; 0, \mathrm{Id})/\mathcal{N}(Z_{\mathrm{A}}; 0, \mathrm{Id}))]$
   **if** $\texttt{bool}$ **then**
      Set $Y_{\mathrm{A}} = \tilde{Y}_{\mathrm{A}}.$
   **else**
      Set $Y_{\mathrm{A}} = \mathrm{A}m^q + \sigma(\mathrm{A}\mathrm{A}^\top)^{1/2}(\mathrm{Id} - 2e_{\mathrm{A}}e_{\mathrm{A}}^\top)Z_{\mathrm{A}}.$
   **end if**
   **return** $(Y_{\mathrm{A}}, \texttt{bool}).$

---

rate than with Algorithm 4. Of course, there is a price to pay for this increased flexibility as the variable $Y$ does not follow the target distribution $q$ anymore.

To start with, consider a linear operator $A \in \mathbb{R}^{d \times d}$ such that $AA^\top$ is invertible, i.e., $A$ is surjective. In Algorithm 8, we show how to maximally couple two $d-$dimensional densities of the form $\mathcal{N}(x; Am^p, \sigma^2 AA^\top)$ and $\mathcal{N}(x; Am^q, \sigma^2 AA^\top)$.

Algorithm 8 operates directly in the "latent" space, i.e., it provides a maximal coupling $(\tilde{Y}_A, Y_A)$ where $\tilde{Y}_A \sim \mathcal{N}(Am^p, \sigma^2 AA^\top \mathrm{Id})$ and $Y_A \sim \mathcal{N}(Am^q, \sigma^2 AA^\top \mathrm{Id})$.

We now present Algorithm 9, which is a non-trivial rewriting of Algorithm 8 operating on the original $(\tilde{Y}, Y)$ and thus induces maximally coupled $(\tilde{Y}_A, Y_A)$. In what follows, we denote $A^\dagger$ the Moore-Penrose inverse of $A$ defined by

$$A^\dagger = A^\top (AA^\top)^{-1}.$$

The validity of Algorithm 9 is based on the following lemma.

**Lemma G.1 (Latent reflection):** *Let* $AA^\top$ *be invertible. Let* $\tilde{Y} = m^p + \sigma Z$ *with* $Z \sim \mathcal{N}(0, \mathrm{Id})$. *Let* $Z_A = (AA^\top)^{-1/2} AZ$. *Let* $e_A = \Delta_A / \|\Delta_A\|$ *where* $\Delta_A = (AA^\top)^{-1/2} A \Delta$ *and* $\Delta = (m^p - m^q)/\sigma$. *We have that*

$$Am^q + \sigma (AA^\top)^{1/2} (\mathrm{Id} - 2e_A e_A^\top) Z_A = A \left[ m^q + \sigma \left( Z - 2 \frac{Z^\top A^\dagger A \Delta}{\Delta^\top A^\dagger A \Delta} \Delta \right) \right]. \tag{22}$$

*In addition, we have that*

$$\exp[-\tfrac{1}{2}(\Delta + 2Z)^\top A^\dagger A \Delta] = \mathcal{N}(Z_A + \Delta_A; 0, \mathrm{Id}) / \mathcal{N}(Z_A; 0, \mathrm{Id}). \tag{23}$$

*Proof.* First, we have the following

$$\begin{aligned}
A(m^p - m^q)(m^p - m^q)A^\dagger AZ &= (AA^\top)^{1/2}(AA^\top)^{-1/2}A(m^p - m^q)(m^p - m^q)A^\top(AA^\top)^{-1}AZ \\
&= (AA^\top)^{1/2}(AA^\top)^{-1/2}A(m^p - m^q)(m^p - m^q)A^\top(AA^\top)^{-1/2}Z_A \\
&= \sigma^2 (AA^\top)^{1/2} \Delta_A \Delta_A^\top Z_A.
\end{aligned}$$

Hence, we have that

$$A\Delta\Delta^\top A^\dagger AZ = (AA^\top)^{1/2} \Delta_A \Delta_A^\top Z_A. \tag{24}$$

Next, we have that for any $u \in \mathbb{R}^d$

$$u^\top A^\dagger A u = u^\top A^\top (AA^\top)^{-1} A u = \|(AA^\top)^{-1/2} A u\|^2. \tag{25}$$

Hence, we have that

$$\Delta^\top A^\dagger A \Delta = \|(AA^\top)^{-1/2} \Delta\|^2 = \|\Delta_A\|^2.$$

Combining this result and (24), we have

$$\begin{aligned}
A \left[ m^q + \sigma \left( Z - 2 \frac{Z^\top A^\dagger A \Delta}{\Delta^\top A^\dagger A \Delta} \Delta \right) \right] &= A \left[ m^q + \sigma \left( Z - 2 \frac{\Delta^\top A^\dagger A Z}{\Delta^\top A^\dagger A \Delta} \Delta \right) \right] \\
&= A \left[ m^q + \sigma \left( Z - 2 \frac{\Delta \Delta^\top}{\Delta^\top A^\dagger A \Delta} A^\dagger A Z \right) \right] \\
&= Am^q + \sigma (AA^\top)^{1/2} (\mathrm{Id} - 2e_A e_A^\top) Z_A,
\end{aligned}$$

which concludes the proof of (22). Second, we have that

$$\begin{aligned}
(\Delta + 2Z)^\top A^\dagger A \Delta &= \Delta^\top A^\dagger A \Delta + Z^\top A^\dagger A \Delta + \Delta^\top A^\dagger A Z \\
&= (Z + \Delta)^\top A^\dagger A (Z + \Delta) - Z^\top A^\dagger A Z \\
&= \|(AA^\top)^{-1/2} A (Z + \Delta)\|^2 - \|(AA^\top)^{-1/2} AZ\|^2 \\
&= \|Z_A + \Delta_A\|^2 - \|Z_A\|^2,
\end{aligned}$$

where we have used (25). This concludes the proof of (23). $\qquad \square$

**Algorithm 9** REJECTION $(p, q, \tilde{Y})$ for two Gaussians with same (full) covariance

**Require:** Matrix A, $p(x) = \mathcal{N}(x; m^p, \sigma^2\mathrm{Id}), q(x) = \mathcal{N}(x; m^q, \sigma^2\mathrm{Id}), \tilde{Y} \sim p$.
  $\Delta = (m^p - m^q)/\sigma$
  Sample $U \sim \mathrm{Unif}[0, 1]$.
  $\texttt{bool} = \mathbb{I}[U \leq \min(1, \exp[-\frac{1}{2}(\Delta + 2Z)^\top A^\dagger A\Delta])]$
  **if** $\texttt{bool}$ **then**
    Set $Y = \tilde{Y}$.
  **else**
    Set $Y = m^q + \sigma\left(Z - 2\frac{Z^\top A^\dagger A\Delta}{\Delta^\top A^\dagger A\Delta}\Delta\right)$.
  **end if**
  **return** $(Y, \texttt{bool})$.

The main advantage of Algorithm 9 compared to Algorithm 8 is that it only requires the knowledge of A and $A^\dagger$ and implicitly provide a maximal coupling between $\tilde{Y}_A$ and $Y_A$. Note that if A is invertible, then we have that $A^\dagger = A^{-1}$ and Algorithm 9 becomes identical to Algorithm 4 and thus returns a maximal coupling between $\tilde{Y}$ and $Y$. However, Algorithm 9 is also applicable when only $AA^\top$ is invertible. In that case, we do *not* recover that $Y \sim \mathcal{N}(x; m^q, \sigma^2\mathrm{Id})$ but the algorithm can still be applied and does induce maximally coupled $(\tilde{Y}_A, Y_A)$.

In particular, given a mapping $f$ and a mapping $g$ such that $g(f(x)) \approx x$ for $x \in \mathbb{R}^d$, we can define Algorithm 10, which is a non-linear approximate version of Algorithm 9. In particular, in Algorithm 10, $f$ can be thought of as an encoder and $g$ as a decoder. In the case where $f(x) = Ax$, then $\Delta^\star = g(f(\Delta)) = A^\dagger A\Delta$. Note that by letting $\Delta^\star = \Delta/\tau$ in Algorithm 10, we recover Algorithm 7.

**Algorithm 10** REJECTION $(p, q, \tilde{Y})$ for two Gaussians with auto-encoders

**Require:** $f, g, p(x) = \mathcal{N}(x; m^p, \sigma^2\mathrm{Id}), q(x) = \mathcal{N}(x; m^q, \sigma^2\mathrm{Id}), \tilde{Y} \sim p$.
  $\Delta = (m^p - m^q)/\sigma, \Delta^\star = (g(f(\Delta)))$
  Sample $U \sim \mathrm{Unif}[0, 1]$.
  $\texttt{bool} = \mathbb{I}[U \leq \min(1, \exp[-\frac{1}{2}(\Delta + 2Z)^\top \Delta^\star])]$
  **if** $\texttt{bool}$ **then**
    Set $Y = \tilde{Y}$.
  **else**
    Set $Y = m^q + \sigma\left(Z - 2\frac{Z^\top \Delta^\star}{\Delta^\top \Delta^\star}\Delta\right)$.
  **end if**
  **return** $(Y, \texttt{bool})$.

# H. Some Theoretical Results

In Appendix H.1, we establish Lemma 4.2 while we prove Theorem 4.3 in Appendix H.2.

## H.1. Control of acceptance ratio

We now provide a lower bound on the expectation of the logarithm of the acceptance ratio for speculative sampling. We have at step $n + 1$ that the target density is $q(y_{n+1}|y_n) = \mathcal{N}(y_{n+1}; m^q_{t_n}(y_n), \sigma^2_n\mathrm{Id})$ and, for an independent target model, the proposal density is $p(y_{n+1}|y_n) = \mathcal{N}(y_{n+1}; m^p_{t_n}(y)n), \sigma^2_n\mathrm{Id})$ where

$$m^q_{t_n}(y) = y + \gamma b^q_{1-t_n}(y), \qquad m^p_{t_n}(y) = y + \gamma b^p_{1-t_n}(y)$$

with

$$b^q_{1-t_n}(y) = -f_{1-t_n}y + \frac{g^2_{1-t_n}}{2}s^q_{1-t_n}(y), \quad b^p_{1-t_n}(y) = -f_{1-t_n}y + \frac{g^2_{1-t_n}}{2}s^p_{1-t_n}(y).$$

The acceptance ratio is then given by

$$a_n = \frac{\mathcal{N}(Z + \Delta_n; 0, \mathrm{Id})}{\mathcal{N}(Z; 0, \mathrm{Id})}$$

for $Z \sim \mathcal{N}(Z; 0, \mathrm{Id})$ where

$$\|\Delta_n\|^2 = \frac{1}{4}\gamma(\varepsilon + \tfrac{1}{\varepsilon})^2 g_{1-t_n}^2 \|s_{1-t_n}^p(\tilde{Y}_n) - s_{1-t_n}^q(\tilde{Y}_n)\|^2.$$

So we obtain that

$$\log a_n = -\frac{1}{2}\|Z + \Delta_n\|^2 + \frac{1}{2}\|Z\|^2$$

so that

$$\mathbb{E}[\log(a_n)|y_n] = -\frac{1}{2}\|\Delta_n\|^2.$$

Now using Jensen's inequality

$$\mathbb{E}[a_n] \geq \exp[\mathbb{E}[\log(a_n)]] = \exp\left[-\frac{1}{2}\mathbb{E}[\|\Delta_n\|^2]\right].$$

This proves the result.

## H.2. Control of acceptance ratio under exact scores

We consider the following setting. Let $(\mathbf{X}_t^i)_{t\in[0,1]}$ for any $i \in \{0, 1\}$ be given by

$$\mathrm{d}\mathbf{X}_t^i = f_t \mathbf{X}_t^i \mathrm{d} + g_t \mathrm{d}\mathbf{B}_t, \qquad \mathbf{X}_0^i \sim \pi_0^i$$

where $f : [0, 1) \to \mathbb{R}$ and $g : [0, 1) \to [0, +\infty)$ are functions introduced further, $\pi_0^0$ and $\pi_0^1$ are distributions over $\mathbb{R}^d$ and $(\mathbf{B}_t^i)_{t\in[0,1]}$ are $d$-dimensional Brownian motions. In what follows, we define for any $t \in [0, 1)$

$$f_t = -1/(1-t), \qquad g_t^2 = 2t/(1-t).$$

In that case, we have that for any $t \in [0, 1]$ and $i \in \{0, 1\}$

$$\mathbf{X}_t^i = \alpha_t \mathbf{X}_0^i + \sigma_t \mathbf{Z}, \qquad \mathbf{Z} \sim \mathcal{N}(0, \mathrm{Id}) \tag{26}$$

with $\alpha_t = 1 - t$ and $\sigma_t = t$. We assume that for any $i \in \{0, 1\}$, $\pi_0^i$ has a density with respect to the Lebesgue measure denoted $p_0^i$. In that case, for any $t \in [0, 1]$ and $i \in \{0, 1\}$, $\mathbf{X}_t^i$ admits a density with respect to the Lebesgue measure denoted $p_t^i$. In this section, we show that for any $t \in (0, 1]$

$$\int_{\mathbb{R}^d} \|\nabla \log p_t^0(x_t) - \nabla \log p_t^1(x_t)\|^2 p_t^0(x_t)\mathrm{d}x_t \leq C(t, p_0^0, p_0^1), \tag{27}$$

such that

1. $\lim_{t\to 1} C(t, p_0^0, p_0^1) = 0$,

2. $D(p_0^0|p_1^0) \to 0$ implies that $C(t, p_0^0, p_0^1) \to 0$, where $D$ is a measure of divergence between $p_0^0$ and $p_0^1$ defined further.

In other words, item 1) shows that the Fisher score between $p_t^0$ and $p_t^1$ gets smaller as $t$ gets larger as expected as $p_1^0 = p_1^1$, a normal density. Item 2) shows that the Fisher score between $p_t^0$ and $p_t^1$ is small if $p_0^0$ and $p_0^1$ are close.

We will also establish in our main result, Theorem H.9, a lower bound for the expectation of the logarithm of the acceptance ratio in our speculative sampling setting based on (27).

**Time control.** First, we provide an upper-bound on the Fisher score that goes to 0 as $t \to 1$. We begin with the following result.

**Lemma H.1 (Convergence of Fisher score):** *Assume that $\int_{\mathbb{R}^d} \|x\|^2 \mathrm{d}\pi_0^i(x) = C_2^i < +\infty$ for $i \in \{0, 1\}$. Then, we have that for any $t \in (0, 1]$ and $i \in \{0, 1\}$*

$$\int_{\mathbb{R}^d} \|\nabla \log p_t^i(x_t) - \nabla \log p_1(x_t)\|^2 p_t^i(x_t) \mathrm{d}x_t \leq (\tfrac{1}{\sigma_t} - \sigma_t)^2 d + \alpha_t^2 C_2^i,$$

*where $p_1$ is the density of $\mathcal{N}(0, \mathrm{Id})$ with respect to the Lebesgue measure. In addition, assume that $\int_{\mathbb{R}^d} \|x\|^4 \mathrm{d}\pi_0^i(x) = C_4^i < +\infty$, we have that for any $t \in (0, 1]$ and $i \in \{0, 1\}$*

$$\int_{\mathbb{R}^d} \|\nabla \log p_t^i(x_t) - \nabla \log p_1(x_t)\|^4 p_t^i(x_t) \mathrm{d}x_t \leq 3(\tfrac{1}{\sigma_t} - \sigma_t)^4 d^2 + \alpha_t^4 C_4^i + 6\alpha_t^2 (\tfrac{1}{\sigma_t} - \sigma_t)^2 C_2^i d$$

$$\leq 12(\tfrac{1}{\sigma_t} - \sigma_t)^4 d^2 + 2\alpha_t^4 C_4^i.$$

*Proof.* Let $i \in \{0, 1\}$. First, using Tweedie's identity, see (Vincent, 2011) for instance, we recall that for any $t \in (0, 1)$, we have that for any $x_t \in \mathbb{R}^d$

$$\nabla \log p_t^i(x_t) = \int_{\mathbb{R}^d} \nabla \log p_{t|0}(x_t|x_0) \, p_{0|t}^i(x_0|x_t) \mathrm{d}x_0 = \mathbb{E}[-\mathbf{Z}/\sigma_t \mid \mathbf{X}_t^i = x_t],$$

where we recall that $\mathbf{X}_t^i = \alpha_t \mathbf{X}_0^i + \sigma_t \mathbf{Z}$, see (26). Hence, using Jensen's inequality, we have that

$$\int_{\mathbb{R}^d} \|\nabla \log p_t^i(x_t) - \nabla \log p_1(x_t)\|^2 p_t^i(x_t) \mathrm{d}x_t = \mathbb{E}[\|\mathbb{E}[\mathbf{Z}/\sigma_t - \mathbf{X}_t^i \mid \mathbf{X}_t^i]\|^2]$$

$$\leq \mathbb{E}[\|(\tfrac{1}{\sigma_t} - \sigma_t)\mathbf{Z} - \alpha_t \mathbf{X}_0^i\|^2]$$

$$\leq (\tfrac{1}{\sigma_t} - \sigma_t)^2 \mathbb{E}[\|\mathbf{Z}\|^2] + \alpha_t^2 \mathbb{E}[\|\mathbf{X}_0^i\|^2],$$

where we have used that $\mathbf{X}_0^i$ and $\mathbf{Z}$ are independent. Finally, using $\mathbb{E}[\|\mathbf{Z}\|^2] = d$, we obtained the first result. The second part of the proof is similar and left to the reader. $\qquad\square$

We recall that for any $\alpha \geq 1$ the $\chi_\alpha$ divergence between two densities over $\mathbb{R}^d$, $p, q$ is given by

$$\chi_\alpha(p|q) = \int_{\mathbb{R}^d} \left(1 - \frac{p(x)}{q(x)}\right)^\alpha q(x) \mathrm{d}x.$$

If $\alpha = 2$, we also have

$$\chi_2(p|q) = \int_{\mathbb{R}^d} \frac{p(x)^2}{q(x)} \mathrm{d}x - 1. \tag{28}$$

In addition, we have the following useful result.

**Lemma H.2 ($\chi_\alpha$-data processing inequality):** *For any $\alpha \geq 1$, $t \in [0, 1]$, $\chi_\alpha(p_t^0|p_t^1) \leq \chi_\alpha(p_0^0|p_0^1)$.*

Note that this data processing is in fact valid for every $f$-divergence with $f$ convex. Combining Lemma H.1 and Lemma H.2, we have the following result.

**Lemma H.3 (Convergence of modified Fisher score):** *Assume that $\int_{\mathbb{R}^d} \|x\|^4 \mathrm{d}\pi_0^i(x) = C_4^i < +\infty$ for $i \in \{0, 1\}$. Let $C_4 = \max(C_4^0, C_4^1)$. Then, we have that for any $t \in (0, 1]$*

$$\int_{\mathbb{R}^d} \|\nabla \log p_t^1(x_t) - \nabla \log p_1(x_t)\|^2 p_t^0(x_t) \mathrm{d}x_t \leq 4(1 + \chi_2(p_0^0|p_0^1))^{1/2}((\tfrac{1}{\sigma_t} - \sigma_t)^2 d + \alpha_t^2 C_4^{1/2}),$$

*where $p_1$ is the density of $\mathcal{N}(0, \mathrm{Id})$ with respect to the Lebesgue measure.*

*Proof.* For any $t \in (0,1)$, let $A_t = \int_{\mathbb{R}^d} \|\nabla \log p_t^0(x_t) - \nabla \log p_1(x_t)\|^2 p_t^0(x_t) \mathrm{d}x_t$. Using the Cauchy–Schwarz inequality and (28), we have that for any $t \in (0,1)$

$$
\begin{aligned}
A_t^2 &= \left( \int_{\mathbb{R}^d} \|\nabla \log p_t^1(x_t) - \nabla \log p_1(x_t)\|^2 \frac{p_t^0(x_t)}{p_t^1(x_t)} p_t^1(x_t) \mathrm{d}x_t \right)^2 \\
&\leq \int_{\mathbb{R}^d} \|\nabla \log p_t^1(x_t) - \nabla \log p_1(x_t)\|^4 p_t^1(x_t) \mathrm{d}x_t \int_{\mathbb{R}^d} \frac{p_t^0(x_t)^2}{p_t^1(x_t)} \mathrm{d}x_t \\
&\leq \int_{\mathbb{R}^d} \|\nabla \log p_t^1(x_t) - \nabla \log p_1(x_t)\|^4 p_t^1(x_t) \mathrm{d}x_t (1 + \chi_2(p_t^0 | p_t^1)).
\end{aligned}
$$

We conclude upon combining Lemma H.1, Lemma H.2, the fact that for any $a, b \geq 0$, $\sqrt{a+b} \leq \sqrt{a} + \sqrt{b}$ and that $\max(\sqrt{12}, \sqrt{2}) \leq 4$. $\qquad\square$

Finally, combining Lemma H.3 and Lemma H.1, we get the following result.

**Proposition H.4 (Control of Fisher score (I)):** *Assume that $\int_{\mathbb{R}^d} \|x\|^4 \mathrm{d}\pi_0^i(x) = C_4^i < +\infty$ for $i \in \{0,1\}$. Let $C_4 = \max(C_4^0, C_4^1)$. Then, we have that for any $t \in (0,1]$*

$$
\int_{\mathbb{R}^d} \|\nabla \log p_t^0(x_t) - \nabla \log p_t^1(x_t)\|^2 p_t^0(x_t) \mathrm{d}x_t \leq 10(1 + \chi_2(p_0^0 | p_0^1))^{1/2} \left( (\tfrac{1}{\sigma_t} - \sigma_t)^2 d + \alpha_t^2 C_4^{1/2} \right).
$$

*Proof.* For any $t \in (0,1)$, we have that

$$
\int_{\mathbb{R}^d} \|\nabla \log p_t^0(x_t) - \nabla \log p_t^1(x_t)\|^2 p_t^0(x_t) \mathrm{d}x_t
$$

$$
\leq 2 \int_{\mathbb{R}^d} \|\nabla \log p_t^0(x_t) - \nabla \log p_1(x_t)\|^2 p_t^0(x_t) \mathrm{d}x_t
$$

$$
+ 2 \int_{\mathbb{R}^d} \|\nabla \log p_t^1(x_t) - \nabla \log p_1(x_t)\|^2 p_t^0(x_t) \mathrm{d}x_t.
$$

We conclude upon combining Lemma H.1 and Lemma H.3. $\qquad\square$

In particular, Proposition H.4 shows that $\lim_{t \to 1} \int_{\mathbb{R}^d} \|\nabla \log p_t^0(x_t) - \nabla \log p_t^1(x_t)\|^2 p_t^0(x_t) \mathrm{d}x_t = 0$.

**Measure control.** We now provide a control on the Fisher score that depends on some divergence between the measures $\pi_0^0$ and $\pi_0^1$. We first recall a useful result on the score which can be found, for instance, in (De Bortoli et al., 2024).

**Lemma H.5 (Target Score Identity):** *Assume that for any $i \in \{0,1\}$, $p_0^i \in C^1(\mathbb{R}^d, \mathbb{R}^d)$ and for any $t \in [0,1]$ and $x_t \in \mathbb{R}^d$, $\int_{\mathbb{R}^d} \|\nabla \log p_0^i(x_0)\| p_{0|t}^i(x_0|x_t) \mathrm{d}x_0 < +\infty$. Then, we have that for any $i \in \{0,1\}$, $t \in [0,1)$ and $x_t \in \mathbb{R}^d$*

$$
\nabla \log p_t^i(x_t) = \tfrac{1}{\alpha_t} \int_{\mathbb{R}^d} \nabla \log p_0^i(x_0) p_{0|t}^i(x_0|x_t) \mathrm{d}x_0.
$$

Next, we show the following result.

**Lemma H.6 (Posterior control):** *We have that for any $\alpha \geq 2$, $\alpha$ even and $t \in (0,1)$*

$$
\int_{\mathbb{R}^d} \chi_\alpha(p_{0|t}^1(x_0|x_t) | p_{0|t}^0(x_0|x_t)) p_t^0(x_t) \mathrm{d}x_t \leq D_{0,\alpha}(\chi_{4\alpha}(p_0^0 | p_0^1)^{1/4} + \chi_{4\alpha}(p_0^1 | p_0^0)^{1/4}),
$$

*with*

$$
D_{0,\alpha} \leq 2^{2\alpha - \frac{3}{2}} (1 + \chi_{2\alpha}(p_0^1 | p_0^0))^{1/2} (1 + \chi_2(p_0^0 | p_0^1))^{1/4}.
$$

*Proof.* First, we have that for any $\alpha \geq 2$, $\alpha$ even and $t \in (0, 1)$

$$\int_{\mathbb{R}^d} \chi_\alpha(p^1_{0|t}(x_0|x_t)|p^0_{0|t}(x_0|x_t))p^0_t(x_t)\mathrm{d}x_t = \int_{\mathbb{R}^d \times \mathbb{R}^d} \left(1 - \frac{p^1_{0|t}(x_0|x_t)}{p^0_{0|t}(x_0|x_t)}\right)^\alpha p^0_{0,t}(x_0, x_t)\mathrm{d}x_0\mathrm{d}x_t$$

$$= \int_{\mathbb{R}^d \times \mathbb{R}^d} \left(1 - \frac{p^1_0(x_0)p^0_t(x_t)}{p^0_0(x_0)p^1_t(x_t)}\right)^\alpha p^0_{0,t}(x_0, x_t)\mathrm{d}x_0\mathrm{d}x_t$$

$$\leq 2^{\alpha-1}\int_{\mathbb{R}^d} \left(1 - \frac{p^1_0(x_0)}{p^0_0(x_0)}\right)^\alpha p_0(x_0)\mathrm{d}x_0$$

$$+ 2^{\alpha-1}\int_{\mathbb{R}^d \times \mathbb{R}^d} \left(\frac{p^1_0(x_0)}{p^0_0(x_0)}\right)^\alpha \left(1 - \frac{p^0_t(x_t)}{p^1_t(x_t)}\right)^\alpha p^0_{0,t}(x_0, x_t)\mathrm{d}x_0\mathrm{d}x_t$$

$$= 2^{\alpha-1}\chi_\alpha(p^1_0|p^0_0) + 2^{\alpha-1}\int_{\mathbb{R}^d \times \mathbb{R}^d} \left(\frac{p^1_0(x_0)}{p^0_0(x_0)}\right)^\alpha \left(1 - \frac{p^0_t(x_0)}{p^1_t(x_0)}\right)^\alpha p^0_{0,t}(x_0, x_t)\mathrm{d}x_0\mathrm{d}x_t. \tag{29}$$

Next, we note that for any $\beta \geq 1, \beta$ even and densities $p, q$

$$\int_{\mathbb{R}^d} \left(\frac{q(x)}{p(x)}\right)^\beta p(x)\mathrm{d}x \leq 2^{\beta-1}(1 + \chi_\beta(q|p)).$$

Using this result and (29), we have that

$$\int_{\mathbb{R}^d} \chi_\alpha(p^1_{0|t}(x_0|x_t)|p^0_{0|t}(x_0|x_t))p^0_t(x_t)\mathrm{d}x_t$$

$$\leq 2^{\alpha-1}\chi_\alpha(p^1_0|p^0_0) + 2^{\alpha-1}\int_{\mathbb{R}^d \times \mathbb{R}^d} \left(\frac{p^1_0(x_0)}{p^0_0(x_0)}\right)^\alpha \left(1 - \frac{p^0_t(x_t)}{p^1_t(x_t)}\right)^\alpha p^0_{0,t}(x_0, x_t)\mathrm{d}x_0\mathrm{d}x_t$$

$$\leq 2^{\alpha-1}\chi_\alpha(p^1_0|p^0_0) + 2^{\alpha-1}2^{\frac{2\alpha-1}{2}}(1 + \chi_{2\alpha}(p^1_0|p^2_0))^{1/2}\left(\int_{\mathbb{R}^d} \left(1 - \frac{p^0_t(x_t)}{p^1_t(x_t)}\right)^{2\alpha} p^0_t(x_t)\mathrm{d}x_t\right)^{1/2}$$

$$\leq 2^{\alpha-1}\chi_\alpha(p^1_0|p^0_0) + 2^{2\alpha-\frac{3}{2}}(1 + \chi_{2\alpha}(p^1_0|p^0_0))^{1/2}\left(\int_{\mathbb{R}^d} \left(1 - \frac{p^0_t(x_0)}{p^1_t(x_0)}\right)^{2\alpha} \frac{p^0_t(x_t)}{p^1_t(x_t)}p^1_t(x_t)\mathrm{d}x_t\right)^{1/2}$$

$$\leq 2^{\alpha-1}\chi_\alpha(p^1_0|p^0_0) + 2^{2\alpha-\frac{3}{2}}(1 + \chi_{2\alpha}(p^1_0|p^0_0))^{1/2}(1 + \chi_2(p^0_t|p^1_t))^{1/4}\chi_{4\alpha}(p^0_t|p^1_t)^{1/4}$$

$$\leq 2^{\alpha-1}\chi_\alpha(p^1_0|p^0_0) + 2^{2\alpha-\frac{3}{2}}(1 + \chi_{2\alpha}(p^1_0|p^0_0))^{1/2}(1 + \chi_2(p^0_0|p^1_0))^{1/4}\chi_{4\alpha}(p^0_0|p^1_0)^{1/4}$$

$$\leq 2^\alpha\chi_{4\alpha}(p^1_0|p^0_0)^{1/4} + 2^{2\alpha-\frac{3}{2}}(1 + \chi_{2\alpha}(p^1_0|p^0_0))^{1/2}(1 + \chi_2(p^0_0|p^1_0))^{1/4}\chi_{4\alpha}(p^0_0|p^1_0)^{1/4}$$

$$\leq 2^{2\alpha-\frac{3}{2}}(1 + \chi_{2\alpha}(p^1_0|p^0_0))^{1/2}(1 + \chi_2(p^0_0|p^1_0))^{1/4}(\chi_{4\alpha}(p^0_0|p^1_0)^{1/4} + \chi_{4\alpha}(p^1_0|p^0_0)^{1/4})$$

where we used the data processing inequality. This concludes the proof. $\square$

Finally, for ease of notation, we introduce for any $t \in [0, 1]$

$$\mathrm{FI}(p^0_t|p^1_t) = \int_{\mathbb{R}^d} \|\nabla \log p^0_t(x_t) - \nabla \log p^1_t(x_t)\|^2 p^0_t(x_t)\mathrm{d}x_t.$$

We obtain the following result.

**Proposition H.7 (Control of Fisher score (II)):** *Assume that for any $i \in \{0, 1\}$, $p^i_0 \in \mathrm{C}^1(\mathbb{R}^d, \mathbb{R}^d)$ and for any $t \in [0, 1]$ and $x_t \in \mathbb{R}^d$, $\int_{\mathbb{R}^d} \|\nabla \log p^i_0(x_0)\|p^i_{0|t}(x_0|x_t)\mathrm{d}x_0 < +\infty$. In addition, assume that for any $i \in \{0, 1\}$, $\int_{\mathbb{R}^d} \|\nabla \log p^i_0(x_0)\|^4(p^0_0(x_0) + p_1(x_0))\mathrm{d}x_0 = D^i_4 < +\infty$. Then for any $t \in [0, 1)$, we have*

$$\int_{\mathbb{R}^d} \|\nabla \log p^0_t(x_t) - \nabla \log p^1_t(x_t)\|^2 p^0_t(x_t)\mathrm{d}x_t \leq \tfrac{2D}{\alpha^2_t}(\mathrm{FI}(p^0_0|p^0_0) + \chi_{16}(p^1_0|p^0_0)^{1/8} + \chi_{16}(p^0_0|p^1_0)^{1/8}),$$

*where $D$ is explicit in the proof.*

*Proof.* For any $t \in (0,1)$, let $A_t = \int_{\mathbb{R}^d} \|\nabla \log p_t^0(x_t) - \nabla \log p_1(x_t)\|^2 p_t^0(x_t) \mathrm{d}x_t$. Using Lemma H.5, we have that for any $t \in (0,1)$

$$A_t = \frac{1}{\alpha_t^2} \int_{\mathbb{R}^d} \left\| \int_{\mathbb{R}^d} \nabla \log p_0^0(x_0) p_{0|t}^0(x_0|x_t) \mathrm{d}x_0 - \int_{\mathbb{R}^d} \nabla \log p_0^1(x_0) p_{0|t}^1(x_0|x_t) \mathrm{d}x_0 \right\|^2 p_t^0(x_t) \mathrm{d}x_t.$$

Hence, for any $t \in (0,1)$, $A_t \leq \frac{2}{\alpha_t^2}(A_t^1 + A_t^2)$ with

$$A_t^1 = \int_{\mathbb{R}^d} \left\| \int_{\mathbb{R}^d} \nabla \log p_0^0(x_0) p_{0|t}^0(x_0|x_t) \mathrm{d}x_0 - \int_{\mathbb{R}^d} \nabla \log p_0^1(x_0) p_{0|t}^0(x_0|x_t) \mathrm{d}x_0 \right\|^2 p_t^0(x_t) \mathrm{d}x_t,$$

$$A_t^2 = \int_{\mathbb{R}^d} \left\| \int_{\mathbb{R}^d} \nabla \log p_0^1(x_0) p_{0|t}^0(x_0|x_t) \mathrm{d}x_0 - \int_{\mathbb{R}^d} \nabla \log p_0^1(x_0) p_{0|t}^1(x_0|x_t) \mathrm{d}x_0 \right\|^2 p_t^0(x_t) \mathrm{d}x_t.$$

Using Jensen's inequality, we have that for any $t \in (0,1)$

$$A_t^1 \leq \int_{\mathbb{R}^d} \|\nabla \log p_0^0(x_0) - \nabla \log p_0^1(x_0)\|^2 p_0^0(x_0) \mathrm{d}x_0. \tag{30}$$

Second, using Jensen's inequality, the Cauchy–Schwarz inequality and Lemma H.6

$$A_t^2 \leq \int_{\mathbb{R}^d \times \mathbb{R}^d} \|\nabla \log p_0^1(x_0)\|^2 \left( 1 - \frac{p_{0|t}^1(x_0|x_t)}{p_{0|t}^0(x_0|x_t)} \right)^2 p_{0,t}^0(x_0, x_t) \mathrm{d}x_0 \mathrm{d}x_t$$

$$\leq \left( \int_{\mathbb{R}^d} \|\nabla \log p_0^1(x_0)\|^4 p_0^0(x_0) \mathrm{d}x_0 \right)^{1/2} \left( \int_{\mathbb{R}^d \times \mathbb{R}^d} \left( 1 - \frac{p_{0|t}^1(x_0|x_t)}{p_{0|t}^0(x_0|x_t)} \right)^4 p_{0,t}^0(x_0, x_t) \mathrm{d}x_0 \mathrm{d}x_t \right)^{1/2}$$

$$\leq D_{0,4}^{1/2} \left( \int_{\mathbb{R}^d} \|\nabla \log p_0^1(x_0)\|^4 p_0^0(x_0) \mathrm{d}x_0 \right)^{1/2} (\chi_{16}(p_0^1|p_0^0)^{1/8} + \chi_{16}(p_0^0|p_0^1)^{1/8}).$$

Combining this result and (30) concludes the proof with $D = 2(1 + D_{0,4}^{1/2} \max(D_4^0, D_4^1))$. $\qquad\square$

Finally, combining Proposition H.4 and Proposition H.7, we get the following proposition.

---

**Proposition H.8 (Control Fisher (III)):** *Assume that for any $i \in \{0,1\}$, $p_0^i \in \mathrm{C}^1(\mathbb{R}^d, \mathbb{R}^d)$ and for any $t \in [0,1]$ and $x_t \in \mathbb{R}^d$, $\int_{\mathbb{R}^d} \|\nabla \log p_0^i(x_0)\| p_{0|t}^i(x_0|x_t) \mathrm{d}x_0 < +\infty$. In addition, assume that for any $i \in \{0,1\}$, $\int_{\mathbb{R}^d} \|\nabla \log p_0^i(x_0)\|^4 (p_0^0(x_0) + p_1(x_0)) \mathrm{d}x_0 = D_4^i < +\infty$. Assume that $\int_{\mathbb{R}^d} \|x\|^4 \mathrm{d}\pi_0^i(x) = C_4^i < +\infty$ for $i \in \{0,1\}$. Then, we have for any $t \in (0,1)$*

$$\int_{\mathbb{R}^d} \|\nabla \log p_t^0(x_t) - \nabla \log p_t^1(x_t)\|^2 p_t^0(x_t) \mathrm{d}x_t$$

$$\leq C \min \left( (\tfrac{1}{\sigma_t} - \sigma_t)^2 + \alpha_t^2, \tfrac{1}{\alpha_t^2}(\mathrm{FI}(p_0^0|p_0^1) + \chi_{16}(p_0^1|p_0^0)^{1/8} + \chi_{16}(p_0^0|p_0^1)^{1/8}) \right),$$

*where $C \geq 0$ can be made explicit.*

---

**Control of acceptance ratio.** We now provide a lower bound on the expectation of the logarithm of the acceptance ratio in the speculative sampling framework. We consider a discretization of the interval $[0,1]$ given by $K \in \mathbb{N}$ and $t_k = k/K$. We let $\gamma = 1/K$. We consider the target model given for any $k \in \{0, K-1\}$ by

$$Y_{k+1}^t = Y_k^t + \gamma\{-f_{1-t_k} Y_k^t + \tfrac{1+\varepsilon^2}{2} g_{1-t_k}^2 \nabla \log p_{1-t_k}^0(Y_k)\} + \sqrt{\gamma} g_{1-t_k} Z_k^t, \qquad Y_0 \sim \mathcal{N}(0, \mathrm{Id}),$$

where $(Z_k^t)_{k \in \mathbb{N}} \overset{\text{i.i.d.}}{\sim} \mathcal{N}(0, \mathrm{Id})$. We now $k_0 \in \{0, \ldots, N-1\}$, $L \in \mathbb{N}$, $k_L = \min(N-1, k_0 + L - 1)$ and consider the draft model associated given for any $k \in \{k_0, k_L\}$ by

$$Y_{k+1}^d = Y_k^d + \gamma\{-f_{1-t_k} Y_k^d + \tfrac{1+\varepsilon^2}{2} \nabla \log p_{1-t_k}^1(Y_k)\} + \sqrt{\gamma} g_{1-t_k} Z_k^d, \qquad Y_{k_0}^d = Y_{k_0}^t \sim \mathcal{N}(0, \mathrm{Id}),$$

where $(Z_k^d)_{k\in\mathbb{N}} \overset{\text{i.i.d.}}{\sim} \mathcal{N}(0, \text{Id})$.

The step $k_0 + 1$ is accepted if $U \leq \min(1, \mathcal{N}(Z_{k_0}^t + \Delta, \text{Id})/\mathcal{N}(Z_{k_0}^t, \text{Id}))$ with

$$\|\Delta_{k_0}\|^2 = \frac{1}{4}\gamma(\varepsilon + \tfrac{1}{\varepsilon})^2 \gamma g_{1-t_{k_0}}^2 \|\nabla \log p_{1-t_{k_0}}^1(Y_{k_0}^t) - \nabla \log p_{1-t_{k_0}}^0(Y_{k_0}^t)\|^2.$$

So we obtain

$$\mathbb{E}[\log(a_{k_0})] = -\frac{1}{2}\mathbb{E}[\|\Delta_{k_0}\|^2].$$

Combining this result with the previous Proposition, we obtain the following result.

**Theorem H.9 (Control of log-acceptance ratio):** *Assume that for any* $i \in \{0, 1\}$, $p_0^i \in \mathrm{C}^1(\mathbb{R}^d, \mathbb{R}^d)$ *and for any* $t \in [0, 1]$ *and* $x_t \in \mathbb{R}^d$, $\int_{\mathbb{R}^d} \|\nabla \log p_0^i(x_0)\| p_{0|t}^i(x_0|x_t)\mathrm{d}x_0 < +\infty$. *In addition, assume that for any* $i \in \{0, 1\}$, $\int_{\mathbb{R}^d} \|\nabla \log p_0^i(x_0)\|^4 (p_0^0(x_0) + p_1(x_0))\mathrm{d}x_0 = D_4^i < +\infty$. *Assume that* $\int_{\mathbb{R}^d} \|x\|^4 \mathrm{d}\pi_0^i(x) = C_4^i < +\infty$ *for* $i \in \{0, 1\}$. *In addition, assume that* $Y_{k_0}^t \sim p_{1-t_{k_0}}$ *then*

$$\mathbb{E}[\log(a_{k_0})]$$
$$\geq -\frac{C}{8}(\varepsilon + \tfrac{1}{\varepsilon})^2 \gamma g_{s_0}^2 \min\left(\left(\tfrac{1}{\sigma_{s_0}} - \sigma_{s_0}\right)^2 + \alpha_{s_0}^2, \tfrac{1}{\alpha_{s_0}^2}(\mathrm{FI}(p_0^0|p_0^1) + \chi_{16}(p_0^1|p_0^0)^{1/8} + \chi_{16}(p_0^0|p_0^1)^{1/8})\right),$$

*where* $s_0 = 1 - t_{k_0}$ *and* $C \geq 0$ *is explicit in the proof.*

The final result is obtained by using Jensen's inequality, i.e, $\mathbb{E}[a_{k_0}] \geq \exp[\mathbb{E}[\log(a_{k_0})]]$.

Let us interpret Theorem H.9. We aim at maximizing $\log(a_{k_0})$ since a high acceptance ratio yields a lower computational cost of the speculative sampling method. We here give a lower bound on its expectation. There are different factors that influence this bound:

- $\gamma \to 0$ yields $\mathbb{E}[\log(a_{k_0})] \geq 0$. Hence a small discretization step is associated with better acceptance of the method. However, we emphasize that a small discretization step also gives a larger total number of steps. Hence the benefits of reducing the stepsize must be weighted by the additional computational requirement of having to run the speculative procedure for a larger number of iterations.

- If $p_0^0 \to p_0^1$ (in Fisher and $\chi_4$ divergence) then $\mathbb{E}[\log(a_{k_0})] \geq 0$. This means that if during speculative sampling, the two models target similar distribution then we obtain a higher acceptance rate. This remark is verified empirically in ... and echoes similar findings in LLMs (Cai et al., 2024).

- If $g_t^2((\tfrac{1}{\sigma_t} - \sigma_t)^2 + \alpha_t^2) \to 0$ as $t \to 1$ then $\mathbb{E}[\log(a_{k_0})] \geq 0$. Hence, in that case for low values of $k_0$, i.e., at the beginning of the denoising process, the acceptance rate is high. This observation is also confirmed empirically and is specific to the diffusion model setting.

In our setting, we have

$$g_t^2((\tfrac{1}{\sigma_t} - \sigma_t)^2 + \alpha_t^2) = 2t(1-t)(1 + (1 + 1/t)^2).$$

Hence $\lim_{t\to 1} g_t^2((\tfrac{1}{\sigma_t} - \sigma_t)^2 + \alpha_t^2) = 0$.

We showcase the $A(t) = g_t^2((\tfrac{1}{\sigma_t} - \sigma_t)^2 + \alpha_t^2)$ in Figure 6.

# I. Experimental details

In this section, we provide details about our experimental setup in Appendix I.1. Our setting for the low dimensional Gaussian Mixture Models (GMMs) is described in Appendix I.2. Similarly, our setting for the image experiments is given in Appendix I.3.

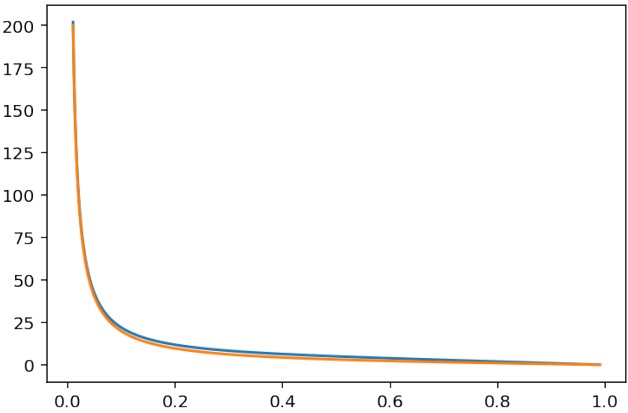

*Figure 6.* The value of $A(t)$ as a function of $t$ for $\alpha_t = 1 - t$ and $\sigma_t = t$ (blue). The value of $A(t)$ for $\alpha_t = \cos((\pi/2)t)$ and $\sigma_t = \sin((\pi/2)t)$ (orange).

## I.1. Experiment setting.

In our setting, we consider the stochastic interpolant framework (Albergo et al., 2023) for greater flexibility. Namely, we consider a noising interpolant given by

$$\mathbf{X}_t = \alpha_t \mathbf{X}_0 + \sigma_t \mathbf{X}_1, \qquad \mathbf{X}_0 \sim \pi_0, \ \mathbf{X}_1 \sim \mathcal{N}(0, \mathrm{Id}), \tag{31}$$

where $t \mapsto \alpha_t$ is a non-increasing function and $t \mapsto \sigma_t$ is a non-decreasing function so that $\alpha_1 = 0$, $\sigma_1 = 1$. The interpolation (31) can be associated with the following forward process

$$\mathrm{d}\mathbf{X}_t = f_t \mathbf{X}_t \mathrm{d}t + g_t \mathrm{d}\mathbf{B}_t, \qquad \mathbf{X}_0 \sim \pi, \tag{32}$$

where for any $t \in (0, 1)$

$$f_t = \partial_t \log(\alpha_t), \qquad g_t^2 = 2\alpha_t \sigma_t \partial_t(\sigma_t/\alpha_t).$$

The time-reversal of the noising process (32) is given by $(\mathbf{Y}_t)_{t \in [0,1]}$ which satisfies

$$\mathrm{d}\mathbf{Y}_t = \left\{ -f_{1-t}\mathbf{Y}_t + g_{1-t}^2 \nabla \log p_{1-t}(\mathbf{Y}_t) \right\} \mathrm{d}t + g_{1-t} \mathrm{d}\mathbf{B}_t, \qquad \mathbf{Y}_0 \sim p_1 \tag{33}$$

where $p_t$ is the density of $\mathbf{X}_t$ with respect to the Lebesgue measure. In practice, we do not typically know $p_1$ and let $\mathbf{Y}_0 \sim \mathcal{N}(0, \sigma_1^2 \mathrm{Id})$. For a given hyperparameter $\varepsilon > 0$, one can also consider

$$\mathrm{d}\mathbf{Y}_t = \left\{ -f_{1-t}\mathbf{Y}_t + \tfrac{1}{2}(1 + \varepsilon^2) g_{1-t}^2 \nabla \log p_{1-t}(\mathbf{Y}_t) \right\} \mathrm{d}t + \varepsilon g_{1-t} \mathrm{d}\mathbf{B}_t, \tag{34}$$

which has the same marginals as (33). This can also be rewritten as

$$\mathrm{d}\mathbf{Y}_t = \left\{ -v_{1-t}(\mathbf{Y}_t) + \tfrac{1}{2}\varepsilon^2 g_{1-t}^2 \nabla \log p_{1-t}(\mathbf{Y}_t) \right\} \mathrm{d}t + \varepsilon g_{1-t} \mathrm{d}\mathbf{B}_t, \tag{35}$$

where the so-called velocity $v_t$ is given by

$$v_t(x) = \mathbb{E}[\partial_t \alpha_t \mathbf{X}_0 + \partial_t \sigma_t \mathbf{X}_1 \mid \mathbf{X}_t = x].$$

Upon combining (34) and (35), we have that for any $t \in (0, 1)$

$$\nabla \log p_t(x) = -\mathbb{E}[\mathbf{X}_1 \mid \mathbf{X}_t = x]/\sigma_t = \frac{2}{g_t^2}(f_t x - v_t(x)). \tag{36}$$

In particular, in order to estimate the score function, we only need to estimate the velocity and vice-versa. In practice, we consider the following loss function

$$\mathcal{L}_\theta = \int_0^1 w_t \mathbb{E}[\|\partial_t \alpha_t \mathbf{X}_0 + \partial_t \sigma_t \mathbf{X}_1 - v_{\theta,t}(\mathbf{X}_t)\|^2] \mathrm{d}t,$$

where $w_t > 0$ is a weighting function, see Esser et al. (2024) for some possible choices for $w_t$. We denote $s_\theta$ the score estimated from $v_\theta$ using (36). At sampling time, we consider the Euler–Maruyama discretisation of (35). More precisely, we define some timesteps $\{t_i\}_{i=0}^N$ with $0 = t_0 < t_1 < \cdots < t_N = 1$ and consider the following Markov chain

$$Y_{k+1} = Y_k + \gamma_k \Big\{ v_{\theta, 1-t_k}(Y_k) + \frac{1}{2}\varepsilon^2 g_{1-t_k}^2 s_{\theta, 1-t_k}(Y_k) \Big\} + \sqrt{\gamma_k}\varepsilon g_{1-t_k} Z_k, \tag{37}$$

where $(Z_k)_{k \in \mathbb{N}}$ is a sequence of independent and identically distributed Gaussian random variables with zero mean and identity covariance matrix and $\gamma_k = t_{k+1} - t_k$. When additional conditioning information is available, one can consider an additional guidance term and (37) is changed into

$$Y_{k+1} = Y_k + \gamma_k \Big\{ (1+\delta)v_{\theta, 1-t_k}(Y_k, c) - \delta v_{\theta, 1-t_k}(Y_k, \emptyset) + \frac{1}{2}\varepsilon^2 g_{1-t_k}^2 s_{\theta, 1-t_k}(Y_k) \Big\} + \sqrt{\gamma_k}\varepsilon g_{1-t_k} Z_k,$$

where $v_{\theta, t}(\cdot, c)$ corresponds to a *conditional* model and $v_{\theta, t}(\cdot, \emptyset)$ to an unconditional one.

### I.2. Low dimensional experiments.

In our low-dimensional setting, we create a dataset by sampling from a mixture of Gaussians. The means are sampled uniformly and independently from $[-2, 2]^d$, where $d$ is the dimension. Each Gaussian component has a covariance matrix of the form $\sigma^2 \mathrm{Id}$, where the standard deviation $\sigma$ is also sampled uniformly and independently from $[0.1, 0.2]$. We test across dimensions $d \in \{2, 4, 8, 16, 32\}$ and numbers of components $n \in \{1, 2, 4, 8, 16\}$.

The velocity of the diffusion model is parameterized with a sequence of MLPs. For all MLP we use the GeLU activation function. The label, corresponding to the component of the mixture is encoded using a layer embedding layer with feature dimension 512. Similarly, the time information is encoded using sinusoidal embedding with feature dimension 512. This encoding is then processed with a MLP with output dimension 512. The time embedding and the label embedding are then concatenated into a conditioning embedding. The conditioning embedding and the input $x_t$ of the velocity network are then processed independently with 3 MLP layers with output dimension $(64, 64, 128)$. The obtained embedding are then concatenated and processed with 3 MLP layers with output dimension $(128, 64, 64)$. Finally a last dense layer with output dimension $d$ is added. We do not consider any normalisation layer. In the case of the training of an independent draft model, the three preprocessing MLP layers are replaced with one MLP layer with output dimension 4. Similarly, the three postprocessing MLP layers are replaced with one MLP layer with output dimension 4. For the sampling, we use 250 sampling steps. We refer to Appendix J for additional results.

### I.3. Image experiments

All FID and IS scores are evaluated with $50,000$ images.

**CIFAR10.** The shape of the samples in the training dataset is $(32 \times 32 \times 3)$. The batch size is set to 128. Images are rescaled between $-1.0$ and $1.0$. We consider an augmentation pipeline similar to the one of (Karras et al., 2022). The augmentation pipeline is applied with a global probability $p = 0.12$. The rest of the augmentation pipeline is similar to the one used in (Karras et al., 2022). In particular, we consider flipping (both $x$ and $y$ axis), anisotropy transformations, non-integer rotation, scaling and non-integer translation.

For the model, we consider a $U$-net architecture with GeLU activations, 4 levels with a residual attention block applied on the second level. The channel multipliers are given by $(1, 2, 2, 2)$. The channel size is 256. We consider a dropout rate of 0.2. The normalization layers are RMS normalization layers. For the attention layers we consider 8 heads. The number of residual blocks is 2. For the skip connection, we add (and normalize) the current activation with the stored activation. The time is embedded using sinusoidal embedding with hidden dimension 256. We embed the 10 different classes and consider conditional models. We also condition the model on the augmentation vector. These two conditionings are added and the time embedding is added on top of this. The conditioning occurs through adaptive normalization layers. We train the model for $1M$ steps with the Adam optimizer and a learning rate of $10^{-4}$ and EMA decay of 0.9999.

**LSUN.** We consider the same configuration as CIFAR10. However, the samples do not have label and we only consider the augmentation conditioning.

## I.4. Latent CIFAR-10 experiments

In the first stage, as an auto-encoder, we use variational auto-encoder (VAE) (Kingma & Welling, 2014) with a smaller term $\beta$ on the KL-term as in $\beta$-VAE (Higgins et al., 2016). The encoder and decoder are represented by U-Net where in encoder, U-Net follows only the downsampling and middle bottleneck paths, while in decoder, U-Net follows middle bottleneck and upsampling paths, which is similar to what is used in (Rombach et al., 2022). We use 128 channels for the corresponding U-Nets without attention in downsampling/upsampling but with attention in the middle (1 head) and with channel multipliers $(1, 2, 4, 4)$. We use SILU activation function. We also use RMSNorm for normalization as opposed to GroupNorm. We also employ perceptual LPSIS loss (Zhang et al., 2018) with coefficient 1 as well as patch-based discriminator as in (Esser et al., 2021). The dimensionality of the latent space is $(4, 4, 32)$ which is 6 times smaller than the original CIFAR-10 image dimensionality $(32, 32, 3)$. We train the autoencoder for 500000 steps. We track the FID on the subset of 12800 images comparing clean images and the reconstructions `decoder(encoder(x))`, and select hyperparameters which achieve the smallest FID. The selected hyperparameters as well as their ranges are:

- Number of discriminator filters = 32. Range $[32, 64, 128]$.

- Number of discriminator layers = 6. Range $[3, 6, 9]$.

- Dropout rate for both encoder and decoder = 0.0. Range $[0, 0.1, 0.2, 0.3]$.

- $\beta$ parameter = $1e - 6$. Range $[1e - 4, 5e - 5, 1e - 5, 5e - 6, 1e - 6, 5e - 7, 1e - 7]$.

- Generator loss coefficient = 0.01. Range $[0.001, 0.01, 0.1, 1.0]$.

- Adversarial loss coefficient = 0.001. Range $[0.001, 0.01, 0.1, 1.0]$.

- Batch size = 1024. Range $[128, 256, 512, 1024]$

For the second stage, we freeze the encoder and decoder and train a diffusion model on the encoded images (we take the means), similar to (Rombach et al., 2022). We use the U-Net with 256 channels, $(2, 2)$ channel multipliers with attention performed (False, True), with attention in the middle with 8 attention heads, RMSNorm, GeLU activation. We train latent diffusion for 160000 iterations with batch size 256. We track FID on the subset of 12800 to select the hyperparameters. The selected hyperparameters as well as their ranges are:

- Prediction target = $x_0$. Range $x_0$ or velocity

- U-Net dropout rate = 0.0. Range $[0, 0.1, 0.2, 0.3]$.

- Learning rate = $1e - 4$. Range $[1e - 3, 1e - 4, 5e - 5]$

- Noise process type = cosine. Range - linear, cosine, rectified flow

Once the models are trained, we employ the same sampling strategy as in CIFAR-10 experiment.

## I.5. PushT dataset.

We consider the PushT dataset. The task here is to push a $T$ shape onto a target shape on a two dimensional plane. The action dimension is 2, the action horizon is 8. We keep an history of length 2 and consider a maximum of 300 steps when unrolling the policy. We consider a prediction horizon of length 16. This means that the dimension of the target is $16 \times 2$. And we condition on the last two previous states (dimension is 5). Hence the conditioning signal has shape $(2 \times 5)$. Once we have predicted 16 actions we execute 8 of them. As specified before we execute a maximum of 300 steps or stop if the reward reaches one, i.e., the $T$ shape is perfectly aligned.

We train the model for $1M$ steps with Adam and stepsize $10^{-4}$. We consider a one-dimensional U-net with time embedding. The architecture follows from (Chi et al., 2023).

At inference time, we rely on the DDPM sampler.

## J. Additional results

We run similar experiments in latent space to showcase the flexibility of our method. We follow the approach described in (Rombach et al., 2022) – we pre-train an autoencoder on the whole dataset and then train a diffusion model on the latent-encoded dataset. We consider the latent space of shape ($4 \times 4 \times 32$) which is 6 times smaller than the dimensionality ($32 \times 32 \times 3$) of CIFAR10. We refer to Appendix I.4 for architectural and training details. We report FID score computed on 50k training samples. Our results are reported in Table 4. We found that using latent diffusion on CIFAR-10 achieved better FID score when the target used only 30 sampling iterations. Nevertheless, we see that our speculative sampling method still provides 3-x speed-up (best is NFE) while maintaining similar to target model quality. We also considered using target model with only 10 NFEs and Table 4 suggests that it achieves considerably worse results. This highlights the strength of our approach.

**Combining speculative sampling and parallel sampling.** We report FID score and NFE for CIFAR-10 with a number of steps of 30. We vary the temperature parameter $\tau$, the churn parameter $\varepsilon$ as well as the number of parallel iterations, see (Shih et al., 2023; Tang et al., 2024). For each combination of hyperparameters we also consider window sizes 5, 10 and 20 and report the best run (in terms of FID).

The original speculative sampling procedure corresponds to $p = 0$. The best FID number that can be achieved with this configuration is 2.23 with a NFE of 15.69. However, by combining our speculative sampling procedure with parallel sampling then we can reach a FID of 2.07 with a NFE of 15.42. This shows the benefits of combining our speculative sampling procedure with other acceleration methods. We report those results in Table 8.

**Combining speculative sampling and step distillation.** We now compare our approach with LD3 (Tong et al., 2024) and (Sabour et al., 2024). We compare the results on CIFAR-10 as reported in LD3 (Tong et al., 2024). Our best speculative sampling method outperformed both LD3 and AYS. We also included our best results obtained with a uniform timesteps spacing and EDM timestep spacing (Karras et al., 2022). These results are based on the same model as "Best speculative". We sweep over $\rho = [1.0, \ldots, 8.0]$ in the case of EDM timestep spacing. This improves the quality of the samples but they remain inferior in quality to the ones obtained with our best speculative model. We re-implemented LD3 (Tong et al., 2024) in our setting and used it to learn a timestep spacing. Our setting is similar to the one of (Tong et al., 2024). Finally, we compare our approach with a distilled generator trained on top of our best model. We focus on Multistep Moment Matching Distillation (MMD) (Salimans et al., 2024).

| Configuration | FID | NFE |
|---|---|---|
| DPM Solver++ (naive - reported) | 2.37 | 20 |
| DPM Solver++ (AYS (Sabour et al., 2024) - reported) | 2.10 | 20 |
| DPM Solver++ (LD3 (Tong et al., 2024) - reported) | 2.36 | 20 |
| Uniform timesteps | 7.14 | 15 |
| EDM timesteps | 4.22 | 15 |
| LD3 timesteps | 3.49 | 15 |
| MultiStep Moment Matching | 2.76 | 15 |
| Best speculative | **2.07** | 15.4 |

*Table 3.* Comparison of model configurations, including our best speculative methods against several baselines. The top section shows reported results from prior work, while the bottom section details our experiments.

## K. Accelerating Langevin Diffusions using Speculative Sampling

We detail in this appendix the application of speculative sampling to Langevin diffusions proposed in Section 5. Assume where we are interested in sampling from an unnormalized density $\pi(x)$ on $\mathbb{R}^d$, i.e.

$$\pi(x) = \frac{\exp(-E(x))}{Z}, \qquad Z = \int \exp(-E(x))\mathrm{d}x,$$

where the energy function $E(x)$ can be evaluated pointwise, but each evaluation is computationally expensive, and $Z$ is an intractable normalizing constant. We are interested here in accelerating MCMC sampling in the context where we have

| Configuration | Draft | | Target (30 steps) | | Target (10 steps) | | Speculative | | |
|---|---|---|---|---|---|---|---|---|---|
| | FID ↓ | IS ↑ | FID ↓ | IS ↑ | FID ↓ | IS ↑ | FID ↓ | IS ↑ | NFE ↓ |
| $\varepsilon = 0.01, \tau = 0.5$ | **80.92** | **5.59** | 2.67 | 11.09 | **39.48** | **7.42** | 2.66 | 11.13 | 18.53 |
| $\varepsilon = 0.01, \tau = 1.0$ | **80.92** | **5.59** | 2.67 | 11.09 | **39.48** | **7.42** | 2.66 | 11.14 | 17.78 |
| $\varepsilon = 0.01, \tau = 2.0$ | **80.92** | **5.59** | 2.67 | 11.09 | **39.48** | **7.42** | 2.66 | 11.14 | 17.09 |
| $\varepsilon = 0.25, \tau = 0.5$ | 82.28 | 5.50 | 2.64 | **11.15** | 87.39 | 4.82 | 2.68 | 11.18 | 10.37 |
| $\varepsilon = 0.25, \tau = 1.0$ | 82.28 | 5.50 | 2.64 | **11.15** | 87.39 | 4.82 | 2.66 | **11.23** | 9.36 |
| $\varepsilon = 0.25, \tau = 2.0$ | 82.28 | 5.50 | 2.64 | **11.15** | 87.39 | 4.82 | 2.66 | 11.21 | 8.36 |
| $\varepsilon = 0.5, \tau = 0.5$ | 83.27 | 5.42 | **2.51** | 11.08 | 118.78 | 3.81 | 2.56 | 11.11 | 9.35 |
| $\varepsilon = 0.5, \tau = 1.0$ | 83.27 | 5.42 | **2.51** | 11.08 | 118.78 | 3.81 | **2.50** | 11.12 | 8.30 |
| $\varepsilon = 0.5, \tau = 2.0$ | 83.27 | 5.42 | **2.51** | 11.08 | 118.78 | 3.81 | 2.52 | 11.07 | 7.30 |
| $\varepsilon = 1.0, \tau = 0.5$ | 97.67 | 4.72 | 37.54 | 7.09 | 182.94 | 2.43 | 37.13 | 7.11 | 9.57 |
| $\varepsilon = 1.0, \tau = 1.0$ | 97.67 | 4.72 | 37.54 | 7.09 | 182.94 | 2.43 | 37.85 | 7.07 | 8.36 |
| $\varepsilon = 1.0, \tau = 2.0$ | 97.67 | 4.72 | 37.54 | 7.09 | 182.94 | 2.43 | 38.32 | 7.09 | **7.19** |

*Table 4.* Latent diffusion on CIFAR-10 with window size = 15 for speculative sampling. For each column, we report the best result in **bold**.

| Configuration | Draft | | Target (500 steps) | | Speculative | | |
|---|---|---|---|---|---|---|---|
| | FID ↓ | IS ↑ | FID ↓ | IS ↑ | FID ↓ | IS ↑ | NFE ↓ |
| $\varepsilon = 0.005, \tau = 0.25$ | **5.76** | 1.93 | 4.66 | **2.02** | 4.69 | 2.01 | 305.39 |
| $\varepsilon = 0.005, \tau = 0.5$ | **5.76** | 1.93 | 4.66 | **2.02** | 4.68 | 2.01 | 286.07 |
| $\varepsilon = 0.005, \tau = 1.0$ | **5.76** | 1.93 | 4.66 | **2.02** | 4.70 | 2.01 | 263.56 |
| $\varepsilon = 0.005, \tau = 2.0$ | **5.76** | 1.93 | 4.66 | **2.02** | 4.72 | 2.01 | 238.01 |
| $\varepsilon = 0.01, \tau = 0.25$ | **5.76** | 1.93 | 4.66 | **2.02** | 4.65 | 2.01 | 257.96 |
| $\varepsilon = 0.01, \tau = 0.5$ | **5.76** | 1.93 | 4.66 | **2.02** | 4.67 | 2.01 | 236.04 |
| $\varepsilon = 0.01, \tau = 1.0$ | **5.76** | 1.93 | 4.66 | **2.02** | 4.69 | 2.00 | 211.47 |
| $\varepsilon = 0.01, \tau = 2.0$ | **5.76** | 1.93 | 4.66 | **2.02** | 4.76 | 2.00 | 184.98 |
| $\varepsilon = 0.05, \tau = 0.25$ | 5.97 | 1.91 | 4.66 | 2.01 | 4.48 | 2.02 | 186.63 |
| $\varepsilon = 0.05, \tau = 0.5$ | 5.97 | 1.91 | 4.66 | 2.01 | 4.53 | 2.00 | 164.07 |
| $\varepsilon = 0.05, \tau = 1.0$ | 5.97 | 1.91 | 4.66 | 2.01 | 4.62 | 2.01 | 140.06 |
| $\varepsilon = 0.05, \tau = 2.0$ | 5.97 | 1.91 | 4.66 | 2.01 | 4.86 | 1.99 | 116.38 |
| $\varepsilon = 0.1, \tau = 0.25$ | 6.46 | 1.91 | 4.52 | 2.00 | 4.36 | **2.03** | 176.02 |
| $\varepsilon = 0.1, \tau = 0.5$ | 6.46 | 1.91 | 4.52 | 2.00 | 4.38 | 2.02 | 154.73 |
| $\varepsilon = 0.1, \tau = 1.0$ | 6.46 | 1.91 | 4.52 | 2.00 | 4.56 | 1.99 | 131.47 |
| $\varepsilon = 0.1, \tau = 2.0$ | 6.46 | 1.91 | 4.52 | 2.00 | 4.79 | 1.97 | 108.40 |
| $\varepsilon = 0.25, \tau = 0.25$ | 10.11 | 1.96 | **4.13** | 1.96 | 3.94 | 2.01 | 172.65 |
| $\varepsilon = 0.25, \tau = 0.5$ | 10.11 | 1.96 | **4.13** | 1.96 | 3.98 | 1.97 | 153.05 |
| $\varepsilon = 0.25, \tau = 1.0$ | 10.11 | 1.96 | **4.13** | 1.96 | 4.24 | 1.97 | 130.71 |
| $\varepsilon = 0.25, \tau = 2.0$ | 10.11 | 1.96 | **4.13** | 1.96 | 4.53 | 1.96 | **107.92** |
| $\varepsilon = 0.5, \tau = 0.25$ | 17.53 | **2.11** | 4.18 | 1.96 | **4.02** | 1.96 | 178.45 |
| $\varepsilon = 0.5, \tau = 0.5$ | 17.53 | **2.11** | 4.18 | 1.96 | **4.02** | 1.95 | 160.68 |
| $\varepsilon = 0.5, \tau = 1.0$ | 17.53 | **2.11** | 4.18 | 1.96 | 4.26 | 1.93 | 139.16 |
| $\varepsilon = 0.5, \tau = 2.0$ | 17.53 | **2.11** | 4.18 | 1.96 | 4.51 | 1.93 | 116.33 |

*Table 5.* LSUN with window size = 50, no last step function, 500 steps. For each column, we report the best result in **bold**.

| Configuration | Draft | | Target (200 steps) | | Speculative | | |
|---|---|---|---|---|---|---|---|
| | FID ↓ | IS ↑ | FID ↓ | IS ↑ | FID ↓ | IS ↑ | NFE ↓ |
| $\varepsilon = 0.001, \tau = 0.25$ | **10.56** | 1.89 | 3.99 | **1.99** | 3.99 | 1.98 | 176.85 |
| $\varepsilon = 0.001, \tau = 0.5$ | **10.56** | 1.89 | 3.99 | **1.99** | 3.99 | 1.98 | 173.49 |
| $\varepsilon = 0.001, \tau = 1.0$ | **10.56** | 1.89 | 3.99 | **1.99** | 3.99 | 1.98 | 168.23 |
| $\varepsilon = 0.001, \tau = 2.0$ | **10.56** | 1.89 | 3.99 | **1.99** | 3.99 | 1.98 | 160.89 |
| $\varepsilon = 0.005, \tau = 0.25$ | 10.58 | 1.89 | 4.02 | 1.98 | 4.00 | 1.98 | 137.95 |
| $\varepsilon = 0.005, \tau = 0.5$ | 10.58 | 1.89 | 4.02 | 1.98 | 3.99 | 1.98 | 131.53 |
| $\varepsilon = 0.005, \tau = 1.0$ | 10.58 | 1.89 | 4.02 | 1.98 | 3.99 | 1.98 | 124.52 |
| $\varepsilon = 0.005, \tau = 2.0$ | 10.58 | 1.89 | 4.02 | 1.98 | 4.00 | 1.98 | 117.13 |
| $\varepsilon = 0.01, \tau = 0.25$ | 10.63 | 1.89 | 3.99 | 1.98 | 3.98 | 1.98 | 121.26 |
| $\varepsilon = 0.01, \tau = 0.5$ | 10.63 | 1.89 | 3.99 | 1.98 | 3.98 | 1.98 | 114.51 |
| $\varepsilon = 0.01, \tau = 1.0$ | 10.63 | 1.89 | 3.99 | 1.98 | 3.99 | 1.98 | 107.26 |
| $\varepsilon = 0.01, \tau = 2.0$ | 10.63 | 1.89 | 3.99 | 1.98 | 4.01 | 1.98 | 99.20 |
| $\varepsilon = 0.05, \tau = 0.25$ | 12.73 | 1.91 | 3.95 | 1.98 | 3.94 | 1.98 | 92.66 |
| $\varepsilon = 0.05, \tau = 0.5$ | 12.73 | 1.91 | 3.95 | 1.98 | 3.96 | 1.97 | 86.26 |
| $\varepsilon = 0.05, \tau = 1.0$ | 12.73 | 1.91 | 3.95 | 1.98 | 4.03 | 1.96 | 78.75 |
| $\varepsilon = 0.05, \tau = 2.0$ | 12.73 | 1.91 | 3.95 | 1.98 | 4.14 | 1.95 | 70.04 |
| $\varepsilon = 0.1, \tau = 0.25$ | 18.56 | 1.99 | 3.92 | 1.99 | 3.89 | 1.97 | 87.74 |
| $\varepsilon = 0.1, \tau = 0.5$ | 18.56 | 1.99 | 3.92 | 1.99 | 3.93 | **1.99** | 82.05 |
| $\varepsilon = 0.1, \tau = 1.0$ | 18.56 | 1.99 | 3.92 | 1.99 | 3.97 | 1.98 | 74.87 |
| $\varepsilon = 0.1, \tau = 2.0$ | 18.56 | 1.99 | 3.92 | 1.99 | 4.16 | 1.94 | 66.28 |
| $\varepsilon = 0.25, \tau = 0.25$ | 33.76 | 2.28 | **3.83** | 1.94 | 3.76 | 1.96 | 85.60 |
| $\varepsilon = 0.25, \tau = 0.5$ | 33.76 | 2.28 | **3.83** | 1.94 | **3.74** | 1.97 | 80.82 |
| $\varepsilon = 0.25, \tau = 1.0$ | 33.76 | 2.28 | **3.83** | 1.94 | 3.94 | 1.95 | 74.27 |
| $\varepsilon = 0.25, \tau = 2.0$ | 33.76 | 2.28 | **3.83** | 1.94 | 4.12 | 1.94 | **66.01** |
| $\varepsilon = 0.5, \tau = 0.25$ | 49.82 | 2.65 | 4.09 | 1.95 | 3.93 | 1.95 | 87.12 |
| $\varepsilon = 0.5, \tau = 0.5$ | 49.82 | 2.65 | 4.09 | 1.95 | 3.97 | 1.95 | 83.29 |
| $\varepsilon = 0.5, \tau = 1.0$ | 49.82 | 2.65 | 4.09 | 1.95 | 4.14 | 1.93 | 77.55 |
| $\varepsilon = 0.5, \tau = 2.0$ | 49.82 | 2.65 | 4.09 | 1.95 | 4.22 | 1.96 | 69.81 |
| $\varepsilon = 1.0, \tau = 0.25$ | 115.98 | **3.44** | 4.76 | 1.93 | 4.75 | 1.95 | 93.13 |
| $\varepsilon = 1.0, \tau = 0.5$ | 115.98 | **3.44** | 4.76 | 1.93 | 4.73 | 1.97 | 90.88 |
| $\varepsilon = 1.0, \tau = 1.0$ | 115.98 | **3.44** | 4.76 | 1.93 | 4.77 | 1.96 | 87.24 |
| $\varepsilon = 1.0, \tau = 2.0$ | 115.98 | **3.44** | 4.76 | 1.93 | 4.85 | 1.95 | 81.40 |

*Table 6.* LSUN with window size = 50, no last step function, 200 steps. For each column, we report the best result in **bold**.

| Configuration | Draft | | Target (100 steps) | | Speculative | | |
|---|---|---|---|---|---|---|---|
| | FID ↓ | IS ↑ | FID ↓ | IS ↑ | FID ↓ | IS ↑ | NFE ↓ |
| $\varepsilon = 0.001, \tau = 0.25$ | **24.04** | 1.99 | 3.81 | 1.95 | 3.78 | 1.95 | 91.82 |
| $\varepsilon = 0.001, \tau = 0.5$ | **24.04** | 1.99 | 3.81 | 1.95 | 3.79 | 1.95 | 90.57 |
| $\varepsilon = 0.001, \tau = 1.0$ | **24.04** | 1.99 | 3.81 | 1.95 | 3.79 | 1.95 | 88.56 |
| $\varepsilon = 0.001, \tau = 2.0$ | **24.04** | 1.99 | 3.81 | 1.95 | 3.79 | 1.95 | 85.46 |
| $\varepsilon = 0.005, \tau = 0.25$ | 24.26 | 2.00 | 3.81 | 1.95 | 3.78 | 1.95 | 73.13 |
| $\varepsilon = 0.005, \tau = 0.5$ | 24.26 | 2.00 | 3.81 | 1.95 | 3.77 | 1.95 | 70.13 |
| $\varepsilon = 0.005, \tau = 1.0$ | 24.26 | 2.00 | 3.81 | 1.95 | 3.77 | 1.95 | 66.67 |
| $\varepsilon = 0.005, \tau = 2.0$ | 24.26 | 2.00 | 3.81 | 1.95 | 3.77 | 1.95 | 63.10 |
| $\varepsilon = 0.01, \tau = 0.25$ | 25.05 | 2.01 | 3.80 | 1.95 | 3.77 | 1.94 | 65.75 |
| $\varepsilon = 0.01, \tau = 0.5$ | 25.05 | 2.01 | 3.80 | 1.95 | 3.77 | 1.94 | 62.68 |
| $\varepsilon = 0.01, \tau = 1.0$ | 25.05 | 2.01 | 3.80 | 1.95 | 3.77 | 1.94 | 59.59 |
| $\varepsilon = 0.01, \tau = 2.0$ | 25.05 | 2.01 | 3.80 | 1.95 | 3.77 | 1.94 | 56.71 |
| $\varepsilon = 0.05, \tau = 0.25$ | 48.62 | 2.27 | 3.75 | 1.96 | **3.75** | 1.94 | 52.44 |
| $\varepsilon = 0.05, \tau = 0.5$ | 48.62 | 2.27 | 3.75 | 1.96 | 3.76 | 1.93 | 50.07 |
| $\varepsilon = 0.05, \tau = 1.0$ | 48.62 | 2.27 | 3.75 | 1.96 | 3.77 | 1.93 | 47.57 |
| $\varepsilon = 0.05, \tau = 2.0$ | 48.62 | 2.27 | 3.75 | 1.96 | 3.85 | 1.93 | 44.52 |
| $\varepsilon = 0.1, \tau = 0.25$ | 69.55 | 2.53 | **3.74** | 1.95 | 3.78 | 1.94 | 49.65 |
| $\varepsilon = 0.1, \tau = 0.5$ | 69.55 | 2.53 | **3.74** | 1.95 | 3.79 | 1.94 | 47.75 |
| $\varepsilon = 0.1, \tau = 1.0$ | 69.55 | 2.53 | **3.74** | 1.95 | 3.79 | 1.93 | 45.51 |
| $\varepsilon = 0.1, \tau = 2.0$ | 69.55 | 2.53 | **3.74** | 1.95 | 3.86 | 1.91 | 42.52 |
| $\varepsilon = 0.25, \tau = 0.25$ | 97.47 | 3.17 | 3.85 | 1.92 | 3.79 | 1.93 | 48.11 |
| $\varepsilon = 0.25, \tau = 0.5$ | 97.47 | 3.17 | 3.85 | 1.92 | 3.82 | 1.94 | 46.76 |
| $\varepsilon = 0.25, \tau = 1.0$ | 97.47 | 3.17 | 3.85 | 1.92 | 3.81 | 1.93 | 44.92 |
| $\varepsilon = 0.25, \tau = 2.0$ | 97.47 | 3.17 | 3.85 | 1.92 | 3.90 | 1.92 | **42.16** |
| $\varepsilon = 0.5, \tau = 0.25$ | 147.36 | **3.62** | 4.08 | 1.97 | 4.01 | 1.95 | 48.38 |
| $\varepsilon = 0.5, \tau = 0.5$ | 147.36 | **3.62** | 4.08 | 1.97 | 4.06 | 1.96 | 47.43 |
| $\varepsilon = 0.5, \tau = 1.0$ | 147.36 | **3.62** | 4.08 | 1.97 | 4.14 | 1.95 | 46.02 |
| $\varepsilon = 0.5, \tau = 2.0$ | 147.36 | **3.62** | 4.08 | 1.97 | 4.21 | 1.95 | 43.70 |
| $\varepsilon = 1.0, \tau = 0.25$ | 231.66 | 2.74 | 5.76 | **2.02** | 5.72 | 2.00 | 50.08 |
| $\varepsilon = 1.0, \tau = 0.5$ | 231.66 | 2.74 | 5.76 | **2.02** | 5.69 | 2.00 | 49.59 |
| $\varepsilon = 1.0, \tau = 1.0$ | 231.66 | 2.74 | 5.76 | **2.02** | 5.70 | 2.01 | 49.00 |
| $\varepsilon = 1.0, \tau = 2.0$ | 231.66 | 2.74 | 5.76 | **2.02** | 5.65 | **2.02** | 47.89 |

*Table 7.* LSUN with window size = 50, no last step functions, 100 steps. For each column, we report the best result in **bold**.

| Configuration | FID ↓ | NFE ↓ |
|---|---|---|
| $p = 0, \varepsilon = 0.25, \tau = 1.0$ | 2.23 | 15.69 |
| $p = 1, \varepsilon = 0.25, \tau = 1.0$ | 2.09 | 23.80 |
| $p = 5, \varepsilon = 0.25, \tau = 1.0$ | 2.09 | 57.85 |
| $p = 0, \varepsilon = 0.5, \tau = 1.0$ | 2.77 | 17.06 |
| $p = 1, \varepsilon = 0.5, \tau = 1.0$ | 2.75 | 23.42 |
| $p = 5, \varepsilon = 0.5, \tau = 1.0$ | 2.75 | 57.80 |
| $p = 0, \varepsilon = 0.25, \tau = 2.0$ | 2.24 | 14.89 |
| $p = 1, \varepsilon = 0.25, \tau = 2.0$ | 2.09 | 21.12 |
| $p = 5, \varepsilon = 0.25, \tau = 2.0$ | 2.08 | 51.45 |
| $p = 0, \varepsilon = 0.5, \tau = 2.0$ | 2.74 | 16.47 |
| $p = 1, \varepsilon = 0.5, \tau = 2.0$ | 2.77 | 20.62 |
| $p = 5, \varepsilon = 0.5, \tau = 2.0$ | 2.77 | 50.40 |
| $p = 0, \varepsilon = 0.25, \tau = 10.0$ | 2.39 | 12.86 |
| $p = 1, \varepsilon = 0.25, \tau = 10.0$ | **2.07** | 15.42 |
| $p = 5, \varepsilon = 0.25, \tau = 10.0$ | 2.07 | 37.50 |
| $p = 0, \varepsilon = 0.5, \tau = 10.0$ | 2.73 | 14.49 |
| $p = 1, \varepsilon = 0.5, \tau = 10.0$ | 2.79 | 16.38 |
| $p = 5, \varepsilon = 0.5, \tau = 10.0$ | 2.79 | 40.25 |

*Table 8.* Results on CIFAR-10 when combining speculative sampling and parallel sampling. The hyperparameter $p$ represents the number of parallel calls.

access to a computationally cheap proxy energy function $\hat{E}(x) \approx E(x)$ defining $\hat{\pi}(x) \propto \exp(-\hat{E}(x))$. Access to such proxies is common in many domains of computational science and engineering, see e.g. (Peherstorfer et al., 2018) for a review.

In this context, a popular modification of the Metropolis–Hastings (MH) algorithm to sample from $\pi$ leveraging an energy proxy was proposed by Christen & Fox (2005). It is known in the literature as delayed acceptance MH (Cui et al., 2011; Sherlock et al., 2017) or two-stage Markov chain Monte Carlo (Peherstorfer et al., 2018). We present here a completely different approach to accelerate another popular MCMC algorithm, namely the Unadjusted Langevin algorithm (ULA).

The Langevin diffusion is defined by
$$\mathrm{d}\mathbf{X}_t = -\nabla E(\mathbf{X}_t)\mathrm{d}t + \sqrt{2}\mathrm{d}\mathbf{B}_t,$$

where $(\mathbf{B}_t)_{t \geq 0}$ is a standard multivariate Brownian motion. The limiting distribution of this diffusion is $\pi$. Practically, we discretize this diffusion to obtain the ULA algorithm, i.e.

$$X_{k+1} = X_k - \gamma \nabla E(X_k) + \sqrt{2\gamma}W_k, \tag{38}$$

for a stepsize $\gamma > 0$ and $W_k \overset{\text{i.i.d.}}{\sim} \mathcal{N}(0, \mathrm{Id})$. Contrary to MH, this algorithm only samples from an approximation of $\pi$ due to the time-discretization but explicit bounds on the bias incurred are available (Durmus & Moulines, 2017). The speculative sampling algorithm is directly applicable to accelerate the simulation of (38). In this case, (38) plays the role of the target model while

$$X_{k+1} = X_k - \gamma \nabla \hat{E}(X_k) + \sqrt{2\gamma}W_k, \tag{39}$$

corresponds to the draft model. In this case, the general speculative sampling from Algorithm 6 simplifies drastically and we obtain Algorithm 11. As a cheap proxy, we can still use the frozen draft model strategy in this context, that is set $\nabla \hat{E}(x_{n+k}) = \nabla E(x_n)$ for $k = 1, ..., n_L$ in (39). Again speculative sampling returns exact samples from ULA. This method can be thought of as a novel pre-fetching technique to accelerate MCMC (Brockwell, 2006; Angelino et al., 2014).

We now demonstrate the efficiency of Algorithm 11. We consider the $\phi - 4$ model (Guth et al., 2022; Milchev et al., 1986) where the energy function is given by

$$E(x) = \frac{\beta}{2} \sum_{|i-j|=1} (x_i - x_j)^2 + \sum_i (x_i^2 - 1)^2,$$

on a grid of shape $(8, 8)$ and $\beta = 100$. Sampling from $\pi$ is complex as this requires sampling so-called ordered states. In this context, the teacher model is the Langevin diffusion sampling $E(x)$ with $100,000$ iterations and stepsize $10^{-3}$, while our speculative sampling algorithm uses the frozen prediction draft model and a window size of 20. We report the mean and the standard deviation of the energy over the last 500 simulated samples over 500 runs. The NFE is reduced by a factor of 2.

| Metrics | Mean energy | Standard deviation energy | NFE |
|---|---|---|---|
| Langevin sampling | 62.27 | 13.32 | 100000 |
| Speculative sampling | 65.90 | 12.48 | 48564 |

*Table 9.* Comparison of sampling metrics for the $\phi - 4$ model.

---

**Algorithm 11** Speculative Sampling for Unadjusted Langevin Diffusion

---

**Require:** Lookahead integer $L$, sequence length $K$, stepsize $\gamma > 0$, target distribution $\pi$ and proxy distribution $\hat{\pi}$.

 Set $Y_0$ arbitrarily and set $n = 0$.
 **while** $n < K$ **do**
  Set $\tilde{Y}_n \leftarrow Y_n$ and $n_L = \min(n + L, K)$.
  **for** $k = n + 1 : n_L$ **do**
   Set $\tilde{Y}_k = \tilde{Y}_{k-1} - \gamma \nabla \hat{E}(\tilde{Y}_{k-1}) + \sqrt{2\gamma} Z_{k-1}$ for $Z_{k-1} \sim \mathcal{N}(0, \mathrm{Id})$.
  **end for**
  In parallel, compute $\nabla E(\tilde{Y}_n), \nabla E(\tilde{Y}_{n+1}), ..., \nabla E(\tilde{Y}_{n_L-1})$.
  **for** $k = n + 1 : n_L$ **do**
   Set $\Delta_{k-1} = \sqrt{\gamma/2}(\nabla E(\tilde{Y}_{k-1}) - \nabla \hat{E}(\tilde{Y}_{k-1}))$ and $e = \Delta_{k-1}/||\Delta_{k-1}||$.
   Sample $U \sim \mathrm{Unif}[0, 1]$.
   `bool` $= \mathbb{I}[U \leq \min(1, \mathcal{N}(Z_{k-1} + \Delta_{k-1}; 0, \mathrm{Id})/\mathcal{N}(Z_{k-1}; 0, \mathrm{Id}))]$.
   **if** `bool` **then**
    Set $Y_k = \tilde{Y}_k$.
   **else**
    Set $Y_k = \tilde{Y}_{k-1} - \gamma E(\tilde{Y}_{k-1}) + \sqrt{2\gamma}(\mathrm{Id} - 2ee^\top)Z_{k-1}$.
   **end if**
   **return** $(Y_k, \text{bool})$.
   **if** `not(bool)` **then**
    Exit For Loop
   **end if**
  **end for**
  Set $n \leftarrow k$.
 **end while**
 **return** $Y_{0:K}$

---

