# OpenReview forum: "Accelerated Diffusion Models via Speculative Sampling"
_ICML.cc/2025/Conference — ICML 2025 poster_

### Official Review · Reviewer_VXs6 · 2025-03-09

**Overall Recommendation:** 3

**Summary:**

This paper proposed a fast sampling method for diffusion models, inspired by the idea of speculative decoding from LLMs. Using a more compact draft model to efficiently generate an image sequence and then verify the whole sequence in parallel with the original diffusion model, the sampling latency can be shortened. Besides leveraging an independent draft model, the authors reuse $b_t$ approximated at the window start, making the original target model itself a draft model. Theoretical analysis and empirical evaluation are provided to prove the effectiveness of the proposed sampling method.


### Update after rebuttal
I appreciate the feedback from the authors. Overall, I think it is a good paper. I would keep my rating and recommend it for acceptance.

**Claims And Evidence:**

Yes

**Essential References Not Discussed:**

No

**Experimental Designs Or Analyses:**

Yes

**Methods And Evaluation Criteria:**

Yes

**Other Comments Or Suggestions:**

Please see questions for authors.

**Other Strengths And Weaknesses:**

### Strengths:
- The writing is clear. The theoretical analysis is solid.
- The empirical evaluation is supportive.

### Weaknesses:
- Some experimental settings are not clearly described.

**Questions For Authors:**

1. What kind of sampler is used for 30-step sampling in Table 1? And what is the original total number of sampling steps of the target model?

2. The idea of applying speculative decoding to diffusion model sampling is interesting. However, since there are already many fast samplers like DPM-Solver [1] for diffusion models, what are the main advantages of speculative sampling over the other samplers? In this paper, there is no experiment comparing them.

3. In Table 1, is the draft model independent or frozen? If independent, how is NFE computed? Is it only dependent on the calls to the target model?

### Reference:
[1] Lu, C., Zhou, Y., Bao, F., Chen, J., Li, C., & Zhu, J. (2022). Dpm-solver: A fast ode solver for diffusion probabilistic model sampling in around 10 steps. Advances in Neural Information Processing Systems, 35, 5775-5787.

**Relation To Broader Scientific Literature:**

This paper shares a similar idea to speculative decoding for LLMs [1], but to my knowledge, it is the first to apply this idea to diffusion model sampling.

T-stitch [2] leverages the idea of using a smaller draft model for fast sampling, but from a totally different perspective. It didn't share the same theoretical foundation as speculative decoding.

[1] Leviathan, Yaniv, Matan Kalman, and Yossi Matias. "Fast inference from transformers via speculative decoding." International Conference on Machine Learning. PMLR, 2023.

[2] Pan, Zizheng, et al. "T-stitch: Accelerating sampling in pre-trained diffusion models with trajectory stitching." arXiv preprint arXiv:2402.14167 (2024).

**Theoretical Claims:**

Yes

---

> ### Author Rebuttal · Authors · 2025-03-31
>
> We would like to thank the reviewer for their positive assessment of our work.
>
> > This paper shares a similar idea to speculative decoding for LLMs [1], but to my knowledge, it is the first to apply this idea to diffusion model sampling. T-stitch [2] leverages the idea of using a smaller draft model for fast sampling, but from a totally different perspective. It didn't share the same theoretical foundation as speculative decoding.
>
> We thank the reviewer for pointing us to [1]. While [1] also leverages two models, their approach differs from ours. Specifically, they employ a cheap model for early denoising stages and an expensive model for later stages. This strategy does not guarantee the same quality as the original superior model. In contrast, our method samples exactly from the superior model, utilizing the cheap model only for parallel verification, thus maintaining the desired quality. We will however include a discussion regarding this very related work in our updated version of the manuscript.
>
> [1] Pan et al. (2024) – T-Stitch: Accelerating Sampling in Pre-Trained Diffusion Models with Trajectory Stitching
>
> > Some experimental settings are not clearly described.
>
> Below we answer the concerns of the reviewer.
>
> > What kind of sampler is used for 30-step sampling in Table 1? And what is the original total number of sampling steps of the target model?
>
> In Table 1 (and all of our experiments), we used the stochastic sampler of flow matching models, see [1] for instance. An explicit form of the sampler is given in Equation (37), line 1843. In the revised version of the paper, we will highlight Equation (37) in the main paper. We emphasize that our method is compatible with other stochastic methods such as DDPM [2].
>
> [1] Albergo et al. (2023) – Stochastic Interpolants: A Unifying Framework for Flows and Diffusions
>
> [2] Ho et al. (2020) – Denoising Diffusion Probabilistic Models
>
> > The idea of applying speculative decoding to diffusion model sampling is interesting. However, since there are already many fast samplers like DPM-Solver [1] for diffusion models, what are the main advantages of speculative sampling over the other samplers? In this paper, there is no experiment comparing them.
>
> We thank the reviewer for this insightful remark. We emphasize that our method is largely orthogonal to other acceleration techniques. In particular we now combine our approach with [1,2], see our detailed answer to Reviewer zMQr, which yields comparable NFE improvements (including against DPM-Solver++ [3]). Furthermore, our method's applicability extends beyond accelerating denoising diffusion models; it can also be used to accelerate the simulation of any diffusion equation with a computationally expensive drift, such as Langevin diffusion to sample from unnormalized target distributions, see our detailed answer to Reviewer E771. We have added experiments as well as an Appendix in the revised version demonstrating the efficiency of the methodology in this context.
>
> [1] Tong et al. (2024) – Learning to Discretize Denoising Diffusion ODEs
>
> [2] Shih et al. (2023) – Parallel Sampling of Diffusion Models
>
> [3] Lu et al. (2022) – DPM-Solver++: Fast Solver for Guided Sampling of Diffusion Probabilistic Models
>
> > In Table 1, is the draft model independent or frozen? If independent, how is NFE computed? Is it only dependent on the calls to the target model?
>
> To clarify, we employed the frozen strategy for all Image Space and Robotics experiments reported in Table 1. The independent strategy was used exclusively in the low-dimensional setting to demonstrate the superior performance of the frozen draft approach. Consequently, in that specific case, only the target model was evaluated, and the NFE reported corresponds solely to that model. We will explicitly address this distinction in the revised manuscript.
>
> > Versatility of the method
>
> We have now included additional experiments regarding comparison with DPM++ solvers, see answer to Reviewer eBDt, as well as an experiment showcasing the efficiency of the method in a sampling context (i.e. a non generative modeling setting), see answer to Reviewer E771.

---

### Official Review · Reviewer_eBDt · 2025-03-13

**Overall Recommendation:** 3

**Summary:**

This paper extends the method of speculative sampling, which has been used in the context of speeding up inference for autoregressive models, to diffusion models. Roughly, this is a method by which one uses a weaker "draft" model to propose sampling steps which are then accepted with some probability by a strong "target" model via rejection sampling. The rejection step can be interpreted as an optimal coupling between the two models; the naive way to implement this coupling in continuous domains fails to achieve meaningful savings, so the authors utilize a smarter optimal coupling that exploits the fact that the sampling steps of the two models are both Gaussian in distribution. In addition, they find that given a target model, instead of training a weaker draft model, it is preferable to simply take the draft model to be a coarser discretization of the target. With this, they find that speculative sampling can improve sampling efficiency over vanilla stochastic diffusion samplers (e.g. 2.45 in 100 NFEs -> 2.34 in 35 NFEs).

**Claims And Evidence:**

The theoretical claims lower bounding the acceptance ratio under their rejection method are sound. While I appreciated the discussion under Theorem 4.3, it would be helpful to discuss further how to think about $g^2_t((1/\sigma_t - \sigma_t)^2 + \alpha^2_t)$ as a function of $t$ when $t$ is bounded away from $1$, for classical schedules. In general, it is a bit difficult to parse how useful Theorem 4.3 is.

The main claim that needs more support is the sentence "Our approach complements [other approaches for acceleration] and can be combined with...better integrators." This isn't implausible but needs experimental support. The improvement from 100 NFEs to 30 NFEs is not particularly striking in the context of the many works on acceleration, especially the ones that use minimal distillation (e.g. LD3 achieves FID of 2.27 on CIFAR using only 10 NFEs and also doesn't require heavy-duty distillation). I would be more convinced about the proposed approach if, combined with one of these existing approaches, it gets a tangible improvement.

**Essential References Not Discussed:**

As mentioned in the "Claims and Evidence" section above, the authors would have benefited from checking whether their method improves upon existing methods that get comparable or even better speedups, e.g. AYS (https://arxiv.org/pdf/2404.14507), LD3 (https://arxiv.org/abs/2405.15506), etc.

**Experimental Designs Or Analyses:**

I checked the soundness and validity of the image experiments as this is the domain with which I am most familiar. I did not find any issues but found them to be insufficiently comprehensive, see "Methods And Evaluation Criteria" above.

**Methods And Evaluation Criteria:**

It would be better if the authors conducted a more thorough empirical investigation. For instance, the image experiments are limited to improving over vanilla Karras et al.-style diffusion on CIFAR-10 and LSUN. The empirical results would be more compelling if the authors evaluated their method against other acceleration methods of comparable computational cost.

**Other Comments Or Suggestions:**

See "Questions for Authors" below

**Other Strengths And Weaknesses:**

Strengths:
- It is natural to ask for a diffusion analogue to the popular speculative sampling approach for LLMs, and prior to this work, it was not known how to obtain a suitable analogue. The reflection coupling trick used in this paper, while simple, is clever and gives a slick way to implement rejection sampling in this continuous context
- The theoretical claims give nontrivial lower bounds on the acceptance probability
- The experimental results provide preliminary evidence that their method is effective
- The idea to use a coarse-graining of the target model as the draft model is interesting and arguably unique to the diffusions setting

Weaknesses:
- To reiterate, my main complaint is that the actual acceleration achieved by this method is not that impressive compared to existing acceleration methods. I would be more convinced if this method were actually complementary to those other methods as claimed, and I would be happy to raise my score if that is actually the case.
- As the authors note, the method crucially relies on stochasticity in the sampling steps, so it does not apply to ODEs. This appears to be a crucial weakness as ODE-based samplers are generally much more NFE-efficient

**Questions For Authors:**

As discussed above, one thing that would make me much more positive about this paper is whether the acceleration afforded by this method is actually orthogonal to other accelerations. This is tricky as many few-NFE methods are based on ODEs. Is there any setup you could try where you could achieve comparable FID with existing acceleration baselines using, say, 10 NFEs?

**Relation To Broader Scientific Literature:**

There is a vast literature on diffusion generative modeling, with many recent works on acceleration of inference. This paper sits squarely within that broader literature, which is adequately overviewed in the Related Work section modulo a couple works (see Essential References Not Discusses below)

**Theoretical Claims:**

The most nontrivial theoretical result in the paper is the lower bound on the acceptance ratio in Theorem 4.3. I skimmed the proof of this in the supplement and the argument involves some standard manipulations with Fisher divergence and Tweedie's formula.

---

> ### Author Rebuttal · Authors · 2025-03-31
>
> We would like to thank the reviewer for their insightful comments.
>
> > how to think about $g_t$ [...] for classical schedules.
>
> Let $A(t) = g_t^2 ((1/\sigma_t - \sigma_t)^2 + \alpha_t^2)$ and consider a few schedules:
>
> * Rectified flow [1]: $\alpha_t = 1-t$, $\sigma_t = t$. We have that $A(t) = 2 t (1-t) (1 + (1 + 1/t)^2)$.
> * Cosine [2]: $\alpha_t = \cos(\pi t /2)$, $\sigma_t = \sin(\pi t /2)$. We have that $A(t) = \frac{\pi}{2}\sin(\pi t) (1 + \mathrm{cotan}(\tfrac{\pi}{2} t))$.
>
> Note that $A(t) \to 0$ as $t \to 1$ and $A(t) \to +\infty$ as $t \to 0$. Hence the acceptance is close to 1 around $t \approx 1$ and worsens when $t \to 0$. This is consistent with our empirical results. We will include a plot of $A(t)$ as a function of $t$ for those schedules.
>
> [1] Liu et al. (2022) – Flow Straight and Fast
>
> [2] Nichol and Dhariwal (2021) – Improved Denoising Diffusion Probabilistic Models
>
> > difficult to parse how useful Theorem 4.3 is.
>
> We believe that Theorem 4.3 is useful because, although we do not expect our bound to be tight, it captures rigorously the impact of the various hyperparameters of the algorithm on the expected acceptance rate. In particular:
> * that decreasing the stepsize increases the acceptance rate at the cost of increasing the number of total steps.
> * The influence of the schedule (see above).
> * The existence of an optimal value for $\varepsilon$ is also observed in practice, see our experimental section.
>
> >  needs more support is the sentence "Our approach complements [other approaches for acceleration] and can be combined with...better integrators." [...] I would be more convinced about the proposed approach if, combined with one of these existing approaches, it gets a tangible improvement.
>
> We have compared our approach with LD3 [1] and AYS[2]. In addition, to further substantiate our claim we have combined our acceleration method with parallel sampling [3] and observed benefits of the method, see the answer to Reviewer zMQr for more details.
> We compare the results on CIFAR-10 as reported in LD3 [1]. Our best speculative sampling method outperformed both LD3 and AYS.
> We also included our best results obtained with a uniform timesteps spacing and EDM timestep spacing [4]. These results are based on the same model as “Best speculative”. We sweep over $\rho = [1.0, \dots, 8.0]$ in the case of EDM timestep spacing. This improves the quality of the samples but they remain inferior in quality to the ones obtained with our best speculative model. We re-implemented LD3 [1] in our setting and used it to learn a timestep spacing. Our setting is similar to the one of [1]. Finally, we compare our approach with a distilled generator trained on top of our best model. We focus on Multistep Moment Matching Distillation (MMD) [5]. We sweep over several hyperparameters for MMD and report the best result.
>
> | Configuration | FID |  NFE |
> | - | - | - |
> | DPM Solver++ (naive – reported) | 2.37 | 20 |
> | DPM Solver++ (AYS [2] – reported) | 2.10 | 20 |
> | DPM Solver++ (LD3 [1] – reported) | 2.36 | 20 |
> | Uniform timesteps | 7.14 | 15 |
> | EDM timesteps | 4.22 | 15 |
> | LD3 timesteps | 3.49 | 15 |
> | MultiStep Moment Matching |  2.76 | 15 |
> | Best speculative | **2.07**  | 15.42 |
>
> We will include those results in the revised version of our paper. We will also comment on LD3 [1] and AYS [2] in the related work section as these works are indeed complementary to ours.
>
> Finally we note that our method is not restricted to diffusion models. It can be applied to accelerate simulation to sample from unnormalized distributions. We have added an experiment on the $\phi_4$ potential demonstrating the efficiency of the method, see the answer to Reviewer E771.
>
> [1] Tong et al. (2024) – Learning to Discretize Denoising Diffusion ODEs
>
> [2] Sabour et al. (2024) – Align Your Steps: Optimizing Sampling Schedules in Diffusion Models
>
> [3] Shih et al. (2023) – Parallel Sampling of Diffusion Models
>
> [4] Karras et al. (2022) – Elucidating the Design Space of Diffusion-Based Generative Models
>
> [5] Salimans et al. (2024) – Multistep Distillation of Diffusion Models via Moment Matching
>
> >  [is the] method actually orthogonal to other accelerations. This is tricky as many few-NFE methods are based on ODEs. [...] setup you could try where you could achieve comparable FID with existing acceleration baselines using [...]?
>
> It is indeed correct that our approach does not apply to deterministic samplers, but we adapt common acceleration methods to the stochastic case. For example in the case of LD3 [1] we train LD3 in our setting by considering the noise added during the sampling to be frozen. Similarly, in [2] (see end of Section 3.1), the authors freeze the noise. Following those principles we are able to combine several acceleration techniques with our approach.
>
> [1] Sabour et al. (2024) – Align Your Steps: Optimizing Sampling Schedules in Diffusion Models
>
> [2] Shih et al. (2023) – Parallel Sampling of Diffusion Models

---

> > ### Comment · Reviewer_eBDt · 2025-04-05
> >
> > Thanks for the detailed response and additional experiments. My sense is that there are diminishing returns to LD3 in the >10 NFE regime, but these preliminary results do suggest that speculative sampling may offer complementary benefits. I'll raise my score to 3.

---

### Official Review · Reviewer_zMQr · 2025-03-13

**Overall Recommendation:** 4

**Summary:**

The paper provides an efficient way to apply speculative decoding from literature of language models to diffusion models, which is challenging because diffusion models use the gaussian distribution instead of discrete distribution. They address this challenge via adjusted rejection sampling with reflection coupling. Experimental results show that speculative sampling actually accelerates sampling from diffusion models even without separate draft models.

**Claims And Evidence:**

- Claim 1: To extend speculative decoding to diffusion models, they propose an efficient rejection sampling using reflection maximal coupling.
  - Evidence: Section 3.2-3.3 provide concise derivations.
  - Evidence: Efficiency is measured by the number of function calling for the target model in Section 6.
- Claim 2: They propose an efficient draft model constructed from the target model itself, by reusing the previous score from target models.
  - Evidence: Figure 3 experimentally shows it is indeed more efficient than the independent draft models.
  - Evidence: The actual speed-ups are also observed on realistic datasets like CIFAR10 and LSUN.
- Claim 3: They provide a complexity analysis on the number of function evaluation, and also analyses for acceptance ratio for speculative sampling.
  - Evidence: Proposition 4.1, Theorem 4.3.

Overall, their claims are well-supported by theory and experiments. Thus I think the contributions of the paper are solid and beneficial to the community of both speculative decoding and diffusion models.

**Essential References Not Discussed:**

N/A

**Experimental Designs Or Analyses:**

Experimental designs make sense.

**Methods And Evaluation Criteria:**

The proposed method makes sense. Evaluation protocols also follow the standard ones.

**Other Comments Or Suggestions:**

N/A

**Other Strengths And Weaknesses:**

See above discussions.

**Questions For Authors:**

- How much does the proposed method accelerate inference of diffusion models when combined with other acceleration methods?

**Relation To Broader Scientific Literature:**

Speculative sampling has been widely used with language models, and it effectively accelerates the inference of large language models. The proposed method and drafting strategy provides such a powerful tool for diffusion models with a straightforward and efficient way, which opens up a new research direction for inference acceleration or diffusion models.

**Theoretical Claims:**

I checked all statements of the paper. I read some proofs given in Appendix, especially in relation to Proposition 4.1, Theorem 4.3.

---

> ### Author Rebuttal · Authors · 2025-03-31
>
> We would like to thank the reviewer for their very positive assessment of our manuscript.
>
> > Overall, their claims are well-supported by theory and experiments. Thus I think the contributions of the paper are solid and beneficial to the community of both speculative decoding and diffusion models.
>
> We appreciate that the reviewer recognizes the value of our work and its benefits for both the LLM and diffusion models communities.
>
> > How much does the proposed method accelerate inference of diffusion models when combined with other acceleration methods?
>
> Below, we combine our speculative sampling method with parallel sampling [1].
>
> | Configuration | FID |  NFE |
> | --- | --- | --- |
> | $p=0$, $\varepsilon=0.25$, $\tau=1.0$ | 2.23 | 15.69 |
> | $p=1$, $\varepsilon=0.25$, $\tau=1.0$ | 2.09 | 23.80 |
> | $p=5$, $\varepsilon=0.25$, $\tau=1.0$ | 2.09 | 57.85 |
> | $p=0$, $\varepsilon=0.5$, $\tau=1.0$ | 2.77 | 17.06 |
> | $p=1$, $\varepsilon=0.5$, $\tau=1.0$ | 2.75 | 23.42 |
> | $p=5$, $\varepsilon=0.5$, $\tau=1.0$ | 2.75 | 57.80 |
> | $p=0$, $\varepsilon=0.25$, $\tau=2.0$ | 2.24 | 14.89 |
> | $p=1$, $\varepsilon=0.25$, $\tau=2.0$ | 2.09 | 21.12 |
> | $p=5$, $\varepsilon=0.25$, $\tau=2.0$ | 2.08  | 51.45 |
> | $p=0$, $\varepsilon=0.5$, $\tau=2.0$ | 2.74 | 16.47 |
> | $p=1$, $\varepsilon=0.5$, $\tau=2.0$ | 2.77 | 20.62 |
> | $p=5$, $\varepsilon=0.5$, $\tau=2.0$ | 2.77 | 50.40 |
> | $p=0$, $\varepsilon=0.25$, $\tau=10.0$ | 2.39 | 12.86 |
> | $p=1$, $\varepsilon=0.25$, $\tau=10.0$ | **2.07** | 15.42 |
> | $p=5$, $\varepsilon=0.25$, $\tau=10.0$ | 2.07 | 37.5 |
> | $p=0$, $\varepsilon=0.5$, $\tau=10.0$ | 2.73 | 14.49 |
> | $p=1$, $\varepsilon=0.5$, $\tau=10.0$ | 2.79 | 16.38 |
> | $p=5$, $\varepsilon=0.5$, $\tau=10.0$ | 2.79 | 40.25 |
>
>
> We report FID score and NFE for CIFAR-10 with a number of steps of $30$. We vary the temperature parameter $\tau$, the churn parameter $\varepsilon$ as well as the number of parallel iterations, see [1,2]. For each combination of hyperparameters we also consider window sizes $5$, $10$ and $20$ and report the best run (in terms of FID).
>
> The original speculative sampling procedure corresponds to $p=0$. The best FID number that can be achieved with this configuration is $2.23$ with a NFE of $15.69$. However, by combining our speculative sampling procedure with parallel sampling then we can reach a FID of $2.07$ with a NFE of $15.42$.
> This shows the benefits of combining our speculative sampling procedure with other acceleration methods. We will report those results in the updated version of our manuscript.
>
> [1] Shih et al. (2023) – Parallel Sampling of Diffusion Models
>
> [2] Tang et al. (2024) – Accelerating Parallel Sampling of Diffusion Models
>
>
> > Captions for the tables can be improved.
>
> We are willing to improve the captions of our table if the reviewer has more feedback for us.
>
> > Versatility of the method
>
> We have now included additional experiments regarding comparison with DPM++ solvers, see answer to Reviewer eBDt, as well as an experiment showcasing the efficiency of the method in a sampling context (i.e. a non generative modeling setting), see answer to Reviewer E771.

---

### Official Review · Reviewer_E771 · 2025-03-19

**Overall Recommendation:** 3

**Summary:**

This work introduces a speculative sampling method for efficient diffusion models using reflection maximal coupling. Instead of relying on a separate draft model, they propose an approach that generates drafts directly from the target model. A complexity analysis establishes a lower bound on acceptance ratios. The experiments show speed-ups in image generation and robotics policy generation tasks.
The rejection implementation proposed leverages the coupling between two Gaussian distribution with different means but identical variances applicable to diffusion models.

A relevant problem statement.

The manuscript well written. Captions for the tables can be improved.

Supplementary material Section C – Algorithm 7. The return statement on Line 980 is probably incorrect.

**Claims And Evidence:**

The general claim of  parallel of parallel sampling and selection helps in the case parallel evaluation but with increased compute and memory.

The lower bound presented in Section 4.2, intuitively makes sense in terms of selected parameters.

**Essential References Not Discussed:**

check above sections.

**Ethical Review Concerns:**

None.

**Experimental Designs Or Analyses:**

Check methods and evaluation criteria section.

**Methods And Evaluation Criteria:**

-	The evaluation is conducted using CIFAR10 and LSUN datasets. More through evaluation is needed using ImageNet-1k classes.
-	The approach assumes parallel evaluation allowing lower latency, but the method increases compute and memory requirements, which are also critical components that are not handled by the approach.
-	Need to compare with other approaches that allow efficient sampling of diffusion models.
o	“Fast Sampling of Diffusion Models via Operator Learning”
o	“Parallel Sampling of Diffusion Models”
o	“Accelerating Parallel Sampling of Diffusion Models”

**Other Comments Or Suggestions:**

The manuscript well written. Captions for the tables can be improved.

Supplementary material Section C – Algorithm 7. The return statement on Line 980 is probably incorrect.

**Other Strengths And Weaknesses:**

None.

**Questions For Authors:**

None.

**Relation To Broader Scientific Literature:**

Good initial step for speculative sampling for diffusion models.

**Theoretical Claims:**

See above sections.

---

> ### Author Rebuttal · Authors · 2025-03-31
>
> We would like to thank the reviewer for their positive assessment of our paper.
>
> > The approach assumes parallel evaluation allowing lower latency, but the method increases compute and memory requirements, which are also critical components that are not handled by the approach.
>
> Our approach indeed trades memory for latency. We believe this trade-off is advantageous for certain applications. For example, in applications requiring extremely low latency, increased memory consumption is often acceptable. Note that this fact is also emphasized in [1] “Currently, for large models like SD, ParaTAA requires the use of multiple GPUs to achieve considerable speedup in wall-clock time. Nonetheless, as advancements in GPU technology and parallel computing infrastructures evolve, we anticipate that the cost will be significantly lower”
>
> [1] Tang et al. (2024) – Accelerating Parallel Sampling of Diffusion Models
>
> > The evaluation is conducted using CIFAR10 and LSUN datasets. More thorough evaluation is needed using ImageNet-1k classes.
>
> We have now evaluated our method on ImageNet (64x64x3) and show similar improvements in terms of NFE. With a baseline at 250 steps (corresponding to a FID/IS of 4.18/49.54), our method achieves a NFE of  95.38 corresponding to a halving of the NFE. The FID/IS score of the speculative sampling approach is 4.35/48.28 (which is of the same order as 4.18/49.54). We acknowledge that our baseline FID/IS is not state-of-the-art, but this was the only model we had the time to retrain by the rebuttal deadline. We are currently training another model in order to improve the base FID/IS score.
>
> The parameters of the base sampler are given by $\varepsilon = 0.25$ and the guidance value is set to $0.25$. In our speculative  sampling experiment we consider a temperature of $\tau=1.0$ and a window size of $10$.
>
> > Need to compare with other approaches that allow efficient sampling of diffusion models. o “Fast Sampling of Diffusion Models via Operator Learning” o “Parallel Sampling of Diffusion Models” o “Accelerating Parallel Sampling of Diffusion Models”
>
> We do not compare with [1] as this approach is very different from ours and does not yield state-of-the-art results. However, we have now compared with [2] and refer to the answer to Reviewer zMQr for more details. In short, our method outperforms [2] and, combining our speculative approach with [2], leads to improved results which also outperforms strong baselines such as [4,5]. However, we emphasize that our approach is not directly comparable to [2,3], since 1) these methods do not sample exactly from the original denoising diffusion models and 2) only converge in the infinite number of steps limit since they rely on the fixed point property of the SDE integrators.
>
> [1] Zheng et al. (2022) – Fast Sampling of Diffusion Models via Operator Learning
>
> [2] Shih et al. (2023) – Parallel Sampling of Diffusion Models
>
> [3] Tang et al. (2024) – Accelerating Parallel Sampling of Diffusion Models
>
> [4] Tong et al. (2024) – Learning to Discretize Denoising Diffusion ODEs
>
> [5] Sabour et al. (2024) – Align Your Steps: Optimizing Sampling Schedules in Diffusion Models
>
>
> > Supplementary material Section C – Algorithm 7. The return statement on Line 980 is probably incorrect.
>
> Thanks, this is indeed a typo.
>
> > Versatility of the method
>
> To showcase the applicability of our method beyond generative modeling, we demonstrate its  efficiency in the context of Monte Carlo sampling. We consider the $\phi_4$ model [1,2] which defines $\pi(x) \propto \exp[-U_\beta(x)]$ for
>
> $$U_\beta(x) = (\beta/2) \sum_{|i-j| = 1} (x_i - x_j)^2 + \sum_{i} (x_i^2 - 1)^2,$$
>
> on a grid of shape $(8,8)$ and $\beta=100$. Sampling from $\pi$ is complex as this requires sampling so-called ordered states. In this context, the teacher model is the Langevin diffusion sampling $\pi(x)$ with $100,000$ iterations and stepsize $10^{-3}$ while our speculative sampling algorithm uses the frozen prediction draft model and a window size of $20$. The results are averaged over $500$ runs. The mean energy is the average of $U(x_i)$ computed on the $500$ samples $x_i$, while the standard deviation energy is the standard deviation of $U(x_i)$.
>
>  | Metrics | Mean energy | Standard deviation energy | NFE |
> | --- | --- | --- |--- |
> | Langevin sampling | 62.27 | 13.32 | 100000 |
> | Speculative sampling| 65.90 | 12.48 | 48564 |
>
>  [1] Guth et al. (2022) – Wavelet Score-Based Generative Modeling
>
> [2] Milchev et al. (1986) – Finite-size scaling analysis of the $\phi_4$ field theory on the square lattice

---

### Decision · Program_Chairs · 2025-05-01

**Decision:**

Accept (poster)

**Comment:**

This paper proposes extending speculative sampling to diffusion models through a reflection-based coupling scheme. The reviewers agreed that adapting speculative sampling to continuous sampling in diffusion models is a valuable contribution, both conceptually and practically. In particular, they praised the clever application of reflection maximal coupling as well as the thorough theoretical analysis bounding the acceptance ratio.

Some reviewer also suggested to refinie the technique and comparing it against additional fast-diffusion samplers. Some also highlighted trade-offs in memory usage and recommended exploring synergy with other acceleration methods (such as parallel sampling). Overall this is a nice contribution and the authors are encouraged to incorporate the reviewers suggestions for the final version, should the paper make the cut.